# HPV infection alters vaginal microbiome through down-regulating host mucosal innate peptides used by *Lactobacilli* as amino acid sources

Alizee Lebeau[1,14], Diane Bruyere[1,14], Patrick Roncarati[1], Paul Peixoto [2,3], Eric Hervouet [2,3], Gael Cobraiville[4], Bernard Taminiau [5], Murielle Masson [6], Carmen Gallego[7], Gabriel Mazzucchelli[8], Nicolas Smargiasso[8], Maximilien Fleron[8,9], Dominique Baiwir [8,9], Elodie Hendrick[1], Charlotte Pilard[1], Thomas Lerho[1], Celia Reynders[1], Marie Ancion[1], Roland Greimers[10], Jean-Claude Twizere [11], Georges Daube [5], Geraldine Schlecht-Louf [7], Françoise Bachelerie[7], Jean-Damien Combes[12], Pierrette Melin [13], Marianne Fillet[4], Philippe Delvenne[1,10], Pascale Hubert[1] & Michael Herfs [1✉]

Despite the high prevalence of both cervico-vaginal human papillomavirus (HPV) infection and bacterial vaginosis (BV) worldwide, their causal relationship remains unclear. While BV has been presumed to be a risk factor for HPV acquisition and related carcinogenesis for a long time, here, supported by both a large retrospective follow-up study ($n = 6,085$) and extensive in vivo data using the K14-HPV16 transgenic mouse model, we report a novel blueprint in which the opposite association also exists. Mechanistically, by interacting with several core members (NEMO, CK1 and β-TrCP) of both NF-κB and Wnt/β-catenin signaling pathways, we show that HPV E7 oncoprotein greatly inhibits host defense peptide expression. Physiologically secreted by the squamous mucosa lining the lower female genital tract, we demonstrate that some of these latter are fundamental factors governing host-microbial interactions. More specifically, several innate molecules down-regulated in case of HPV infection are hydrolyzed, internalized and used by the predominant *Lactobacillus* species as amino acid source sustaining their growth/survival. Collectively, this study reveals a new viral immune evasion strategy which, by its persistent/negative impact on lactic acid bacteria, ultimately causes the dysbiosis of vaginal microbiota.

[1] Laboratory of Experimental Pathology, GIGA-Cancer, University of Liege, Liege, Belgium. [2] INSERM, EFS BFC, UMR 1098, Interactions Hôte-Greffon-Tumeur/Ingénierie Cellulaire et Génique, University of Bourgogne Franche-Comté, Besançon, France. [3] EPIGENEXP platform, University of Bourgogne Franche-Comté, Besançon, France. [4] Laboratory for the Analysis of Medicines, Center for Interdisciplinary Research on Medicines (CIRM), University of Liege, Liege, Belgium. [5] Department of Food Sciences-Microbiology, Fundamental and Applied Research for Animals and Health (FARAH), Faculty of Veterinary Medicine, University of Liege, Liege, Belgium. [6] Ecole Supérieure de Biotechnologie Strasbourg, UMR 7242, CNRS, University of Strasbourg, Illkirch, France. [7] INSERM UMR 996, Inflammation Microbiome and Immunosurveillance, University of Paris-Saclay, Clamart, France. [8] Laboratory of Mass Spectrometry, Department of Chemistry, University of Liege, Liege, Belgium. [9] GIGA Proteomic Facility, University of Liege, Liege, Belgium. [10] Department of Pathology, University Hospital Center of Liege, Liege, Belgium. [11] Laboratory of Signaling and Protein Interactions, GIGA-Molecular Biology of Diseases, University of Liege, Liege, Belgium. [12] Infections and Cancer Epidemiology Group, International Agency for Research on Cancer, World Health Organization, Lyon, France. [13] Department of Clinical Microbiology, University Hospital Center of Liege, Liege, Belgium. [14]These authors contributed equally: Alizee Lebeau, Diane Bruyere. ✉email: M.Herfs@uliege.be

Affecting at any point of time more than 300 million individuals worldwide, human papillomavirus (HPV) is the most common sexually transmitted infection[1]. To date, over 225 genotypes have been fully characterized and about one fifth, belonging to the alpha genus, can be detected in the anogenital mucosa. Although most infections are cleared or maintained in an asymptomatic or latent state by the immune system, carcinogenic (high-risk) HPV strains (most notably HPV16 and 18) cause virtually all squamous intraepithelial lesions [low-grade (LSIL) and high-grade (HSIL)] and cell carcinoma (SCC) arising from the uterine cervix as well as a large fraction (~50%) of vaginal/vulvar (pre)cancers. In total, HPV infections account for ~5% of the worldwide cancer burden with an estimated 550,000 new cases diagnosed annually in the lower genital tract[2,3]. The persistence of an active infection for years or decades indicates that these viruses have evolved a number of mechanisms to escape both innate and adaptive immune responses. Indeed, by directly interacting with some core proteins or by indirectly altering their activity (post-translational modifications) or their gene expression pattern (promoter hypermethylation and histone modifications), viral E6 and E7 oncoproteins have been especially shown to antagonize the cGAS-STING DNA sensing pathway[4], to suppress the interferon secretion and signaling[5–7], to impair Toll-like receptor 9 and major histocompatibility complex class I transcription[8,9] and to reduce chemotactic and proinflammatory gene expression[10].

In contrast to the skin and the gut which are colonized by a complex microbiome, the human vaginal ecosystem is associated with a low microbial diversity largely dominated (>90%) by a few *Lactobacillus* species (mainly *L. crispatus*, *L. jensenii* and *L. iners*)[11–13]. Characterized by the replacement of the normally dominant lactic acid bacteria by a more diverse bacterial mixture predominated by *Gardnerella vaginalis* and other anaerobic bacteria (e.g., *Atopobium vaginae*, *Prevotella_ge*, *Mobiluncus_ge*, *Sneathia_ge*,…)[14], bacterial vaginosis (BV) is a common vaginal disorder among women of reproductive age. Although at least 50% of women with BV are asymptomatic, this microbial imbalance (also called dysbiosis) can manifest clinically by a vaginal discharge, the presence of clue cells (recognized by cytologic review) and a "fishy" odor related to the production of volatile amines by anaerobes[15]. Three decades of epidemiologic studies reported the multiplicity of sexual partners, African descent, vaginal douching and cigarette smoking as risk factors for the acquisition of BV[15]. However, none of these latter on their own can reliably explain the prevalence of this condition and the etiopathogenesis of BV remains unclear[15,16]. Most likely, this imbalance in the vaginal flora is multifactorial and involves complex interactions between extrinsic factors, the different species of bacteria constituting the endogenous vaginal microbiome and the host mucosa.

In addition to causing symptoms for some women, BV has been shown to increase the risk of preterm delivery[17] as well as gynecologic complications such as endometritis, cervicitis and postoperative pelvic infections[18]. Moreover, both the rise in the vaginal pH and the reduced level of hydrogen peroxide ($H_2O_2$) resulting from the low abundance of *Lactobacilli* is presumed to promote the acquisition of both bacterial (e.g., *Chlamydia trachomatis* and *Neisseria gonorrhoeae*) and viral (e.g., herpes simplex virus type 2, HIV and HPV) sexually transmitted pathogens[19–22]. The degradation of the protective mucus barrier through the sialidase activity of anaerobic micro-organisms could also contribute to this latter susceptibility for developing infections in the lower genital tract[23]. Regarding HPV infections, besides favoring their acquisition, it is generally considered that the oxidative stress resulting from microbial dysbiosis also promotes the subsequent progression of HPV-positive (pre)

neoplastic lesions. In agreement, recent evidence has shown that an anaerobic vaginal microbiome composition is associated with a lower regression rate of HPV-related diseases[24]. Despite these important findings, the interplay/association between BV and HPV infection is still unclear. Indeed, while longitudinal studies clearly reported increased rates of incident HPV among BV-positive women (for a meta-analysis, see[25]), the risk of BV occurrence following HPV infection has not been systematically pursued/calculated. Overall, data related to the influence of HPV on vaginal microbiome are limited and, as mentioned in several review articles[26,27], when BV and HPV coexist, we cannot exclude that, in a significant proportion of women, the viral infection preceded in time and, by altering the host mucosa secretome, ultimately caused BV development.

The present multi-approach study shows a causal relationship between HPV infection and BV. Mechanistically, the drastic down-regulation of host defense peptides, related to the interactions of HPV E7 oncoprotein with several key proteins (NEMO, CK1, β-TrCP) involved in both NF-kB and Wnt/β-catenin signaling pathways, has been shown to be instrumental in this process. Indeed, we unexpectedly uncovered that the innate molecules most secreted by the vaginal/cervical mucosa do not display any antimicrobial activity on *Lactobacillus* species but rather, are cleaved and used as amino acid source by these lactic acid bacteria, sustaining their growth/survival. The accumulating (retrospective clinical and in vitro) data have been finally confirmed in vivo using the K14-HPV16 transgenic mouse model in which (pre)cancer development was associated with a vaginal dysbiosis.

## Results

**A large retrospective follow-up analysis including over 6,000 patients identifies a two-way interaction between HPV infection and BV development.** In the last decade, several systematic reviews of the literature (and meta-analysis) clearly indicated that HPV infection and BV are epidemiologically related[25,26,28]. However, some uncertainties still exist concerning the temporal sequence between these two pathological conditions. Indeed, while BV has been considered as a risk factor for HPV acquisition/persistence for a long time, the inverse relationship remains unclear. To address this important issue, a retrospective cohort study including women who underwent at least 2 Pap smear screenings over an 8-year period was performed. At each visit, complete data related to a potential abnormal cytology result (Bethesda classification), the existence of BV (Hay/Ison grading system) and high-risk HPV infection (Abbott RealTime HPV assay) were available for all enrolled patients. At the first visit, 3,481 out of 6,085 (57.2%) patients did not display any evidence of HPV infection and/or BV. A mixed bacterial flora (associated or not with the presence of clue cells) was reported in about one third (2,053/6,085, 33.7%) of patients with normal cytology (HPV negative). Single or multiple HPV infection was detected in 262 (4.3%) patients with a morphologically normal cervico-vaginal flora. At last, 289 (4.8%) patients were simultaneously positive for high-risk HPV and BV. The general characteristics of the four defined groups and the geographic repartition of patients are summarized in Fig. 1a and Supplementary Fig. 1, respectively. Of note, patients' age (distribution) was not significantly different between the groups. No obvious difference in terms of geographic distribution of patients was noticed either. Overall, 12,390 follow-up visits were completed by the selected women (median follow-up time: 66.3 months) and, in order to precisely determine the temporal relationship between BV and genital HPV infection, the probability of acquiring one condition when the other one was already present or not was estimated. During follow-up visits, a positive HPV test (associated with cytological abnormalities)

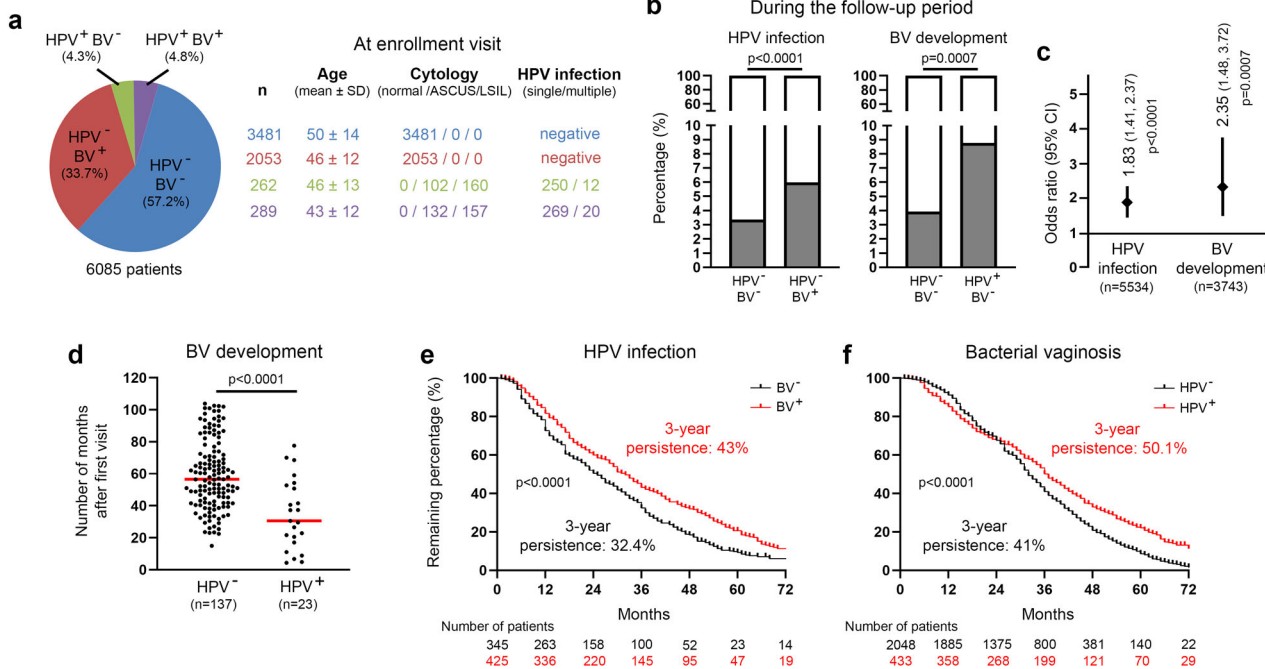

**Fig. 1 Retrospective clinical follow-up analysis evaluating the relationship between genital HPV infection and BV occurrence/persistence. a** General characteristics of the study population (n = 6,085). **b** Probability of high-risk HPV infection or BV development during the follow-up period according to the status for the other gynecological disorder at first (enrollment) visit. **c** Forest plot showing the odds ratio (OR) and 95% confidence intervals (CI) for developing one pathological condition when the other one was preceding in time [HPV infection (OR: 1.83, 95% CI: 1.41–2.37); BV development (OR: 2.35, 95% CI: 1.48–3.72)]. **d** Number of months for BV occurrence in HPV-positive and -negative patients. **e** Kaplan-Meier estimates for the persistence of HPV infection according to the BV status (negative *versus* positive). The clearance of HPV infections was ascertained by both cytology (Bethesda system) and PCR-based HPV test (Abbott High-risk HPV assay). **f** Kaplan–Meier curve for the persistence of BV according to the HPV status. Two consecutive negative results (using Hay/Ison grading system) at least 12 months apart were required to consider a patient really/durably cured of BV. *P* values were determined using two-sided Fisher's exact test **b**, **c**, two-sided unpaired *t*-tests (**d**) and log-rank (Mantel-Cox) test **e**, **f**. Source data are provided as a Source Data file.

and BV was observed in 117 (117/3,481, 3.4%) and 137 (137/3,481, 3.9%) control women (negative for HPV and BV at first visit), respectively. Eight patients were positive for both HPV and BV at the same follow-up visit. Importantly, the development of each disorder was significantly more frequently diagnosed when the other one was preceding in time [HPV: 123/2,053, 6% (OR: 1.83, *p* < 0.0001); BV: 23/262, 8.8% (OR: 2.35, *p* < 0.001)] (Fig. 1b, c). These latter results were not adjusted for age, race, or other potential confounders [e.g., socioeconomic status of patients or number of sex partners (due to the lack of available information)]. In parallel, the median time for BV occurrence in HPV-infected and -uninfected women was investigated and, interestingly, a significant difference (34.53 months *versus* 59.44, *p* < 0.0001) was observed (Fig. 1d). To further evaluate the interplay between HPV and BV, the persistence of HPV infection according to the BV status (Fig. 1e) as well as the inverse evaluation (Fig. 1f) were determined. Out of 791 HPV-positive patients (diagnosed at first visit or during follow-up), 21 (2.7%) were excluded from the present longitudinal analysis because they underwent examinations in a different hospital following HPV diagnosis and, therefore, no data allowing to assess the persistence of the viral infection were available in our records. For a similar reason, 0.8% (21/2,502) of BV-positive patients were not taken into consideration either. As shown in Fig. 1e, the duration of HPV infections was significantly longer in BV-positive women compared to their counterparts displaying a normal vaginal microbiome dominated by *Lactobacillus spp* (persistent infections after 3 years: 43% *versus* 32.4%, *p* < 0.0001). Remarkably, the opposite observation was also made (Fig. 1f). Indeed, the 3-year BV persistence was 50.1% for HPV-positive patients, as opposed to 41% for uninfected individuals (*p* < 0.0001).

**HPV oncoproteins impair innate (antimicrobial) peptide expression in vaginal/cervical squamous mucosa.** By the constitutive or inducible production of soluble molecules (especially innate peptides), the host mucosa actively participates to the regulation of bacterial flora. These complex host-microbiota interactions are still the subject of intense investigations and are very likely specific for each organ as suggested by the important disparities between each microbiome and mucosal surface (skin, gut, oral, vaginal). According to the antimicrobial peptide database (http://aps.unms.edu), the squamous epithelial cells (keratinocytes) lining the lower part of the gynecologic tract secrete a dozen of innate (antimicrobial) peptides[29,30]. By laser capture microdissection, 44 independent frozen human tissue specimens (11 normal squamous epithelia, 10 LSIL, 10 HSIL, and 13 SCC) were sampled. In order to avoid bias related to HPV status, all (pre)neoplastic lesions displayed diffuse (basal or full-thickness) p16INK4a immunoreactivity in keeping with carcinogenic HPV infection. To allow the comparison of the expression level of all analyzed host defense peptides, the amplification efficiency of each qPCR reaction was determined (Fig. 2a). As shown in Fig. 2b, c, with the exception of HβD3, all antimicrobial peptides were down-regulated in HPV-positive lesions compared to normal squamous vaginal/ectocervical epithelium. The inhibition of "defensin-like" peptides (mainly S100A7 and elafin) was especially considered as essential given that, in normal/uninfected conditions, these latter were up to 1000 fold (>10 Ct) more expressed than the epithelial members of the defensin family. Similar reduced expressions of both defensins and "defensin-like" peptides in HPV-positive (pre)neoplastic lesions were also observed at the protein level, as shown by immunohistochemical

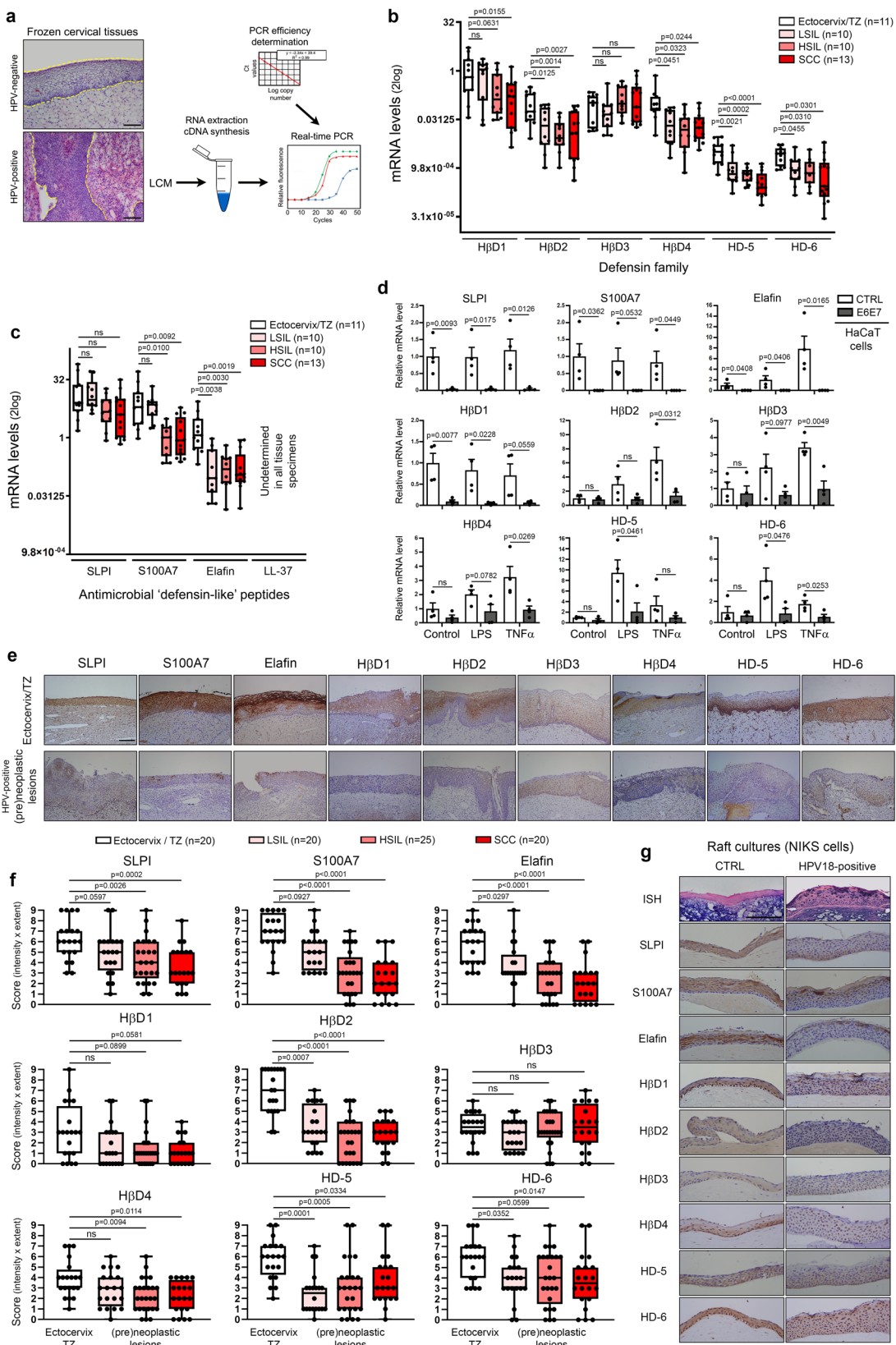

experiments (Fig. 2e, f). Regarding LL-37, this cathelicidin-related antimicrobial peptide was undetectable both at the mRNA and protein levels (Fig. 2c and Supplementary Fig. 2). It is interesting to notice that the morphologically normal (p16$^{INK4a}$-negative) squamous epithelium adjacent to HPV-positive (pre)neoplastic lesions also displayed a significantly reduced expression of several

innate peptides (S100A7, elafin, HβD2, HβD4 and HD-5) (Supplementary Fig. 3). In order to determine whether or not HPV (and its viral oncoproteins) is directly responsible for the down-regulation of defensin(-like) peptides observed in (pre)neoplastic lesions, human keratinocytes were stably transduced with HPV16 E6E7 oncogenes and the expression level of each defense peptide

**Fig. 2 HPV oncoproteins impair both the constitutive and inducible innate (antimicrobial) peptide expression. a** Schematic illustration of the steps involved in gene expression analysis using microdissected frozen specimens. mRNA expression of epithelial-specific members of the defensin **b** and "defensin-like" **c** families was measured by RT-qPCR in HPV-positive (pre)neoplastic lesions (LSIL, $n = 10$; HSIL, $n = 10$; SCC, $n = 13$). Uninfected squamous samples [ectocervix, vagina, transformation zone (TZ)] ($n = 11$) were used as control. Gene expression was normalized using four calibrator genes (HPRT, GAPDH, 18S and TBP). Box limits: 25th to 75th percentiles; line: median; whiskers: minimum to maximum. **d** RT-qPCR analysis of SLPI, S100A7, elafin, HβD1-4 and HD-5/6 expression in immortalized keratinocytes stably transduced or not with HPV16 E6/E7 oncoproteins and stimulated with TNFα or LPS. Each experiment was normalized to the amount of HPRT mRNA from the same sample. Results represent the means ± SEM of four independent experiments. **e** HPV-negative and positive tissue specimens stained for antimicrobial peptides expressed by the squamous epithelium lining the lower part of the female reproductive tract. A reduced immunoreactivity for most of these innate factors was clearly observed in HPV-infected (pre)neoplastic lesions. **f** Semi-quantitative evaluation of innate peptide expression (intensity and extent of the immunostainings) in both normal epithelium from HPV-negative samples ($n = 20$) and (pre)neoplastic lesions (LSIL, $n = 20$; HSIL, $n = 25$; SCC, $n = 20$). Box limits: 25th to 75th percentiles; line: median; whiskers: minimum to maximum. **g** Representative control and HPV18-positive organotypic raft culture sections stained for all analyzed host defense (antimicrobial) peptides. As a control, HPV DNA was detected by in situ hybridization and, consistent with an episomal infection, a diffuse punctate pattern was observed. Images are representative of three independent experiments. The scale bar represents 100 μm. *P* values were determined using one-way ANOVA followed by Dunnett's multiple comparison post-hoc test **b**, **c**, two-sided unpaired *t*-tests **d** and ANOVA Kruskal–Wallis test followed by Dunn post-hoc test **f**. ns: not significant ($p > 0.1$). Source data are provided as a Source Data file.

potentially expressed by the squamous mucosa and being part of the innate immune barrier of the female lower genital tract was investigated (Fig. 2d). HβD2-4 expression was shown to be highly stimulated by TNFα whereas LPS was more efficient to induce HD-5/6 expression. Significantly, not only did the presence of viral oncoproteins greatly impair (up to 7 Ct, 128 fold) the expression of so called "constitutive" peptides (SLPI, S100A7, elafin and HβD1), but also the induction of HβD2-4 as well as HD-5/6 following the exposure to TNFα/LPS was significantly altered. A drastic reduced secretion of "constitutive" peptides in case of HPV16 E6E7 transduction was also reported in cell supernatants (ELISA) (Supplementary Fig. 4). In parallel, we utilized a second in vitro model mimicking more closely the early steps of the natural infection with carcinogenic HPV (Fig. 2g). Using organotypic raft cultures, keratinocytes maintaining episomal HPV18 genome displayed weaker immunoreactivities for SLPI, S100A7, elafin, HβD2 and HβD4 compared to uninfected cells. The other analyzed peptides were down-regulated at a lower extent.

**HPV impairs TNFα/LPS-induced innate peptide expression through E7-dependent NEMO degradation and subsequent suppression of NF-κB activation.** Proinflammatory factors such as TNFα, IL-1β and LPS have been previously shown to induce the expression of elafin as well as several members of the defensin family[31–33]. Given these results and the identification of putative NF-κB/p65 binding sites within the promoters of *DEFB2-4*, *DEFA5-6* and *PI3* (elafin) genes [estimations made using the Eukaryotic promoter database[34]], we first evaluated the requirement of NF-κB signaling pathway activation in TNFα/LPS-induced innate peptide expression. As expected, knockdown of p65 with siRNA or indirectly via blockade of the degradation of IκBα (BAY 11-7082) resulted in a significant decrease of defensin/elafin mRNA levels in normal/uninfected keratinocytes (Fig. 3a, b). To evaluate the potential alteration of NF-κB in case of HPV infection as well as the role of each individual viral oncoprotein, the occupancy of NF-κB binding sites on both *PI3/elafin* and *DEFB2* promoters was analyzed by ChIP in keratinocytes stably transduced with HPV16 E6 or E7. As shown in Fig. 3c, upon TNFα stimulation, a weaker occupancy of both gene promoters by p65 was reported in E7-positive cells. E6 viral oncoprotein did not seem to disrupt NF-κB signaling pathway. In order to determine the mechanism underlying E7-dependent alteration of NF-κB activation, the presence of both TNFα (TNFR1-2) and LPS (CD14 and TLR4) receptors at the cell membrane was first analyzed by flow cytometry. No significant difference was observed between cells transduced or not with HPV E6/E7 viral oncogenes

(Fig. 3d). IκBα degradation and p65 nuclear translocation upon TNFα stimulation were then assessed. Interestingly, the presence of E7 oncoproteins strongly reduced IκBα degradation (Fig. 3e), explaining the cytoplasmic sequestration of p65 (Fig. 3f, g). Further confirming these data, the percentage of epithelial cells displaying nuclear p65 immunoreactivity was significantly lower in HPV-positive cancers compared to their viral-unrelated counterparts (Fig. 3h). Based on these clear-cut results, the direct interaction of HPV16 E7 oncoprotein with each protein members of the IKK kinase complex was evaluated using *Gaussia princeps* luciferase complementation assay (GPCA). Strikingly, a strong interaction between HPV16 E7 and NEMO was highlighted and confirmed by co-IP in both directions (Fig. 3i–j and Supplementary Fig. 5). Similar results were obtained with HPV E7 from several other genotypes (high-risk alpha: HPV18, 33 and 39; beta: HPV8, 38 and 49) (Supplementary Fig. 6). In order to further characterize this interaction between E7 and IKK regulatory subunit (NEMO), three truncated/mutated forms of HPV16 E7 were also tested: the CR1 + CR2 region (consisting of 1–36 amino acids), the C-terminal domain (37–98 amino acids) and the C24G/E26G construct mutated within the LxCxE motif. As shown in Fig. 3k, the GPCA signal was drastically reduced with the CR1 + CR2 construct, supporting that NEMO interacts with the C-terminal region of E7. Finally, protein stability/half-life was measured in cultured cells after treatment with a protein synthesis inhibitor (cycloheximide). In contrast to the high NEMO stability observed in normal cells, HPV16 E7 oncoprotein led to a marked degradation of this latter protein (Fig. 3l).

**HPV E7 oncoprotein inhibits constitutive expression of both elafin and S100A7 through promoting β-catenin stabilization/signaling and subsequent up-regulation of c-myc.** Based on data collected by several approaches (e.g., microarray gene expression profiling or ChIP-sequencing), it is estimated that ~15% of all human genes are regulated (positively or negatively) by the oncogenic protein c-myc[35]. Interestingly, both elafin and S100A7, which are drastically down-regulated in HPV-infected tissues, are listed within the high-affinity group of c-myc targets[36]. These latter results obtained by high-throughput screening were, however, never confirmed/validated. Highlighting the involvement of this transcriptional factor in elafin/S100A7 repression, by silencing c-myc (Fig. 4a) or chemically repressing its dimerization with Max (Fig. 4b), we significantly and strongly restored both elafin and S100A7 expression in E6E7-transduced cells. Importantly, close results were observed by using siRNA targeting β-catenin (Fig. 4a), a key factor involved in the Wnt signaling pathway and well-known to activate c-myc transcription[37]. In order to evaluate the potential impact of

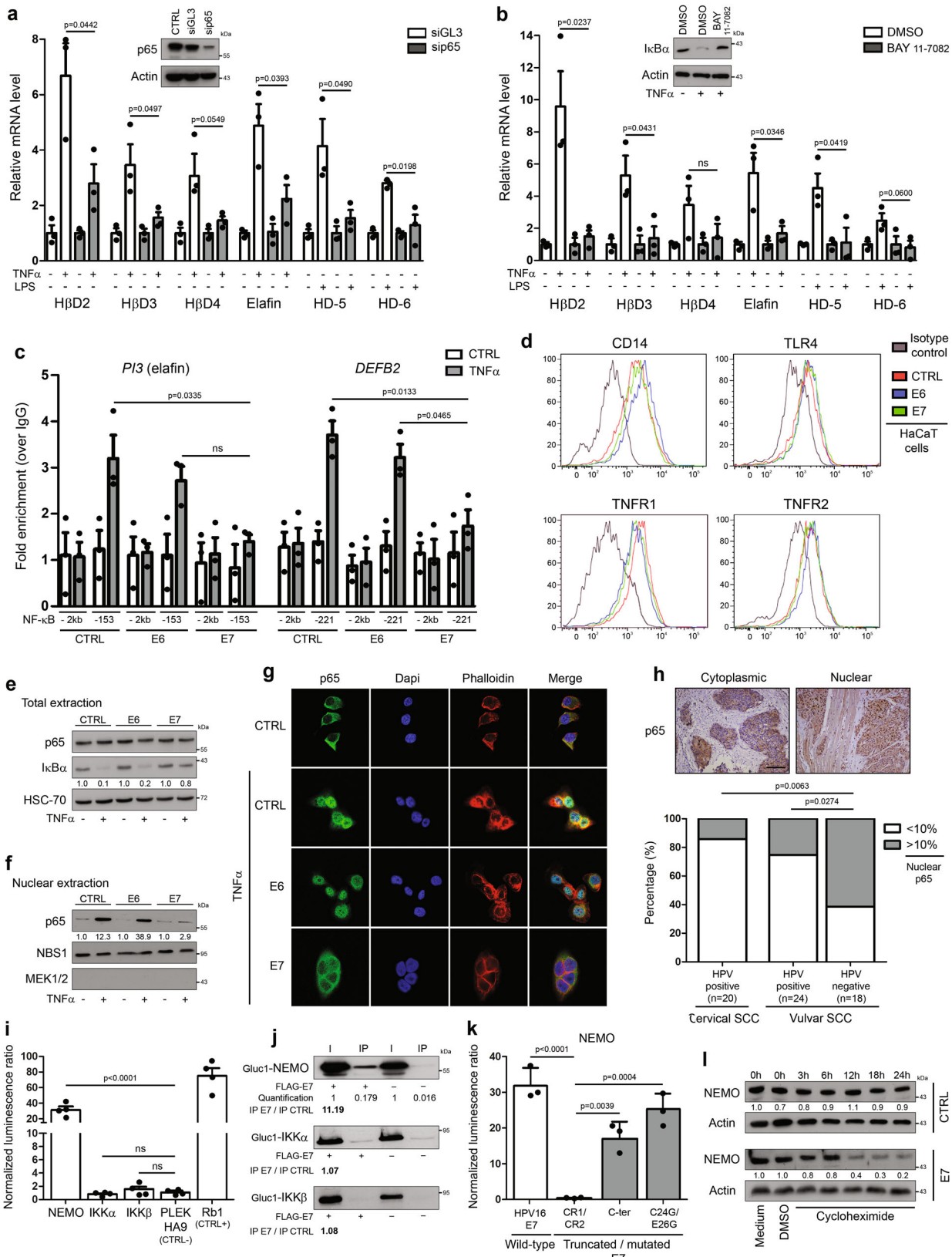

each individual viral oncoprotein on c-myc expression, its mRNA level was investigated in HPV16 E6 or E7-transduced keratinocytes. Compared to control cells, a significant increase in c-myc transcripts was only detected in HPV16 E7-positive cells (Fig. 4c) and the requirement of β-catenin in this up-regulation was highlighted (Fig. 4d). Confirming these data, an increased nuclear c-myc

immunoreactivity was clearly observed in HPV18-positive organo-typic raft cultures, E7-transduced keratinocytes as well as in HPV-positive (pre)neoplastic lesions compared to respective controls (Fig. 4e, f, i). Exclusively detected at the cell membrane in uninfected cells/tissues, an intense cytoplasmic/nuclear β-catenin immunor-eactivity was reported in case of HPV infection/E7 transduction

**Fig. 3 E7 viral oncoprotein reduces drastically TNFα/LPS-induced innate peptide expression through inducing NEMO degradation and impairing subsequent p65 nuclear translocation.** TNFα/LPS-induced defensin/elafin expression was analyzed by RT-qPCR in non-infected cells transfected with siRNA targeting p65 **a** or treated with 5 μM BAY 11-7082 **b**. Cells transfected with control siRNA (siGL3) or treated with DMSO were used as negative controls. Each experiment was normalized to the amount of HPRT mRNA from the same sample. Results represent the means ± SEM of three independent experiments. **c** Occupancy of NF-κB binding sites on *PI3* (-153 bp) and *DEFB2* (-221 bp) promoters was evaluated by ChIP in control, HPV16 E6-positive and E7-positive cells. Primers targeting a region with no putative NF-κB sites (-2 kb) in both promoters served as negative controls. Results represent the means ± SEM of three independent experiments. **d** Analysis of cell surface expression of TNFα (TNFR1-2) and LPS (CD14 and TLR4) receptors by control, HPV16 E6 and E7-transduced cells using flow cytometry. Keratinocytes stably transduced or not with HPV16 E6 or E7 were first treated with TNFα and both the degradation of IκBα **e** and the nuclear translocation of p65 was then analyzed by Western blot **f** or immunofluorescence **g**. The absence of colocalization between p65 and DAPI in E7-transduced cells (cytoplasmic sequestration of p65) should be noticed. **h** Anti-p65 immunostaining in both HPV-positive (n = 44) and negative SCC (n = 18). Low densities of epithelial cells displaying a nuclear p65 immunoreactivity were observed in virus-related tumors. Of note, tissue specimens of vulvar cancer were included due to the impossibility of analyzing HPV-negative SCC arising from the uterine cervix. **i** Binding of protein members of the IKK kinase complex with HPV16 E7 oncoprotein assessed by *Gaussia princeps* luciferase complementation assay (GPCA). Retinoblastoma-associated protein (Rb1) and PLEKHA9 were used as positive and negative control, respectively. Results represent the means ± SEM of four independent experiments. **j** Co-IP of NEMO, IKKα and IKKβ with HPV16 E7 oncoprotein. **k** GPCA analyzing the binding of truncated/mutated forms of HPV16 E7 with NEMO. Results represent the means ± SEM of three independent experiments. **l** NEMO stability in both control and HPV16 E7-transduced cells following treatment with 100 μM cycloheximide. The scale bar represents 100 μm. *P* values were determined using two-sided unpaired *t*-tests **a**, **b**, one-way ANOVA followed by Dunnett's multiple comparison post-hoc test **c**, **i**, Fisher's exact test **h** and one-way ANOVA followed by Bonferroni post-hoc test **k**. ns: not significant (*p* > 0.1). The Western blot and co-IP analyses were independently performed three times and twice, respectively. One representative experiment is shown. Source data are provided as a Source Data file.

(Fig. 4e, f, h). As shown in Fig. 4f, E6 viral oncoprotein modified neither β-catenin cellular localization nor c-myc protein level. As β-catenin is well-known to interact with the cytoplasmic tail of E-cadherin in normal conditions, we then analyzed the expression of this latter protein in our in vitro/in situ models. Reduced anti-E-cadherin immunostainings were observed in the large majority (>75%) of HSIL and cervical cancers (Fig. 4g) as well as in HPV-positive raft cultures (Fig. 4e) and E7-transduced cells (Fig. 4f). The patchy E-cadherin immunoreactivity observed in both E7-positive cells (Fig. 4f) and most tissue specimens guided our choice to analyze the methylation status of E-cadherin gene (*CDH1*). As expected, *CDH1* promoter displayed more methylated CpG islands in E7-transduced cells compared to both control and E6-positive cells (Fig. 4j). The reactivation of silenced *CDH1* gene following 5-aza-deoxycytidine treatment further supported the implication of DNA methylation in E-cadherin down-regulation observed in case of HPV infection (Fig. 4k).

As extensively discussed by Jeanes et al.[38], the loss of E-cadherin alone (without concomitant alteration of the β-catenin degradation complex) does not increase β-catenin stabilization/signaling. Given the intense cytoplasmic/nuclear β-catenin immunoreactivity observed in vitro/in situ (Fig. 4e, f, h), the potential interactions between HPV16 E7 oncoprotein and members of the β-catenin degradation pathway were investigated by GPCA. Importantly, high (positive) GPCA signals were obtained with the alpha and gamma (γ1-3) isoforms of Casein Kinase 1 (CK1), the ubiquitin E3 ligase β-TrCP and GSK3β (Fig. 4l). Confirmed by co-IP in both directions, both E7-CK1 and E7-β-TrCP interactions required the C-terminal region of the viral oncoprotein (Fig. 4m, n, Supplementary Figs. 5 and 7). Regarding the interaction between HPV E7 and GSK3β, no clear confirmation by co-IP was obtained arguing that it is likely a false positive of the GPCA screening method (Supplementary Fig. 8). The interactions with both alpha and gamma isoforms of CK1 as well as with β-TrCP were also reported with E7 oncoprotein from other HPV genotypes (high-risk alpha: HPV18, 33 and 39; beta: HPV8, 38 and 49) (Supplementary Fig. 9). Finally, compared to control cells, a marked degradation of CK1 (associated with a stabilization of β-catenin) was observed in HPV16 E7-transduced cells following cycloheximide treatment (Fig. 4o). Further confirming all these latter results, the CK1-dependent phosphorylation of β-catenin (ser45) (first step of the β-catenin ubiquitination-dependent proteolysis) was drastically inhibited by HPV16 E7 (Fig. 4p).

**Innate peptides predominantly and constitutively secreted by the cervical/vaginal mucosa promote Lactobacillus survival.** In order to determine the antimicrobial activity of each host innate peptide down-regulated in case of HPV infection, the predominant *Lactobacillus* species constituting the vaginal microbiome (*L. crispatus*, *L. jensenii* and *L. iners*) as well as *Gardnerella vaginalis* were incubated with several concentrations of each peptide for 6 h. The bacterial suspensions were then plated on Columbia Blood agar plates for 24 or 48 h and the percentage of surviving colonies was finally quantified by computerized counts and verified by manual counting (Fig. 5a). Similar to data reported with commensal gut bacteria[39], the influence of each individual peptide was specific and varied according to each bacterial strain (Fig. 5b). Despite these observed variations, three main observations were made: (1) defensins (most notably HβD2-4 and HD-5) inhibit the growth of all tested bacteria, (2) all peptides (except HβD1) exhibit an antimicrobial activity on *Gardnerella vaginalis* and (3) unexpectedly, the constitutive and most expressed peptides by the cervical/vaginal squamous mucosa show a positive effect on *Lactobacillus* survival. Given that the redox potential may vary within the vaginal lumen and that this parameter has been previously shown to modify the activity of some defensins (especially HβD1) against a few bacterial species[39,40], the defense peptides were first incubated with DTT (2 mM) before reproducing our systematic analysis. The chemical peptide synthesis, the purity as well as the reduction of disulfide bridges following DTT addition were validated by MALDI mass spectrometry (Supplementary Fig. 10). As shown in Fig. 5c, the antimicrobial activity of β-defensins was globally unchanged in a reducing environment (with the exception of a slight increase in the antibacterial activity of HβD3 against *L. crispatus*) and the significant protective effect displayed by both elafin and S100A7 on *Lactobacilli* was still observed. Changing buffer (sodium to potassium phosphate buffer) or pH from 7.4 to 5.8 did not modify the positive effect of these latter peptides on *Lactobacillus* survival either (Fig. 5d–e).

**Host defense peptides promoting Lactobacillus survival are cleaved, internalized and used as amino acid source.** In order to determine whether the beneficial effect of innate peptides constitutively expressed by the vaginal/cervical mucosa appears

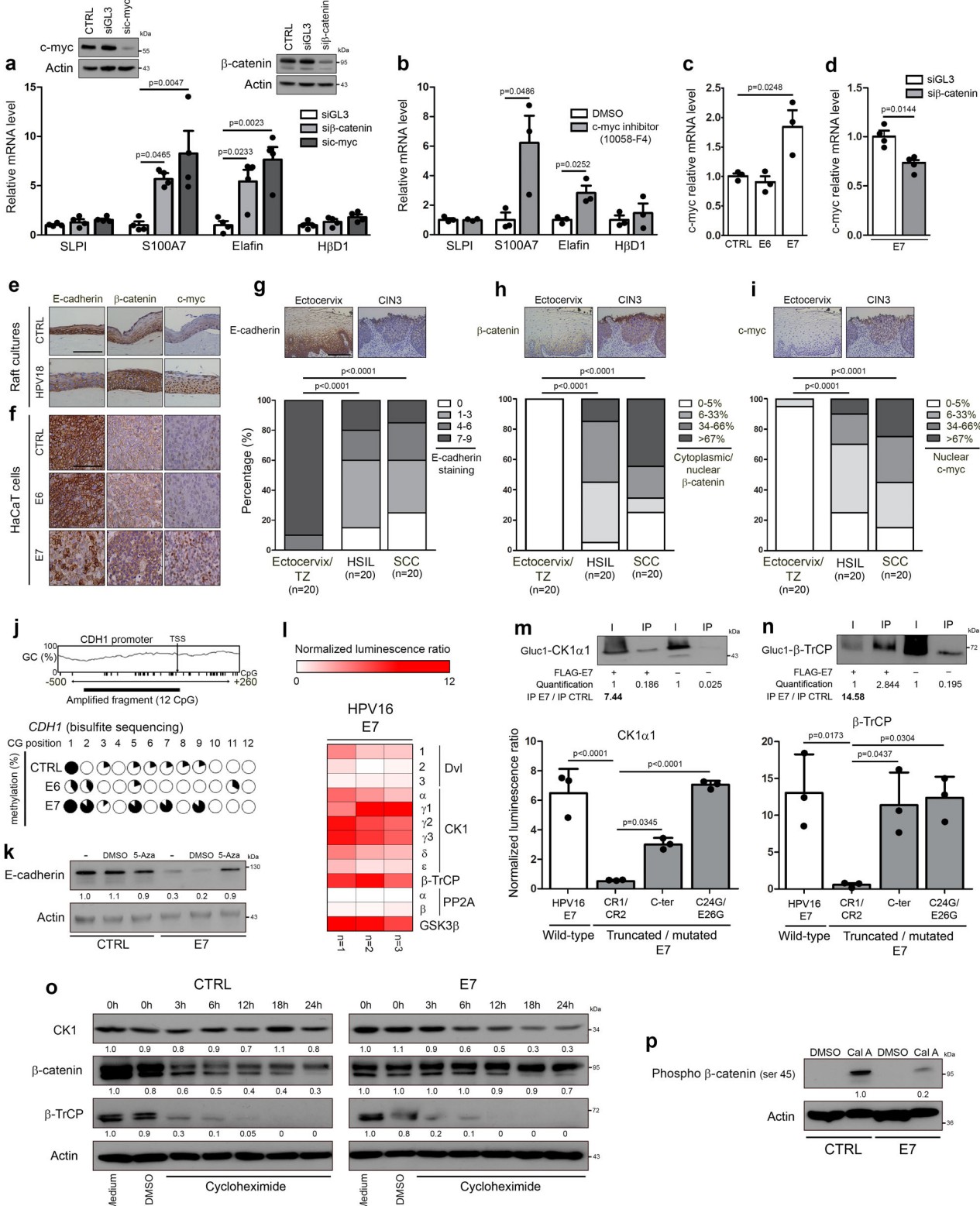

directly or needs some latency, elafin and S100A7 were added to lactic acid bacteria and several time points were investigated. Importantly, for all *Lactobacillus* species, a significant increased percentage of surviving colonies was only detected after 3 or 6 h of incubation (Fig. 6a). As commonly practiced for both eliminating variables and focusing on the compounds of interest, no nutrient (serum) was added during the incubation (starvation assay), explaining why the number of surviving colonies was

decreasing over the time. By mass spectrometry, we then evaluated the ability of these dominant cervico-vaginal lactic acid bacteria to convert proteins/peptides present in their environment to smaller peptides and ultimately to free amino acids. Strikingly, almost all (>80%) of native elafin and S100A7 had disappeared after 6 h, indicating their proteolysis. (Fig. 6b and Supplementary Fig. 11). By flow cytometry, we also showed an internalization of fluorescent-labeled peptides (or more precisely

**Fig. 4 E7 viral oncoprotein inhibits constitutive innate peptide expression (elafin and S100A7) through promoting β-catenin stabilization/signaling and subsequent c-myc expression.** The reactivation of so called "constitutive" peptide (SLPI, S100A7, elafin and HβD1) expression was analyzed by RT-qPCR in HPV16 E6E7-positive cells transfected with siRNA targeting c-myc or β-catenin **a** or treated with 40 μM 10058-F4 **b**. Cells transfected with control siRNA (siGL3) or treated with DMSO were used as negative controls. Each experiment was normalized to the amount of HPRT mRNA from the same sample. Results represent the means ± SEM of three **b** or four **a** independent experiments. The level of c-myc mRNA was determined by RT-qPCR in control, HPV16 E6 and E7-transduced cells **c**. Results represent the means ± SEM of three independent experiments. The effect of knockdown of β-catenin on c-myc expression in HPV16 E7-positive cells was assessed by RT-qPCR **d**. Results represent the means ± SEM of four independent experiments. **e** Representative pictures of control and HPV18-positive organotypic raft culture sections stained for E-cadherin, β-catenin and c-myc. **f** Representative examples of anti-E-cadherin, β-catenin and c-myc immunoreactivities displayed by control, E6-positive and E7-positive keratinocytes. Anti-E-cadherin **g**, anti-β-catenin **h** and anti-c-myc immunostainings **i** in both HPV-negative specimens [ectocervix/transformation zone (TZ), $n = 20$] and HPV-positive (pre)neoplastic lesions (HSIL, $n = 20$; SCC, $n = 20$). Note the reduced expression of E-cadherin associated with cytoplasmic/nuclear β-catenin immunoreactivity observed in HPV-positive tissues. Regarding c-myc, its expression was almost exclusively detected in HPV-positive (pre)neoplastic lesions. **j** The methylation status of E-cadherin gene (*CDH1*) promoter in control, HPV16 E6 and E7-transduced cells was assessed by bisulfite genomic sequencing. The variation in GC content within *CDH1* promoter as well as the percentage of methylation of each individual CpG island contained in the PCR fragment analyzed by bisulfite sequencing are shown. **k** The reactivation of hypermethylated *CDH1* gene was determined by Western blot following 5-aza-deoxycytidine (5-Aza, 5 μM) treatment. **l** Binding of protein members of the β-catenin degradation complex with HPV16 E7 oncoprotein was assessed by *Gaussia princeps* luciferase complementation assay (GPCA). Heatmap representing the score (normalized luminescence ratio) of each analyzed protein for potential interaction with HPV E7 is shown. Three independent experiments were performed. The interactions between HPV16 E7 oncoprotein and CK1α1 **m** or β-TrCP **n** were validated by both Co-IP and GPCA using the truncated/mutated forms of E7. Results represent the means ± SEM of three independent experiments. **o** CK1, β-catenin and β-TrCP stability in both control and HPV16 E7-transduced cells following treatment with cycloheximide (100 μM). **p** CK1-dependent phosphorylation of β-catenin (ser45) in both control and E7-positive cells following treatment with calyculin A (50 nM). The scale bar represents 100 μm. *P* values were determined using one-way ANOVA followed by Dunnett's multiple comparison post-hoc test **a**, **c**, two-sided unpaired *t*-tests **b**, **d**, $\chi^2$ test **g**, **h**, **i** and one-way ANOVA followed by Bonferroni post-hoc test **m**, **n**. The Western blot and co-IP analyses were independently performed three times and twice, respectively. One representative experiment is shown. Source data are provided as a Source Data file.

their hydrolyzed products) within the cytoplasm of bacteria as soon as 1h30 following peptide additions (Fig. 6c). Using synthetized elafin fragments containing $^{13}C_6^{15}N_2$-labeled lysines, we analyzed whether these latter were found in the amino acid sequence of some bacterial proteins. In total, 20,100 peptides (*L. crispatus*: 8,221; *L. jensenii*: 10,855; *L. iners*: 1,024 peptides), corresponding to 2,088 proteins (*L. crispatus*: 816; *L. jensenii*: 1,016; *L. iners*: 256 proteins), were identified by mass spectrometry. Importantly, 52 bacterial peptides that had incorporated $^{13}C_6^{15}N_2$-labeled lysines (from exogenous elafin) were detected (Supplementary Data 1). As shown in Fig. 6e, STRING analysis of positive proteins for $^{13}C_6^{15}N_2$-labeled lysines highlighted enrichments among several pathways (most notably ribonucleoprotein and protein biosynthesis as well as energy metabolism). Although the statistical significance was not reached in all conditions, an increase of ATP production was finally reported in bacteria cultured with elafin or S100A7 (Fig. 6d), indicating even more a metabolization of innate peptides into amino acids which are essential for bacterial survival/growth.

**The development of cervical/vaginal HPV-associated neoplasia in the K14-HPV16 transgenic mouse model is associated with both reduced innate peptide expression and vaginal dysbiosis.** In order to validate in vivo the physiological relevance of our in vitro data (and to further confirm our retrospective clinical results in patients who underwent routine cytology/HPV testing), squamous carcinogenesis within the lower genital tract of K14-HPV16 transgenic mice was first induced by chronic estrogen treatment. A similar procedure (use of a subcutaneous 17β-estradiol implant) was done with control (FVB/n) mice. After 12 weeks of treatment, 11 out of 12 (91.7%) K14-HPV16 mice displayed histological evidence of intraepithelial neoplasia (5 LSIL and 6 HSIL), in keeping with both the increased proliferative index and the thickening of the squamous epithelium lining the external part of the cervix, vagina and vulva (Fig. 7a). A cervical/vaginal hyperplasia was detected in the latest transgenic mouse (1/12, 8.3%). No sign of preneoplastic lesion was observed in the control group. The squamous mucosa was microdissected and the

mRNA level of mouse orthologs of human defensin(-like) peptides was determined. Of note, there is no mouse ortholog for the human *PI3/elafin* and *DEFB4* genes. As shown in Fig. 7b and, similarly to our results collected with human tissue specimens (Fig. 2b, c), all (excepted mβD4) antimicrobial peptides were significantly down-regulated in HPV16-expressing epithelia compared to their normal (uninfected) counterparts. Mouse βD12 was undetectable in all HPV-positive as well as in 7 out of 10 control samples. In parallel to these transcriptional analyses, cervico-vaginal lavage samples were collected in both K14-HPV16 and control mice at week 0 and after 12 weeks of estrogen treatment. In order to avoid bias related to reproductive cycling status of mice, vaginal microbiota was always taken during estrous. Following DNA extraction, V5-V6 hypervariable regions of the 16S rRNA gene were amplified by standard PCR and sequencing was performed (Illumina MiSeq). From 5,266,888 raw sequencing reads, 4,782,589 reads were retained after data cleaning and chimera removal, with a mean read length of 281 nucleotides. In total, 4,619 operational taxonomic units (OTUs) were obtained when a clustering distance of 0.03 was used. The ecological indices (intrinsic diversity, richness and evenness) of microbial populations detected in the lavage samples were first assessed at both the genus and species levels (Fig. 7c and Supplementary Fig. 12) and no significant difference was detected between the four defined groups [FVB / n (week 0 *versus* week 12) and K14-HPV16 (week 0 *versus* week 12)]. It is interesting to notice that the composition of murine vaginal microbiota was not altered by long-term estrogen treatment (Supplementary Figure 13). Strikingly, when data reported in K14-HPV16 mice were separated depending on the grade of (pre)neoplastic lesions observed within the lower genital tract, a significant increase in bacterial richness was detected in mice displaying a high-grade precancer (HSIL). Concurrently, both the bacterial α-diversity and evenness were reduced (Fig. 7d and Supplementary Fig. 12), meaning that, despite a higher global number of genera/species, the distribution of these latter (relative abundance) in vaginal lumen is less uniform in mice with histologically-proven HSIL compared to their control (week 0) counterparts or those diagnosed with hyperplasia/LSIL. As revealed by AMOVA-based

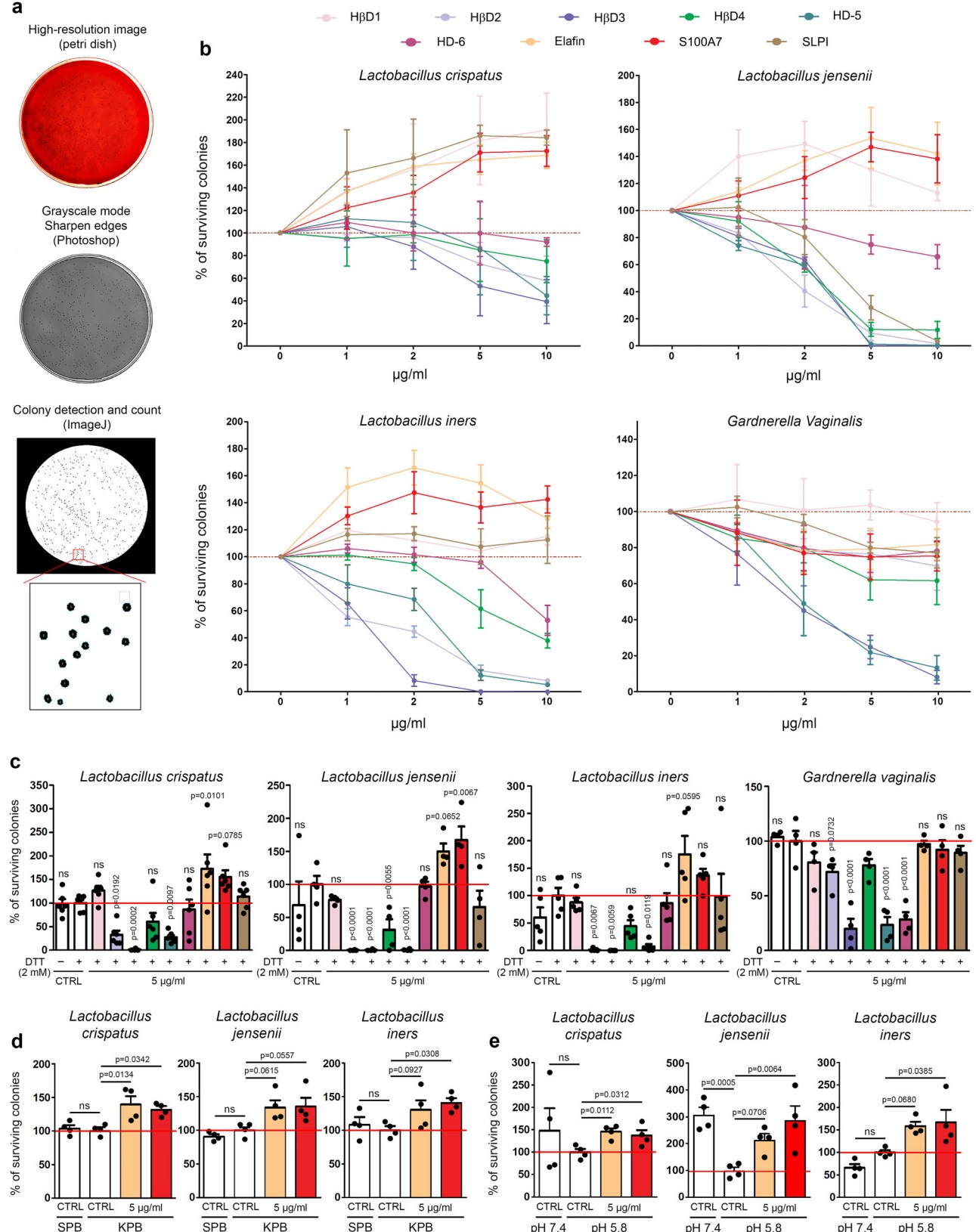

group clustering and clearly illustrated by the β-diversity plot (Fig. 7e), the microbial profiles detected in HSIL-positive K14-HPV16 mice were significantly distinct to those from hyperplasia/LSIL-positive ($p = 0.002$) or control mice ($p = 0.0005$). In addition, the homogeneity in the cervico-vaginal microflora tended to be different in mice diagnosed with a precancerous lesion

compared to the control group, as indicated by HOMOVA testing (HSIL *versus* control, $p = 0.09$; hyperplasia/LSIL *versus* control, $p = 0.08$). The distribution of the twelve main bacterial orders and their relative abundance in the four groups of mice are shown in Fig. 7f. Accounting for over 90% of the total microbial community in the vast majority of samples (44/48, 91.7%), the

**Fig. 5 Innate peptides abundantly and constitutively secreted by the squamous mucosa lining the lower gynecologic tract exhibit a positive effect on *Lactobacillus* survival. a** Schematic illustration of the different steps involved in bacterial colony detection and count. **b** Systematic analysis of both defensin and "defensin-like" peptide activity on predominant bacterial species detected in normal (*L. crispatus, L. jensenii* and *L. iners*) and pathological (BV) (*G. vaginalis*) conditions. Results represent the means ± SEM of three (*L. crispatus, L. jensenii* and *L. iners*) or five (*G. vaginalis*) independent experiments. **c** Effect of reducing environment on antimicrobial/protective activity of tested innate peptides. Results represent the means ± SEM of four (*L. jensenii* and *G. vaginalis*), five (*L. iners*) or six (*L. crispatus*) independent experiments. The positive effect on *Lactobacillus* survival displayed by the most abundant innate peptides secreted by cervical/vaginal mucosa (elafin and S100A7) was also assessed in both potassium phosphate buffer (KPB) (**d**) and acidic (pH 5.8) conditions (**e**). For each experiment, peptides were incubated with bacteria for 6 h. Results represent the means ± SEM of four independent experiments. *P* values were determined using one-way ANOVA followed by Dunnett's multiple comparison post-hoc test (**c**, **e**). ns: not significant ($p > 0.1$). Source data are provided as a Source Data file.

bacteria belonging to the order of Lactobacillales or Pasteurellales were clearly predominant in the cervix/vagina of our mouse models, regardless of the HPV status. Whereas no statistical difference was observed in the FVB/n (HPV-negative) group, interestingly, (pre)cancer development in the K14-HPV16 transgenic mice was associated with a significant increase in the abundance of genera (especially *Rodentibacter_ge*) from the *Pasteurellaceae* family (Fig. 7g). When paired lavage samples were compared (week 0 *versus* week 12), a marked augmentation was detected in all animals (6/6, 100%) displaying a HSIL. In the opposite, the relative abundance of genera (especially *Lactobacillus_ge* and *Streptococcus_ge*) from the Lactobacillales order has been shown to drastically decrease in HPV16-expressing mice with a high-grade/extended (pre)neoplastic lesion (Fig. 7h).

## Discussion

The development of HPV-related (pre)neoplastic lesions within the lower genital tract and BV are two common conditions among women of reproductive age. With a risk increasing with the multiplicity of sexual partners, both disorders have incidence rates peaking among women aged between 25 and 50 and then dropping gradually. In the last 20 years, many studies have been focused on both high-risk HPV genotypes and BV and, most of the time, an epidemiological link, characterized by higher rates of HPV infection (and persistence) among patients with BV, was reported[25,26,28]. In the opposite, and although suggested in a few recent publications[41–43], it remains unclear whether or not HPV can exert a critical influence on vaginal microbiome. To address this essential issue, a large retrospective follow-up study was conducted. Although a few sources of potential imprecision exist (e.g., the lack of data about the socioeconomic status of patients) that could slightly impact the collected results, remarkably, a bi-directional association was clearly reported. Indeed, each individual pathological disorder was more frequently diagnosed when the other one was preceding in time. Similarly to previous longitudinal measurements[25], an overall odds ratio of 1.83 (95% CI: 1.41–2.37, $p < 0.0001$) was obtained for HPV infection among BV-positive women. As for BV development, a 2.35-fold increased risk (95% CI: 1.48–3.72, $p < 0.0007$) was observed in case of prior HPV infection. Of interest, whatever their HPV status, BV women were typically treated with antibiotics (metronidazole or clindamycin) and the follow-up of HPV-positive patients did not differ by BV status. Taken together, this supports that these findings cannot be explained by differences in management between defined groups. Actually, these results are consistent with recent 16S rRNA sequencing studies that reported both higher vaginal microbial diversity and reduced relative abundance of *Lactobacillus* species in HPV-positive women compared to uninfected individuals[44–46]. Furthermore, the levels of both *L. crispatus* and *L. jensenii* have been shown to progressively decrease with preneoplastic disease severity, irrespective of the detected HPV genotypes[46]. Although the collected data are highly significant, an odds ratio of 2.35 with a cohort

including over 6,000 patients may seem quite modest. It is, however, important to mention that not all HPV infections can substantially alter the vaginal microbiome. Indeed, a substantial proportion of HPV infections (even those implicating high-risk genotypes) are asymptomatic and/or spontaneously regressive within a few months. It is reasonable to think that these latter only modestly influence the vaginal ecosystem and, consequently, reduce the acquired odds ratio. In agreement, in our in vivo model, an imbalance in the vaginal flora was only detected in HPV16-expressing mice that had developed high-grade intrae-pithelial neoplasia. In addition to an increased risk of occurrence, strikingly, a significant effect on disease persistence was also clearly detected. Of note, the consensus guidelines recommend repeated cytology/HPV testing every 6 months for HPV-positive women while intervals of 3 years are usually acceptable in case of HPV negativity[47]. This difference likely explains why the positive impact of HPV infection on BV persistence was only detected after a 2.5-year period and then increased over the time (bias of follow-up frequency between patient groups). Therefore, besides the higher risk of HPV infection, viral persistence and related carcinogenesis following BV development, persistent/sympto-matic HPV infection may also induce changes within the vaginal lumen that would ultimately disrupt the microbial balance. Despite bacterial diversity differences between humans and mice (more particularly, in normal condition, human vaginal micro-biome is largely dominated by *Lactobacillus* species whereas, in rodents, higher abundances of *Proteobacteria* and *Actinobacteria* are encountered), the dysbiosis of vaginal microbiome detected following HPV-related carcinogenesis in the K14-HPV16 trans-genic mouse model further supports the retrospective clinical data and, overall, this novel paradigm.

Using a large number of tissue specimens as well as several in vitro 2D/3D models, we here showed that HPV leads to a drastic reduction of innate peptide expression, mostly resulting of the interaction of E7 oncoprotein with several key members of both NF-κB and Wnt/β-catenin signaling cascades. Indeed, with the exception of HβD3 which has been recently shown to have a dual regulation by NF-κB and p53[48], all constitutive or inducible defense peptides normally secreted by the squamous epithelium lining both the vagina and the outer part of the uterine cervix were down-regulated in case of HPV positivity. In contrast to all the other peptides, we reported that SLPI and HβD1 were affected neither by NF-κB nor by the Wnt/β-catenin pathway. Targets of interferon regulatory factors (IRF) (IRF-1 and IRF-7)[49], their impaired expression detected in HPV-positive cells is very likely related to the well-characterized (direct or via TLR9 suppression) abrogation of these latter proteins by HPV oncoproteins[50,51]. Overall, we can reasonably speculate that the virus impairs the production of epithelial-specific members of the defensin and "defensin-like" families as early as the first steps of infection/(pre) cancer development as part of a broad effort to suppress the inflammatory response. As recently highlighted[52,53], the host defense peptides display properties which extend well beyond

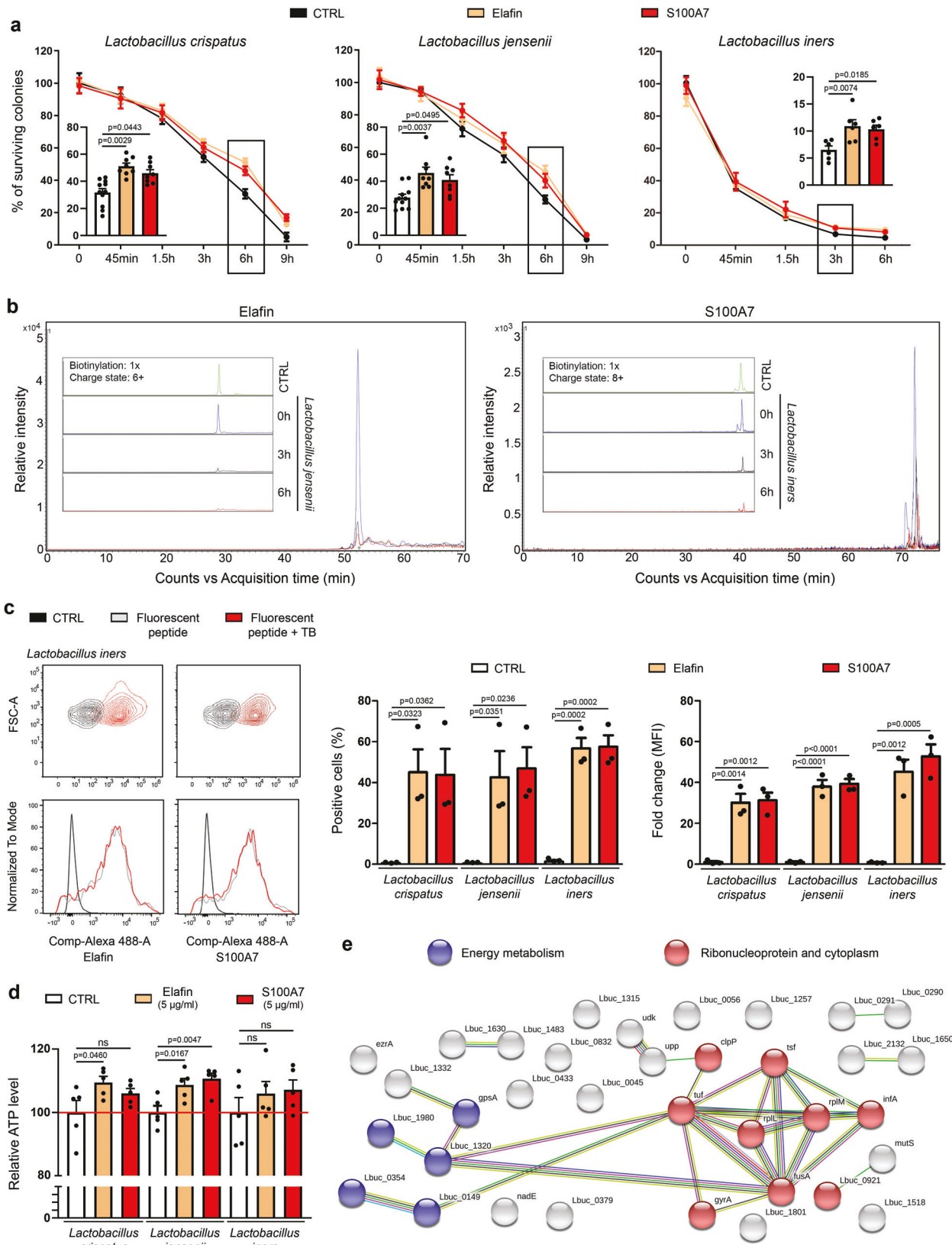

their antimicrobial activity. For example, defensin(-like) peptides (e.g., HβD2, HD-5, and S100A7) have a well-documented chemotactic capacity for lymphocytes, monocytes and dendritic cells and their reduced secretion would greatly help the virus to escape the host immunity[54–56]. Further promoting a tolerogenic environment, the lack of HβDs would also disrupt the activation and

maturation of antigen-presenting cells[57]. Moreover, some defensins (most notably HD-5) have been demonstrated to block HPV infection via various mechanisms[58,59]. Therefore, the low HD-5 expression detected in infected epithelial cells could be particularly beneficial for the virus during the productive replication stage. Last but not least, this global decrease in innate peptide

**Fig. 6 Host defense peptides are metabolized and used as amino acid source by *Lactobacillus* species. a** Elafin and S100A7 were incubated with *Lactobacillus* species for 45 min to 9 h in the absence of nutrients (starvation assay) and the percentage of surviving colonies was determined. Results (each time point) represent the means ± SEM of six (*L. iners*: CTRL, elafin and S100A7), eight (*L. crispatus/L. jensenii*: elafin and S100A7) or eleven (*L. crispatus/L. jensenii*: CTRL) independent experiments. **b** The hydrolysis of host defense peptides (elafin and S100A7) by the dominant *Lactobacillus* species constituting the vaginal microbiome was evaluated at different time points (0 h, 3 h and 6 h) by mass spectrometry. Note the disappearance of native proteins after 3 to 6 h. **c** The internalization of fluorescent-labeled peptides within the cytoplasm of bacteria was then assessed by flow cytometry. Both the percentage of positive cells and mean fluorescence intensity (MFI) are shown. Results represent the means ± SEM of three independent experiments. **d** ATP production after 6 h of incubation between peptides and bacteria. Results represent the means ± SEM of five independent experiments. **e** STRING analysis of the bacterial proteins that have incorporated $^{13}C_6{}^{15}N_2$-labeled lysines (from exogenous elafin). All proteins identified by mass spectrometry in *L. crispatus*, *L. jensenii* or *L. iners* were pooled. *P* values were determined using two-way ANOVA followed by Bonferroni post-hoc test **a** and one-way ANOVA followed by Dunnett's multiple comparison post-hoc test **c**, **d**. ns: not significant ($p > 0.1$). Source data are provided as a Source Data file.

expression was not only detected in HPV-infected cells but also in the morphologically normal squamous epithelium adjacent to HPV-positive (pre)neoplastic lesions. The concomitant down-regulation of pro-inflammatory cytokines (e.g., TNFα, IL-1β) resulting from both NF-κB and Wnt/β-catenin signaling impairment is very likely to explain the extension of this immunosuppressive effect in the lesional (micro)environment (absence of paracrine stimulation).

We originally postulated that the altered antimicrobial peptide secretion profile of the host mucosa following HPV infection could differently impact *Lactobacillus* and anaerobic bacteria species constituting the vaginal microbiome. However, we were far to expect that defense peptides such as elafin and S100A7, which are very highly expressed by vaginal/cervical keratinocytes, do not display any antimicrobial activities on lactic acid bacteria but, rather, can be cleaved, internalized and used by these latter as amino acid source sustaining their survival. Changing the redox state of peptides, environment (buffer) or pH did not impact this positive effect on *Lactobacilli*. The citation "what doesn't kill you, makes you stronger" seems particularly adapted in the present context. Essential to compensate their inability to synthesize amino acids (auxotrophy), the interest for the hydrolytic capacities of *Lactobacilli* has largely risen during the last few years. Originally characterized on *L. casei*, *L. helveticus* and *L. delbrueckii* which are involved in the manufacture of cheeses, yogurts and fermented milks, the proteolytic system of *Lactobacillus* species not only supplies amino acids to the bacteria, but also produces bioactive peptides with health-protective effects[60]. As demonstrated by mass spectrometry, the dominant cervico-vaginal lactic acid bacteria have also developed the capability to hydrolyze proteins into free amino acids which are required for their growth. Interestingly, several studies reported important variations in caseinolytic properties between different *L. helveticus* strains used in food science and nutrition[61]. Whether disparities in protein hydrolysis patterns could explain the differential protective abilities displayed by cervico-vaginal *Lactobacillus* species (e.g., *L. crispatus* versus *L. iners*) or whether some cleaved products have beneficial properties for the host is still unknown but merits further investigations.

With the aim of persisting, replicating and escaping the immunity, oncogenic viruses developed multiple strategies for subverting host signaling cascades. Among these latter, both NF-κB and Wnt/β-catenin signaling pathways play critical roles in proliferation/apoptosis, inflammation and differentiation and are, therefore, considered as preferential targets for viral oncoproteins. To the best of our knowledge, this study provides the first evidence of interactions between HPV E7 oncoprotein and NEMO, CK1 and β-TrCP. Collectively, the results obtained by GPCA support that these three host proteins very likely interact directly to the C-terminal region of the viral oncoprotein. Interestingly, NEMO, CK1 and β-TrCP not only interacted with E7 from high-risk alpha HPVs (HPV16, 18, 33 and 39), but also with E7 from

beta HPV genotypes (HPV8, 38 and 49) which have recently emerged as "facilitators" in UV-related cutaneous SCC development[62]. Altogether, our findings indicate the importance of these newly discovered targets for viral persistence/carcinogenesis and add to other mechanisms (e.g., impairment of p65 acetylation[63]) used by HPV for hijacking both NF-κB and canonical Wnt pathways. Essential to direct IKKβ activity towards IκBα[64], it is interesting to note that small T antigen of Merkel cell polyomavirus has also been shown to target the NEMO adaptor protein to inhibit NF-κB-dependent inflammatory/antiviral responses[65]. In agreement with previous studies reporting alterations of host DNA methylome in HPV E7 and hepatitis B virus (HBV) X protein-positive cells[66,67], altered E-cadherin expression (related to *CDH1* promoter hypermethylation) was observed both in vitro and in situ. Regarding the β-catenin degradation complex, we demonstrated that E7 interacts with CK1, inducing its degradation. As a result, both reduced phosphorylation (ser45) and stabilization of β-catenin were observed with, ultimately, the up-regulation of its targeted genes (e.g., c-myc). Finally, the interaction between the ubiquitin E3 ligase β-TrCP and HPV E7 oncoprotein could not only increase β-catenin half-life but also participate in NF-κB inhibition due to the fact that this latter protein is involved in ubiquitination and degradation of both β-catenin and IκBα[68,69]. Interestingly, the sequence of HPV E7 does not contain a putative $DSG(X)_{2+n}S$ motif (a common characteristic of protein targets for β-TrCP)[70], supporting that the viral oncoprotein is actually not a substrate for β-TrCP but, in the opposite, E7 binds to this host protein, impeding its activity.

Taken together, our findings indicate that HPV inhibits basal and pro-inflammatory-induced host defense peptide expression through subverting NF-κB and Wnt/β-catenin signaling cascades. As a consequence/side effect of HPV immune evasion, the amino acid source sustaining the survival of *Lactobacillus* species is greatly reduced, promoting an imbalance in the vaginal flora (Fig. 8). The oxidative stress resulting from BV establishment/persistence would then promote the progression of HPV-related (pre)neoplastic lesions supporting that the association between HPV and BV is complex and bi-directional.

## Methods
All experiments performed in this study adhere to all relevant ethical regulations.

**Retrospective cohort analysis**. All women who underwent at least 2 routine cervical Pap smear during the period 2010–2018 at the University Hospital of Liege (Belgium) or at its associated regional hospitals [Citadelle Regional Hospital (Liege, Belgium), ND Bruyeres (Chenee, Belgium), Regional Hospital of Huy (Huy, Belgium), Bois de l'abbaye Hospital (Seraing, Belgium)] were eligible for the present retrospective research which aimed at analyzing the temporal relationship between BV and genital HPV infections. All samples were daily centralized at University Hospital of Liege where both HPV testing and microscopic examinations were performed. The exclusion criteria were as follows: patients younger than 21 years of age; absence of HPV testing in case of abnormal cytological findings (e.g., ASC-US, ASC-H); lack of data related to vaginal bacterial flora; immunosuppressive

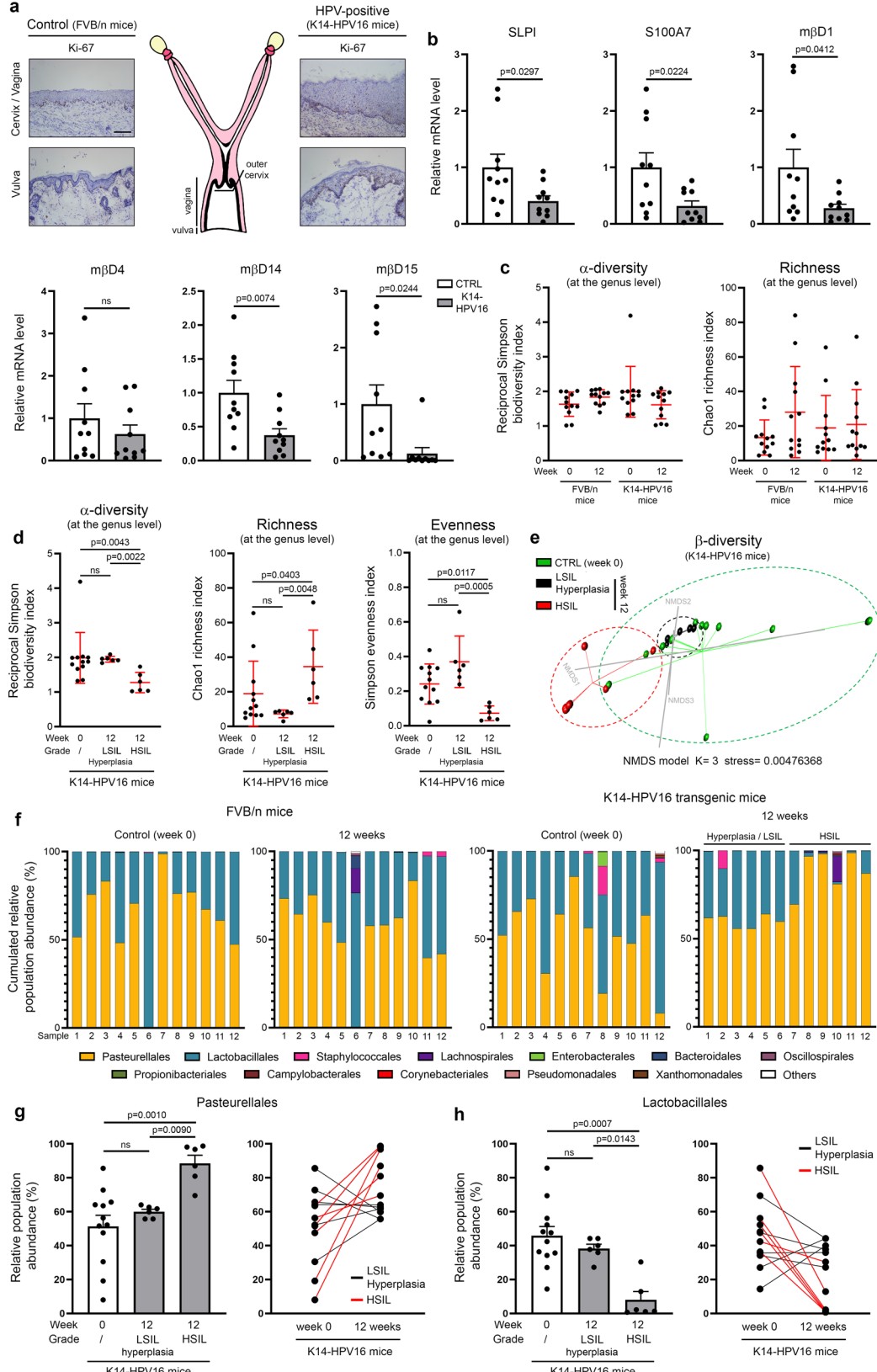

conditions. In total, six thousand one hundred seventeen women [mean age: 48 ± 13 years (range, 21–84)] were selected. Thirty-two patients (32/6,117, 0.52%) were excluded for missing data (incomplete viral-cytological information) during the follow-up period. Mean number of visits was 3 (range, 2–13). All Pap smear samples were analyzed by experienced cytopathologists according to the Bethesda system for reporting cervical cytology. HPV testing from residual liquid-based cytology specimens was performed using the Abbott RealTime High-Risk HPV

assay (Abbott, Wiesbaden, Germany) which enables simultaneous detection of all WHO/IARC-classified carcinogenic (high-risk) HPV types (HPV16, 18, 31, 33, 35, 39, 45, 51, 52, 56, 58, 59, 66, and 68). The vaginal bacterial flora was systematically assessed for all patients during the entire study period and BV was diagnosed using the Hay/Ison grading system, a simpler version of Nugent's score displaying the same high performance for BV diagnosis[71,72]. First described in 2002[71], this method evaluates a shift from a vaginal flora rich in *Lactobacilli* to a polymicrobial

**Fig. 7 Estrogen-induced cervical/vaginal carcinogenesis in transgenic mice expressing HPV16 induces an imbalance in the vaginal microflora. a** Schematic representation of the mouse reproductive tract. The morphology of the squamous epithelium lining the cervix/vagina and vulva in K14-HPV16 and control (FVB/n) mice is shown. Both the increased percentage of proliferative (Ki-67-positive) cells and the thickening of the epithelium in case of HPV16 oncogene expression should be noticed. **b** mRNA level of SLPI, S100A7, mouse orthologs of HβD1-3 (mβD1, mβD4, mβD14) and HD-5/6 (mβD12, mβD15) was measured by RT-qPCR. Microdissected frozen squamous epithelia from FVB/n and K14-HPV16 mice were analyzed. Each experiment was normalized to the amount of both HPRT and GAPDH mRNAs from the same sample. Results represent the means ± SEM of ten independent experiments. **c** Bacterial intrinsic diversity (reciprocal Simpson biodiversity index) and richness (deduced from Chao1 index). The reported values (at the genus level) for each individual mouse in the four defined groups [FVB/n (week 0 *versus* week 12) and K14-HPV16 (week 0 *versus* week 12)] are shown (n = 12 per group). The means ± SD are represented. **d** Bacterial α-diversity, genus richness and evenness (Simpson index) for K14-HPV16 mice. Data were separated depending on 17β-estradiol treatment duration [week 0 (n = 12) *versus* week 12 (n = 12)] and preneoplastic lesion grade [hyperplasia/LSIL (n = 6) *versus* HSIL (n = 6)]. The means ± SD are represented. **e** β-diversity of the vaginal microbial profile in K14-HPV16 mice was visualized using a Bray-Curtis dissimilarity matrix-based non-parametric dimensional scaling (NMDS) model (three dimensions). **f** Stacked bar charts depicting the relative abundance of the twelve main bacterial orders detected in control (FVB/n) and K14-HPV16 mice by 16 S V5-V6 amplicon sequencing. Relative abundance of bacteria belonging to the order of Pasteurellales **g** and Lactobacillales **h** in K14-HPV16 mice depending on 17β-estradiol treatment duration [week 0 (n = 12) *versus* week 12 (n = 12)] and HPV-related lesion grade [hyperplasia/LSIL (n = 6) *versus* HSIL (n = 6)]. The means ± SEM are represented. The comparison of relative abundance of bacteria in paired (week 0 *versus* week 12) lavage samples is also shown. The scale bar represents 100 μm. *P* values were determined using two-sided unpaired t-tests **b** and non-parametric Kruskal-Wallis test corrected with a two-stage linear step-up procedure of Benjamini, Krieger and Yekutieli **c**, **d**, **g**, **h**. ns: not significant (*p* > 0.1). Source data are provided as a Source Data file.

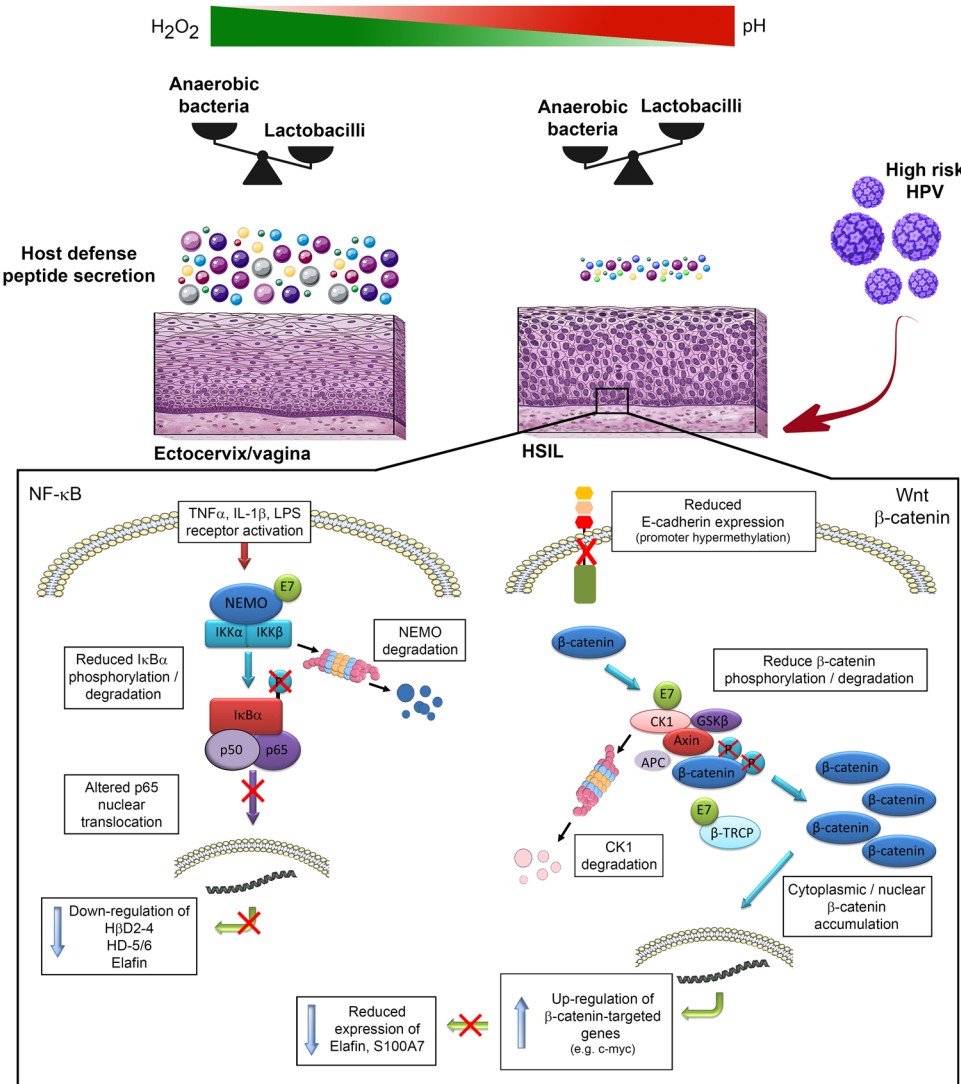

**Fig. 8 Schematic representation of the proposed model.** Left In HPV-positive cells, E7 oncoprotein impairs pro-inflammatory-induced innate peptide expression through inducing NEMO degradation and subsequent p65 cytoplasmic sequestration. Right In parallel, E7 reduces E-cadherin expression and alters β-catenin degradation complex by interacting with both CK1 and β-TrCP. As a result, defense peptides (e.g., elafin and S100A7) which are used by *Lactobacillus* species as amino acid source are greatly down-regulated by the β-catenin target gene c-myc. An imbalance in the vaginal flora is, therefore, achieved as a consequence of HPV persistence and subsequent immune evasion.

(anaerobic) microbiome and divides smears into three categories: Grade I (normal, *lactobacillus spp*. morphotype only), Grade II (intermediate, reduced *lactobacillus spp*. morphotype and equal amount of mixed bacterial morphotypes) and Grade III (BV, mixed bacterial morphotypes with few or absent *lactobacillus spp*. morphotype). Considering the previous studies reporting that most (>90%) smears assessed as Grade II are actually classified as normal when Nugent's or Amsel's criteria are used as well as the very high specificity (>95%) of clue cells for BV diagnosis[71–74], Grade II patients without clue cells were, therefore, routinely classified as normal whereas their counterparts exhibiting clue cells were considered as BV-positive (similarly to Grade III patients). Overall, the STROBE (STrengthening the Reporting of OBservational studies in Epidemiology) guidelines has been followed as accurately as possible. The data collection from personal health records was approved by the Ethics Committee of the University Hospital of Liege (#2021-117).

**Human tissue samples, classification and laser capture microdissection**. A total of 44 frozen and 127 paraffin-embedded cervical/vulvar specimens were retrieved from the Tissue Biobank of the University Hospital Center of Liege (Belgium). All patients underwent a surgical procedure between 2011 and 2015. An informed consent was obtained from all participants and the protocol was approved by the local ethics committee. These tissue samples included 31 normal squamous tissues (ectocervix/TZ), 65 dysplastic lesions (30 LSIL and 35 HSIL) and 75 primary invasive squamous cell carcinomas (SCC) (33 cervical and 42 vulvar cancers). The original diagnoses were re-examined by senior histopathologists and, to avoid misclassification, both the proliferative index (nuclear Ki67 staining) and p16INK4a expression of each specimen were determined by immunohistochemistry. In order to perform our transcriptional analyses on "pure" populations of epithelial cells (and to avoid any contamination with stromal structures), several serial sections (6 μm thick) of each frozen cervical specimen were microdissected using a Leica LMD7000 system (Leica, Wetzlar, Germany) (GIGA in vitro Imaging platform, University of Liege). To obtain sufficient RNA concentrations, microdissected squamous cell populations from different slides but from the same tissue sample were pooled.

**Transgenic mouse model and hormone treatment**. FVB/n mice expressing HPV16 under the control of the keratin 14 promoter (K14-HPV16 transgenic mice) were obtained from the National Cancer Institute Mouse Repository. Given that (pre)cancer development within the lower genital tract has been previously shown to be hormone-dependent in this model[75,76], 6 to 8-week-old female mice (n = 12 per condition) were subcutaneously implanted with a Belma's E2 implant (E2-M/90, Belma Technologies, Liege, Belgium), allowing reproducible long-term release of 17β-estradiol (1–2 μg/24 h) for 90 days[77]. A similar procedure was performed with control (FVB/n) mice. Mice were euthanized after 12 weeks of treatment and reproductive tracts were retrieved, embedded in Tissue-Tek® OCT compound (Sakura Finetek, Torrance, CA, USA) prior to cryosectioning and subsequent transcriptional and immunohistochemical evaluations. The protocol was approved by the Institutional Ethics Committee of the University of Liege (#2019-60).

**Sample collection/preparation, estrous determination, 16S-based metagenomics and sequence analysis**. Mice were randomly distributed in cages and housed under standard conditions [12 h light/12 h dark cycle, ad libitum access to food and water, constant temperature (19–21 °C) and humidity (40–50%)] in the GIGA-Mouse facility platform (University of Liege) during the entire course of the study. The gynecological tract of both control (FVB/n) and K14-HPV16 transgenic mice was colonized naturally (mother's impact at birth and the environment). Vaginal microbiome was not influenced by the investigators. In order to avoid environmental contaminations by soil bacteria, a sterilized recycled paper litter was preferred to the wood pellets. Before collecting vaginal microbiota (2 ×50 μl PBS lavage samples collected in sequence using gel-loading pipette tips and pooled), the perineal region of each individual mouse was carefully cleaned with an alcohol swab. The previous steps were performed with sterile materials in a laminar flow hood. Seven μl from each samples were spread on a glass slide, fixed/stained using Diff-Quick reagents (Medion Diagnostics, Düdingen, Switzerland) and the estrous cycle stage was determined using established criteria[78].

DNA was isolated from cervico-vaginal lavage samples using NucleoSpin Tissue kit (Macherey Nagel, Düren, Germany). Amplicon library preparation and sequencing were then performed in the GIGA-Genomics platform (University of Liege) according to the Illumina metagenomics workflow (https://support.illumina.com). Primers targeting the V5-V6 hypervariable regions of the 16S rRNA gene were used for standard PCR amplification. The primer sequences are listed in Supplementary Table 1. Of note, 7 random nucleotides were added to the 5′ region of forward primer to increase complexity and, therefore, improve sequence read quality. The Quant-iT PicoGreen dsDNA Kit (Invitrogen) and the KAPA Library Quantification Kit for Illumina platforms (KapaBiosystems, Wilmington, MA, USA) were used to normalize the DNA libraries and to determine the pooled library concentration, respectively. Pair-end sequencing (2 × 300 cycles) was done on the Illumina MiSeq platform. As previously described[79,80], both the alignment and taxonomical assignment (from phylum to species) of post-sequencing data were performed using the MOTHUR software package (v141.1), based on the

SILVA database (v1.38) of full-length 16S rDNA sequences. Potentially chimeric sequences were detected using the VSEARCH algorithm. For each sample, a subsampling dataset containing 10,000 representative cleaned reads was retained and used to generate OTUs (cut off: 0.03) as well as to evaluate several ecological indicators. The validity of the process was confirmed by a Good's coverage at the genus level of at least 99.8% in all samples and no statistical difference between the defined groups. The bacterial α-diversity and richness of the samples were deduced from the inverse Simpson index and Chao1 index, respectively. The Simpson-derived evenness was also determined. Measuring the diversity between different communities, the β-diversity was estimated using a Bray-Curtis dissimilarity matrix.

**Cell culture and stable clones**. Immortalized human keratinocytes (HaCaT cells) cells expressing HPV16 E6 and/or E7 or green fluorescent protein (control cells) were generated by using lentiviral vectors (GIGA-Viral Vectors, University of Liege). Briefly, HPV16 E6 and E7 coding sequences were synthetized and cloned into the pLV-EF1a-IRES Luciferase plasmid (VectorBuilder, Chicago, IL, USA). Lentiviral particles were obtained by co-transfecting each lentivector with pSPAX2 (Addgene, Watertown, MA, USA), a VSV-G-encoding vector, into Lenti-X 293 T cells (Clontech, Mountain View, CA, USA). As previously described[81,82], viral supernatants were then collected 48 h and 72 h post-transfection, filtrated and concentrated by ultracentrifugation. Before cell transduction, the qPCR Lentivirus Titration kit (ABM, Richmond, BC, Canada) was used for titrating the lentiviral vectors. Positively transduced cells were selected with 100 μg/ml hygromycin B or 10 μg/ml blasticidin (InvivoGen, San Diego, CA, USA) and were grown in Dulbecco's modified Eagle's medium containing 10% fetal calf serum and supplied with 1% sodium pyruvate and 1% nonessential amino acid (Gibco, Thermo Fisher Scientific, Waltham, MA, USA).

**Organotypic raft cultures**. Spontaneously immortalized human keratinocytes (NIKS cells) and foreskin fibroblasts (kindly provided by Prof. Paul F. Lambert, University of Wisconsin School of Medicine and Public Health, Madison, WI, USA) were routinely cultured in incomplete F medium on mitomycin C-treated J2 3T3 feeder cells and in Ham's F12 medium containing 10% fetal bovine serum, respectively. All supplements added in culture media were previously described[83,84]. Before being used in organotypic cultures, 2.5 × 10^6 NIKS cells were transfected with both recircularized HPV18 DNA (2 μg) and pcDNA6 plasmid (0.5 μg) using Effectene Transfection Reagent (Qiagen, Hilden, Germany). After blasticidin selection (7 μg/ml) for 6 days, 1.5 × 10^6 HPV18-positive cells were seeded onto a dermal equivalent composed of rat-tail type 1 collagen (Sigma Aldrich, Saint Louis, MI, USA) and 1 × 10^6 human foreskin fibroblasts. The daily procedure allowing the production of a pluristratified epithelium using this organotypic culturing approach was extensively detailed previously[83,84]. After 18 days, rafts were fixed in formalin and embedded in paraffin for both immunohistochemical and in situ hybridization analyses.

**Bacterial cultures and antimicrobial assays**. Bacterial strains *Lactobacillus crispatus* DSM 20356, *Lactobacillus iners* CIP 105923 and *Garnerella vaginalis* CCUG 43336 were kindly provided by the Department of Clinical Microbiology at University Hospital Center of Liege. *Lactobacillus jensenii* LMG 6414 was obtained from BCCM/LMG Bacteria Collection (University of Gent, Belgium). Log-phase bacteria were grown on Columbia agar with 5% sheep blood (Becton Dickinson, Franklin Lakes, NJ, USA) at 37 °C in a humidified CO₂ atmosphere (*Lactobacilli*) or in anaerobic jars (*G. vaginalis*). To analyze the antimicrobial effect of innate peptides expressed by vaginal/cervical mucosa, 5 × 10^4 colony-forming unit (CFU)/ml were mixed with different peptides (concentration range: 1–10 μg/ml) in 10 mM sodium phosphate buffer (SPB) (final volume 100 μl) and incubated for 45 min to 9 h at 37 °C. Where indicated, the peptides were incubated with 2 mM dithiothreitol (DTT) (Invitrogen, Carlsbad, CA, USA) for 30 min at 37 °C before adding the bacteria. In indicated experiments, incubation was carried out in potassium phosphate buffer (KPB) (pH: 7.4, 10 mM). The following peptides were tested: HβD1 (PeptaNova, Sandhausen, Germany), HβD2 (PeptaNova), HβD3 (PeptaNova), HβD4 (PeptaNova), HD-5 (PeptaNova), HD-6 (PeptaNova), elafin (PeptaNova), S100A7 (Abnova, Taipei City, Taiwan) and SLPI (R&D Systems, Minneapolis, MI, USA). Bacterial suspensions were then plated on Columbia Blood agar plates for 24–48 h to determine CFU. High-resolution images for each condition were taken. Colony detection and counts were finally performed using ImageJ software (NIH, Bethesda, MD, USA) and verified by manual counting.

**Matrix-assisted laser desorption/ionization (MALDI) mass spectrometry**. The different innate peptides were incubated with or without DTT (2 mM, pH 7.4) for 30 min at 37 °C and then alkylated with 20 mM iodoacetamide for 30 min at 25 °C. The peptides were deposited on MALDI plate with α-cyano-4-hydroxy cinnamic acid (HCCA) saturated solution for co-crystallization (equal volume). The spectra were acquired with a rapifleX TOF/TOF instrument (Bruker, Bremen, Germany) in linear positive mode.

**Flow cytometry**. Cells were scraped and washed twice with PBS before staining reaction (30 min at 4 °C). After washing/centrifugation, cells were fixed with 1%

paraformaldehyde. Flow cytometry analyses were performed with a FACSCanto II flow cytometer and data (mean fluorescence intensity and positive cell percentages) were acquired using FACSDiva software (version 8.0.1, BD Biosciences, San Jose, CA, USA). Results were analyzed with FlowJo (version 10.5.3, TreeStar, Ashland, OR, USA). The following antibodies were used: anti-CD14 Phycoerythrin (PE) (1/33, clone TUK4, Dako, Glostrup, Denmark), anti-TLR4 PE (1/33, FAB6248P, R&D systems), anti-TNFR1 PE (1/33, clone W15099A, Biolegend, San Diego, CA, USA) and anti-TNFR2 PE (1/33, clone 3G7A02, Biolegend). The gating strategy is detailed in Supplementary Fig. 14.

To determine the incorporation of selected peptides (or their hydrolyzed products) within the cytoplasm of several bacterial species, free amine groups (lysine residues and N-terminus) were first conjugated with Atto 488 dye according to manufacturer's recommendations (Atto 488 protein labeling kit, Sigma Aldrich). A total $5 \times 10^4$ colony-forming unit (CFU)/ml were then mixed with fluorescent labeled peptides (5 µg/ml) in 10 mM sodium phosphate buffer (final volume 100 µl) and incubated for 1 h30 at 37 °C before being analyzed by flow cytometry. Where indicated, trypan blue was added before the analysis in order to exclusively consider cytoplasmic fluorescence.

**Mass spectrometry (MS)**. To determine the hydrolysis of innate peptides by the different *Lactobacillus* species present in the female reproductive tract, free amine groups were first biotinylated and labeled peptides (5 µg/ml) were incubated with $5 \times 10^4$ CFU/ml for 3 h or 6 h at 37 °C in 10 mM SPB. Both labeled peptides and their hydrolyzed product were then purified using NeutrAvidin Agarose beads according to manufacturer's recommendations (cell surface protein isolation kit, ThermoFisher Scientific). MS experiments were performed with a 6560 Ion mobility Q-TOF system (Agilent Technologies, Waldbronn, Germany). The LC-chip was interfaced to the MS via the ChipCube interface (Agilent Technologies). The separation was achieved on a large capacity chip integrating a 160 nl enrichment column and a 150 mm × 75 µm ID analytical column (both packed with a Zorbax $C_{18}$ phase) in 1 h gradient time. Other operational conditions as well as data-dependent acquisition settings were previously described[85]. Datafiles were opened using MassHunter Qualitative Analysis software (version 7.0, Agilent Technologies) and individual extracted ion chromatograms were generated at the MS level for each m/z value of interest.

To precisely verify whether the innate peptides highly secreted by the vaginal/cervical mucosa can be used by the predominant *Lactobacillus* species as amino acid source, two elafin fragments (H-AQEPVK*GPVSTK*PGSGCPII-OH and H-PGIK*K*CCEGSCGMACFVPQ-OH), each containing two $^{13}C_6^{15}N_2$-labeled lysines, were synthesized (Pepscan, Lelystad, The Netherlands). Five µg/ml of each peptide were incubated with $6 \times 10^8$ CFU/ml for 6 h at 37 °C in 10 mM SPB. Following two steps of washing, bacterial cells were suspended in 20 µl of Tris-HCl (50 mM, pH 8) containing urea (8 M) and then sonicated for 15 min using a Bioruptor Plus (Diagenode, Seraing, Belgium). The supernatants were retrieved and the samples were successively reduced [5 mM dithiothreitol (DTT), 30 min at 37 °C], alkylated [15 mM iodoacetamide (IAA), 30 min at room temperature] and digested in 2 steps at 37 °C [a mixture of trypsin-Lys-C (Promega, Madison, WI, USA) for 4 h in initial buffer at a ratio enzymes/proteins of 1/25 (w/w) followed by 16 h after dilution to 1 M urea with 50 mM Tris HCl buffer]. Liquid chromatography-tandem MS were then performed using an Acquity M-Class UPLC (Waters Corporation, Milford, MA, USA) coupled to a Q Exactive Plus Hybrid Quadrupole-Orbitrap MS (Thermo Fisher Scientific). The positive ion mode nanoelectrospray was utilized. Symmetry C18 5 µm (180 µm × 20 mm) and HSS T3 C18 1.8 µm (75 µm × 250 mm) were used as trap and analytical column, respectively (Waters). The digested peptides were loaded on the trap column in 98% water 0.1% formic acid (solvent A) (flow rate: 20 µl/min) during 3 min and subsequently separated on the analytical column. The flow rate was constant (600 nL/min) with the following linear gradient: initial conditions 2% acetonitrile 0.1% formic acid (solvent B), at 5 min: 7% solvent B, at 135 min: 30% solvent B, at 150 min: 40% solvent B, at 154 min: 90% solvent B until 158 min and then back in the initial conditions at 162 min until 177 min. The total analytical run time was 180 min. The TopN-MS/MS method was used and N was set to 12. The resolution was set to 70,000 for full MS spectrum acquisition (range 400 to 1600 m/z) and 17,500 for MS/MS. The automatic gain control target was set to $1 \times 10^6$ and $1 \times 10^5$ for MS and MS/MS, respectively. A 2 m/z isolation window was used, the maximum injection time was 50 ms and a normalized collision energy of 28 was utilized for the activation of ions. Data analysis/protein identification was performed using SEQUEST HT search engine and Proteome Discoverer 2.1 software (Thermo Scientific). Databases were downloaded from Uniprot for each bacterial strain of interest (*L. jensenii*: 3763 sequences; *L. crispatus*: 5120 sequences; *L. iners*: 1872 sequences). The parameters for the search were as follows: maximum 2 miss-cleavages and cysteine carbamidomethylation as fixed modification and deamidation (asparagine, glutamine), oxidation (methionine) and label $^{13}C_6^{15}N_2$ of Lysine as variable modifications. The tolerances of precursor and fragment mass were 5 ppm and 0.02 Da, respectively. Percolator node was used to select peptides with a Target FDR set at maximum 1%.

**ELISA**. $1 \times 10^6$ cells/well of a six-well plate were seeded in appropriate growth medium. After 48 h, the secretions of constitutive innate peptides (elafin, S100A7, SLPI and HβD1) by control (HaCat) and HPV16 E6E7-transduced cells were quantified by ELISA using the following commercially available kits: human S100A7 ELISA kit (Biorbyt, Cambridge, UK), human Trappin-2/elafin ELISA kit (Sigma Aldrich), human SLPI ELISA kit (MyBioSource, San Diego, CA, USA) and human HβD1 ELISA kit (MyBioSource).

**ATP measurement**. ATP level in bacteria was determined using the BacTiter-Glo Microbial Cell Viability Assay (Promega) according to manufacturer's recommendations. For each bacterial species, $6 \times 10^7$ CFU/ml were mixed with elafin or S100A7 (5 µg) in 10 mM SPB and incubated 6 h at 37 °C. One hundred µl of bacterial-peptide solution were then mixed with an equal volume of BacTiter-Glo reagent and incubated for 5 min at room temperature. Luminescence was measured using a Centro LB960 microplate luminometer (Berthold Technologies, Bad Wildbad, Germany).

**RNA interference and inhibitors**. Immortalized keratinocytes ($1.5 \times 10^5$ cells per well of a six-well plate) were transfected with ON-TARGETplus SMARTpool siRNAs targeting p65, β-catenin or c-Myc (Dharmacon, Lafayette, CO, USA) using Oligofectamine Transfection Reagent (Invitrogen). Oligos targeting the firefly luciferase gene (siGL2) were used as control. For each experiment, siRNA (40 nM) and 6 µl of Oligofectamine were diluted in 1 ml Optimem (Invitrogen). The cells were incubated with the mixture for 4 h at 37 °C. Twenty-four hours after transfection, cells were either harvested or stimulated with LPS (2 µg/ml) or TNFα (20 ng/ml) and experiments were performed. For the chemical inhibition of NF-kB or c-Myc, BAY 11-7082 (5 µM, Sigma Aldrich) or 10058-F4 (40 µM, Selleckchem, Houston, TX, USA) was added to cell cultures before RNA expression analyses. In order to determine the stability of a given protein, monolayer cultures were pretreated with 100 µM cycloheximide (Sigma Aldrich). Where indicated, DNA methyltransferases were inhibited with 5 µM 5-aza-deoxycytidine (Sigma Aldrich). To analyze the CK1-dependent phosphorylation of β-catenin (ser45), cells were treated with 50 nM calyculin A (Sigma Aldrich) for 30 min before protein extraction.

**Immunohistochemistry, immunofluorescence and in situ hybridization**. Immunohistochemical/fluorescence analyses were performed using standard protocols previously described[86,87] and included the following primary antibodies: anti-SLPI (1/400, NBP1-76803, Novus Biologicals, Centennial, CO, USA), anti-S100A7 (1/500, HPA006997, Atlas Antibodies, Bromma, Sweden), anti-elafin (1/150, clone FL-117, Santa Cruz Biotechnology, Dallas, TX, USA), anti-LL37/CAMP (1/1000, HPA029874, Atlas Antibodies), anti-HβD1 (1/50, PA5-51286, Thermo Fisher Scientific), anti-HβD2 (1/100, ab63982, Abcam, Cambridge, MA, USA), anti-HβD3 (1/300, NB200-117, Novus Biologicals), anti-HβD4 (1/100, clone L13-10-D1, Abcam), anti-HD-5 (1/200, HPA015775, Atlas Antibodies), anti-HD-6 (1/50, NBP1-84281, Novus Biologicals), anti-p65 (1/500, clone D14E12, Cell Signaling Technology, Danvers, MA, USA), anti-E-cadherin (ready to use, clone 36, Ventana Medical Systems, Tucson, AZ, USA), anti-β-catenin (1/100, clone E247, Abcam), anti-c-Myc (1/100, ab32072, Abcam), anti-p16$^{INK4a}$ (1/300, clone JC8; Santa Cruz Biotechnology) and anti-ki67 (ready to use, clone 30-9; Ventana Medical Systems). Immunoperoxidase staining was performed using the mouse or rabbit Envision detection system (Dako) according to manufacturer's instructions. In immunofluorescence, the Alexa Fluor® 488 anti-rabbit antibody (Invitrogen) was used for the secondary reaction and pictures were taken with a Leica SP5 confocal microscope (GIGA in vitro Imaging platform, University of Liege). The actin filaments and nuclei were selectively stained with Alexa Fluor® 568 phalloidin (Invitrogen) and 4,6-diamidino-2-phenylindole (DAPI), respectively. Mouse and rabbit control IgGs (Santa Cruz Biotechnology) were used as negative controls.

In paraffin-embedded tissues/rafts, the detection of HPV DNA by in situ hybridization was performed using the Ventana INFORM HPV III family 16 probe (Ventana Medical Systems) according to the supplier's recommendations.

**Immunohistochemical assessment**. All immunolabelled tissues were evaluated independently by experienced histopathologists. E-cadherin and antimicrobial peptide expressions were assessed by using a semi-quantitative score of the immunoreactivity. As previously described[55,88], an arbitrary scale of the staining intensity (0: undetectable, 1: low, 2: moderate, 3: strong) and extent (0: <5% positive cells, 1: 6–33%, 2: 34–66%, 3: >67%) was used. A global score for each tissue specimen (ranged between 0 and 9) was yielded by multiplying the results obtained with these two scales. The percentage of cells displaying cytoplasmic/nuclear β-catenin immunoreactivity was also evaluated. Scoring of c-Myc was based on nuclear staining in the epithelial cells. Collected results were stratified as follows: 0–5%, 6–33%, 34–66%, 51–75% and > 67%. Regarding p65, the percentage (<10% *versus* > 10%) of cells exhibiting a nuclear immunoractivity was determined.

**Bisulfite genomic sequencing**. Freshly extracted genomic DNA (500 ng) was processed using Bisulfite Conversion kit (Active Motif, La Hulpe, Belgium) according to supplier's recommendations. Conversed *CDH1* (E-cadherin) promoter was amplified using the specific primers listed in Supplementary Table 1. These latter were designed with the Methprimer software (https://www.urogene.org/methprimer/). PCR products were cloned into pJET1.2 vector using CloneJET PCR cloning kit (Thermo Fisher Scientific) and then ligation products were transformed into competent JM109 bacteria using heat shock. Resistant colonies were isolated and recombinant plasmids were

purified (NucleoSpin Plasmid kit, Macherey Nagel). Finally, at least 6 independent plasmids for each condition were sequenced using Sanger method.

**RT-qPCR.** Total RNA was extracted and purified using NucleoSpin RNA isolation kit (Macherey Nagel) according to manufacturer's instructions. The ReliaPrep RNA Miniprep system (Promega) was used for both human and mouse micro-dissected samples. One microgram was reverse transcribed using RevertAid Reverse Transcriptase and oligo dT primers (Thermo Scientific). Quantitative PCR was performed using the FastStart Universal SYBR Green Master mix (Roche, Basel, Switzerland). Both human and mouse primer sequences used for PCR analysis are listed in Supplementary Table 1. The public database PrimerBank was used for the retrieval of most of these sequences[89]. The experiments were performed in triplicate and threshold cycle numbers (Ct) were determined using the ABI-Prism 7900HT Sequence Detection System (Applied Biosystems, Foster City, CA, USA). The relative amounts of mRNA per sample were finally calculated using the ΔΔCt method. Four calibrator genes (HPRT, GAPDH, 18S and TBP) were used to compare the relative quantities of mRNAs between tissue specimens from different patients. With tissue samples from different mice, the expression of 2 housekeeping genes (HPRT and GAPDH) was determined. In vitro experiments (with the same cell line) were normalized to the amount of HPRT mRNA. Where indicated, the amplification efficiency of qPCR reactions was determined using the qPCR efficiency calculator software (Thermo Fisher Scientific).

**Western blotting analysis.** After extraction in 1% SDS buffer (total extraction) containing protease and phosphatase inhibitors (Roche) and quantification (BCA protein assay kit; Pierce, Rockford, IL, USA), twenty micrograms of proteins were separated on 10% SDS-PAGE and transferred to PVDF membranes (Roche). For the extraction of separate protein fractions, the cytoplasmic (20 mM HEPES pH 7.9, 0.1 Mm EDTA, 2 mM MgCl2, 10 mM KCl, 0.2% Nonidet P-40) and nuclear (20 mM HEPES pH 7.9, 0.2 mM EDTA, 1.5 mM MgCl2, 0.63 mM NaCl, 25% glycerol) buffers were used. Blocking was subsequently performed with 5% skim milk in TBS-Tween 0.1% for 1 h and the membranes were then incubated overnight at 4 °C with the following primary antibodies: anti-p65 (1/1000, clone D14E12, Cell Signaling Technology), anti-IκBα (1/1000, clone L35A5, Cell Signaling Technology), anti-NEMO (1/1000, clone FL-419, Santa Cruz Biotechnology), anti-c-Myc (1/1000, ab32072, Abcam), anti-E-cadherin (1/1000, clone 36, BD Biosciences), anti-β-catenin (1/1000, clone E247, Abcam), anti-CK1 (1/1000, #2655, Cell Signaling Technology), anti-β-TrCP (1/1000, #11984, Cell Signaling Technology), anti-phospho-β-catenin (1/1000, Ser45, #9564, Cell Signaling Technology), anti-actin (1/2000, clone AC-15, Sigma Aldrich), anti-HSC-70 (1/2000, clone B-6, Santa Cruz Biotechnology), anti-NBS-1 (1/1000, BD Biosciences) and anti-MEK2 (1/1000, #9125, Cell Signaling Technology). The protein bands were detected using an enhanced chemiluminescence system (Westar ECL-Sun, Cyanagen, Bologna, Italy) and finally quantified by densitometry using ImageJ software. Actin, HSC-70, or NBS-1 was used for the normalization. For uncropped blots, see the Source Data file.

**Gaussia princeps luciferase complementation assay (GPCA).** The coding (or mutated/truncated) sequences of HPV E7 from several alpha and beta genotypes (alpha: HPV16, 18, 33, and 39; beta: HPV8, 38, and 49) were transferred to Gateway-compatible GPCA destination vector pSPICA-N2. The open reading frames (ORFs) encoding proteins involved in IKK or β-catenin destruction complex were obtained from the human ORFeome v7.1 and 8.1 (Dana-Farber Cancer Institute, Boston, MA, USA) and were transferred into pSPICA-N1 plasmid. GPCA expression vectors (pSPICA-N1 and N2) allow the expression of two complementary fragments of the *G. princeps* luciferase (Gluc1 or Gluc2) linked to the N-terminal ends of tested proteins. All constructions were controlled by Sanger sequencing (GIGA-Genomics platform, University of Liege). GPCA experiments were then performed as extensively described previously[90,91]. Briefly, HEK-293T cells were simultaneously transfected with 100 ng of pSPICA-N1-IKK or β-catenin destruction complex proteins and 100 ng of pSPICA-N2-E7 using PEI Max (Polysciences, Warrington, PA, USA). After 24 h, cells were washed, lysed and luciferase enzymatic activity was measured for 10 seconds using a Centro LB960 microplate luminometer (Berthold Technologies, Bad Wildbad, Germany). The luciferase assay system was purchased from Promega. As previously described[90], results were expressed as a luminescence value normalized over the sum of controls [normalized luminescence ratio (NLR)].

**Co-immunoprecipitation.** The detailed procedure for co-immunoprecipitation was previously described[90]. Briefly, the coding sequence of HPV E7 (from alpha HPV16 and 18 as well as beta HPV8) was first cloned into the pCineo-3xFLAG plasmid. HEK 293 T cells were then transfected (PEI Max) with 1.5 μg of FLAG-HPV E7 plasmid or empty vector and 1.5 μg of pSPICA-N1 plasmid expressing proteins involved in IKK or β-catenin destruction complex fused to the Gluc1 fragments of the *G. princeps* luciferase (used in GPCA experiments). After 24 h, cells were lysed, centrifuged at 16,000 g for 20 min and the lysate was incubated overnight at 4 °C with anti-FLAG M2 magnetic beads (ready to use, Sigma Aldrich). After several washes, IP proteins as well as total proteins (input corresponding to 5%) were analyzed by Western blot. The anti-GLuc (1/1000, #E8023, New England Biolabs, Ipswich, MA, USA) was used as primary antibody. For the novel E7 targets highlighted in the present study, the experiments were also

reproduced in the inverse direction. To do so, the immunoprecipitation step was performed using magnetic beads (Cell Signaling) and the following antibodies: anti-NEMO (1/50, ab178872, Abcam), anti-CK1α1 (1/30, ab206652, Abcam) and anti-β-TrCP (1/200, clone D13F10, Cell Signaling). The anti-FLAG antibody (1/2000, clone M2, Sigma Aldrich) was then used for detecting FLAG-E7. Band intensities in the IP lanes were normalized according to intensities in the input lanes and IP (HPV E7)/IP (control) ratio was calculated.

**Chromatin immunoprecipitation (ChIP).** ChIP assays were performed as previously described[92]. Briefly, nuclei were lysed with a buffer containing 50 mM Tris/HCl pH8, 10 mM EDTA, 0.5% SDS and protease/phosphatase inhibitor (Roche). Each sample was sonicated for 10 min using a Diagenode Bioruptor (Diagenode) and then centrifuged at 14,000 g for 15 min. Supernatant (isolated chromatin) was incubated overnight at 4 °C with either anti-p65 antibody (1/50, clone D14E12, Cell Signaling Technology) or control rabbit IgG (C15410206, Diagenode) in ChIP dilution buffer (0.01% SDS, 1.1% Triton X-100, 1.1 mM EDTA, 20 mM Tris/HCl pH8, 167 mM NaCl, protease inhibitor). Protein G magnetic beads (prepared as previously described) were then added in each sample and incubated for 4 h under rotation at 4 °C. After washing and reversal of crosslinking, immunoprecipitated DNA was purified (phenol/chloroform extraction) and subjected to qPCR using *PI3/elafin* and *HβD2* promoter NF-κB binding site specific primers. Sequences targeting a more distal region (-2 kb) on both promoters were used as negative control (Supplementary Table 1).

**Statistical analysis.** Statistical analysis was performed using the GraphPad Prism 8 software (San Diego, CA, USA). For normally distributed variables, two group comparisons were performed using unpaired *t*-tests. One-way or two-way ANOVA was applied to determine the statistical significance of differences between more than two groups/conditions. Dunnett's or Bonferroni post-hoc test was used for multiple comparisons. In the case of discrete (staining scores for assessing innate peptide expression) or non-normally distributed (bacterial population abundances) variables, a Mann-Whitney or Kruskal-Wallis test was performed according to the number of groups. The comparison of phenotypic/immunohistochemical variables (0–5%, 6–33%, 34–66%, and >67% of cells displaying a nuclear c-myc/p65 immunoreactivity or membrane *versus* cytoplasmic/nuclear β-catenin localization) between independent groups was performed using a Fisher's exact test or a $\chi^2$ test according to the number of variables. For the different microbial population ecological indices (α-diversity, richness, evenness), the differences between defined groups were evaluated using a non-parametric Kruskal–Wallis test followed by a two-stage linear step-up procedure of Benjamini, Krieger and Yekutieli. Using the MOTHUR software, sample clustering and beta-dispersion were assessed in a Bray–Curtis dissimilarity matrix with AMOVA (analysis of molecular variance) and HOMOVA (homogeneity of molecular variance) tests, respectively. Regarding the retrospective follow-up study assessing the interplay between HPV infection and BV, both odds ratios and corresponding 95% confidence intervals were calculated using a Fisher's exact test (contingency table) and the results were visualized in a Forest plot. The persistence of each pathological disorder in the presence or absence of the other one was compared using a log-rank (Mantel-Cox) test and the results were presented as Kaplan–Meier plots.

**Reporting summary.** Further information on research design is available in the Nature Research Reporting Summary linked to this article.

## Data availability

Raw amplicon sequencing libraries have been deposited in the NCBI Sequence Read Archive (SRA) repository with the accession code: PRJNA670165. Both the alignment and taxonomical assignment (from phylum to species) of post-sequencing data were performed, based on the SILVA database (v1.38). The proteomics data have been deposited at the ProteomeXchange Consortium via the PRIDE partner repository with the following dataset identifier: PXD022113. Source data for all Figures are provided with this paper. Source data are provided with this paper.

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

## Acknowledgements

We thank the Biobank of the University of Liege, the Laboratory of Clinical Microbiology as well as the GIGA-Immunohistochemistry, in vitro imaging, genomics and viral vectors facilities (University of Liege) for their assistance. We are also grateful to Dr Stephanie Gofflot, Dr Wouter Coppieters, Dr Emmanuel Di Valentin, Manon Deckers, Latifa Karim, Cecile Meex, Raphael Thonon, Kamilia El Kandoussi, and Nancy Rosiere for their technical assistance. We sincerely thank Prof Bernard Joris and Dr Ana Amoroso (Laboratory of Bacterial Physiology and Genetics, University of Liege) as well as Prof Christine Jacobs-Wagner (Microbial Sciences Institute, Yale University) as well as Dr Olivier Peulen (Metastasis Research Laboratory, University of Liege) for helpful discussions. This work was supported in part by the University of Liege [Crédits Sectoriels de Recherche en Sciences de la Santé 2018-2020 (M.H.)], the Belgian Fund for Scientific Research [FNRS; MIS F.4520.20, CDR J.0088.21 (M.H.)], the Televie [PDR Televie 7.8507.19 (M.H.)], the Léon Frédéricq Foundation and the Seventh Framework Program for Research and Technological Development [European Commission: Infect-ERA 2015 (HPV-Motiva)]. C.P., T.L., C.R., and M.A. are Televie/FRIA fellows. A.L., D.B., and EloH are postdoctoral researchers. MH is a Research Associate at the FNRS.

## Author contributions

M.H. designed the study; A.L., D.B., P.R., P.P., E.H., G.C., C.G., G.M., N.S., DomiB, EloH, C.P., T.L., C.R., M.A., P.H., and M.H. performed experiments; R.G. and P.D. collected retrospective clinical follow-up data; A.L., D.B., G.C., B.T., G.M., N.S., M.F., DomiB., G.D., J.-D.C., M.F., P.H., and M.H. interpreted the data; M.M., J.-C.T., G.S.-L., F.B., P.M., and P.D. provided resources; A.L., D.B., and M.H. generated the figures; M.H. wrote the manuscript. All authors discussed the results and commented on the manuscript.

## Competing interests

The authors declare no competing interests.
