## [Peer Review File · Nature Communications]

Reviewers' comments:

Reviewer #1 (Remarks to the Author):

Review of Lebeau et al. Nat. Comm.

The manuscript by Lebeau et al. describes the interaction between human papillomavirus and the vaginal microbiome and its relationship to cancer development. Bacterial vaginosis and HPV infection have long been suspected of having links related to cancer development in women. Thus, the authors initially set out to investigate the literature for links between HPV infection and bacterial vaginosis, where they found a statistically significant association. Subsequently, they went through extensive experimentation of the gene expression changes induced by HPV infection where they found downregulation of antimicrobial peptide expression. This finding led to an investigation of the interactions between the vaginal microbiota and these peptides, where they surprisingly show that vaginal Lactobacilli are not only resistant to the antimicrobial activity of these small proteins, but can utilize them as a carbon source. As my area of expertise is primarily in microbiology and the microbiome, I have focused my review on these areas. In sum, the microbiome data is insufficiently analyzed with inappropriate statistical tests, lack of needed statistical tests and an overall apparent lack of understanding of how to analyze microbiome data.

Major Critiques.

It is not clear from the experiments what the DTT treatment did to the action of the antimicrobial peptides. The authors present a single control in figure 5c that I believe represents exposure with or without DTT. However, this control is not present in the experimental samples where the bacteria are exposed to the antimicrobial peptides. Thus, one cannot interpret if the DTT had any effect on the activity of the antimicrobial peptides as described in the results section. The authors mention that they did this experiment because reducing environment is known to affect AP action, however, as presented, there is no evidence that this occurred in their experiment, thus, it is not apparent that the DTT had any effect. The authors mention that DTT has this effect, but it cannot be seen as presented. Perhaps we are expected to compare the values from Fig 1b to 1c? If so, statistical tests will be needed and presentation of this data made more clear.

The authors also mention that pH did not modify the effect, but in Figure 5e. There appears to be a significant effect of pH on *Lactobacillus jensenii*? Can this be explained?

Presentation of the data in Fig 6a is suspect. One cannot see the sample size or distributions shown. It is paramount to observe the variation and sample size here to determine if the correct tests have been applied, especially considering the very small effect size on Lactobacillus inners.

Furthermore, the authors do not plot longer time points than 3-6 hours. It appears that the effect is just starting, were the assays performed longer? Does the effect increase or go away?

The data shown in Fig. 7B is insufficient for publication. Firstly, the statistical test applied did not reach significance to an alpha-level of 0.05. Thus, these experiments are not statistically significant. In fact it appears that much of the trend is driven by one or two animals, making this finding highly suspect. The authors will need to definitively show that their mouse model has reduced expression of the peptides they are measuring to make the downstream microbiology experiments relevant.

Was there a statistical test applied in Fig 1c to ensure that these measures are not different?

The number of reads per mouse is an irrelevant measure of the microbial community differences. This is often more determinative of DNA extraction or PCR efficiency. Fig 1c

Can the authors include more classic microbiome stacked bar graphs for Fig 7d? The heatmap and numbers in the matrix are hard to interpret and a very unusual way of showing microbiome data. What about PCoA plots and PERMANOVA testing? This is the classic method of testing microbiome beta-diversity differences. These classic microbiome statistics are necessary to determine if the microbiome is in fact different between the mouse groups.

Microbiome data is not normally distributed, and this appears to also be the case for this study. Testing of the relative abundance of different bacterial genera as done in Fig. 7e and f cannot be done with t-tests. Mann-whitney u-tests or other non-parametric tests are required.

Overall the microbiome data is very poorly analyzed and I suggest the authors find someone with more experience with this type of data to present it more appropriately.

Minor critiques

Firstly, not having line numbers on the manuscript makes noting the location of specific issues difficult. All manuscripts submitted should have line numbers. I'll do my best to refer to areas of concern.

I am concerned about the statistical approach in Fig. 2 f. It appears that the authors have combined the three classes of neoplasia together for comparison to the non-cancerous controls. This is odd, considering this is not done in any other part of the figure and lends one to wonder whether or not this was done merely to achieve statistical significance. I would expect to see more similar analysis to that of fig. b and c. If the difference is not significant within the different classes of neoplasia than that should be noted. It is fine to combine them in my opinion, but this should be made clearer and an explanation for why they were combined in Fig 2f, but not any others should be provided.

It is also not clear in fig. 2b,c what the sample sizes are, compared to Fig. 2d where it is more clear.

Many of the methodological workflow images, though helpful, are really not necessary for a journal at this level. Especially the image in Fig. 5a.

The meaning of 'physiological' and 'pathological' conditions referring to the different bacteria is not clear. This should be better explained or removed.

What does 'before being plated for up to 48h' mean? Literally, this would mean that someone took 48 hours to plate the bacteria. Do the authors mean incubated for 48 hours after exposure to antimicrobial peptides for 6 ours? It is unclear from the methods and results how long the exposures and plating was done for.

Acid lactic bacteria needs to be switched for correctness.

Much of the description of the microbiome sequencing and primer testing in the results is unnecessary.

Reviewer #2 (Remarks to the Author):

The authors describe a series of long experiments exploring whether HPV infection alters the vaginal microbiome dynamic equilibrium through impairing host mucosa-Lactobacillus spp. mutualism. The authors should be applauded for the series of extensive experiments conducted to explore their hypothesis and the rigorous experimental design.

The findings that the HPV inhibits basal and pro-inflammatory-induced host defense peptide expression through subverting NF- κ B and Wnt/ β -catenin signaling cascades are novel. This HPV immune evasion adversely impacts on survival of Lactobacillus species promoting an imbalance in the vaginal flora which in turn they hypothesise drives progression.

Although these findings are novel and allow further exploration of causality, the current format of the manuscript is difficult to follow. The aims and course of experimental thinking and design does not come out in the manuscript. There is a long list of experiments although the reader struggles to follow the objectives of each experiment and the hypotheses the aims and experiments are targeting. Although the description would be appropriate for a thesis, the manuscript requires extensive re-writing for a research paper. The meta-analysis is slightly out of space and should be a separate paper. This has been conducted previously and I could find how the authors describe this in context with other meta-analysis. In the context of experiments exploring the vaginal microbiome through next generation sequencing, wouldn't a meta-analysis on the vaginal microbiome and HPV be more appropriate?

Abstract: The abstract in its current format is difficult to follow. Although the journal recommends unstructured abstracts, the presentation should follow the intro, aims, methods, results and conclusion 'informal' structure.

Introduction:

- This is rather long and difficult to follow. The introduction should be shortened. Some results are presented in the introduction and the aims are not clearly stated.
- I was not sure what the yellow section signify.

Methods:

- what was the reason for inclusion of vulvar cases of cancer
- although there is a long and thorough description of the experiments, there is no clear understanding of the flow and reason for these experiments and the questions they are trying to

answer. This should be shortened and many details could be transferred in the supplement. This section need to lead the reader on the results that they will read and the aims that these experiments are trying to answer and explore.

Results: very detailed presentation of results but should be shortened and present main findings. Results showed in Figures and tables should not be repeated for ease of reading.

the authors have not added page numbers so it is difficult to comment

the sentence: These latter results were, however, never confirmed/validated.

This sentence is not clear and not clear. confirmed by whom? or validated? this should be commented rather in the limitations of the discussion.

Discussion:

- not sure why the authors did a meta-analysis on BV and HPV and not vaginal microbiome and HPV
- not sure what this sentence refers to as their meta-analysis did not include 16sRNA sequencing data 'Furthermore, the levels of both *L. crispatus* and *L. jensenii* were progressively decreasing with disease severity, irrespective of the detected HPV genotypes'
- Reference in the figure should not be made in the discussion but in results. The discussion should attempt to draw conclusions, describe strengths and limitation and bring in context with the existing literature.
- The presenting the argument that the HPV causes the dysbiosis, but then that dysbiosis promotes disease progression. Do they have evidence for the latter? The authors conclude by saying 'but it could work in either direction'?

Reviewer #3 (Remarks to the Author):

The manuscript by Lebeau et al., provides a number of different studies from a literature review of BV and HPV, a mouse model and from tissue culture cells to suggest that HPV changes the vaginal microbiome to promote BV. The main problem with this story is that HPV infects only a small proportion of the cells in the cervical-vaginal region and it is hard to imagine how it could have intracellular influences on a majority of cells not infected. It is important to state which HPV type E7 is under investigation as the biochemical properties are not all uniform. That there is an association between HPV and BV or an altered vaginal microbiome is not hard to imagine, as the risk factors for one likely overlap the risk factors for the other. Strong data showing an etiological relationship

requires proper prospective studies, which as the authors acknowledge are not very abundant. Thus, we are left with an association not strongly supporting the idea that HPV precedes BV, particularly when many studies recruit women and/or follow them with detection of HPV. The K14-HPV16 transgenic mouse model might be interesting, but the administration of hormones (estrogen) likely influences the mouse microbiome making it hard to interpret the reported studies. Lastly, the cellular studies might be true for cells infected with HPV, but as noted above they represent a minority of the cells in the vagina and the authors have not accounted for this. Why did the investigators select the HPV E7 from the types listed (HPV16, 18, 33, 39, 8, 38 and 49)? The manuscript was written in such a manner that it was often difficult to follow the logic of experiments.

Many facts or data are stated without specific reference to the experiments or the statistical comparison p-values in the written document. For instance, what was the evidence that keratinocytes stably maintained episomal HPV18 genomes (page 24)?

Point by point response ("HPV infection alters vaginal microbiome dynamic equilibrium through impairing host mucosa-Lactobacillus spp. mutualism", *Nature Communications*, NCOMMS-20-14429B-Z)

Reviewers' comments:

Reviewer #1 (Remarks to the Author):

The manuscript by Lebeau et al. describes the interaction between human papillomavirus and the vaginal microbiome and its relationship to cancer development. Bacterial vaginosis and HPV infection have long been suspected of having links related to cancer development in women. Thus, the authors initially set out to investigate the literature for links between HPV infection and bacterial vaginosis, where they found a statistically significant association. Subsequently, they went through extensive experimentation of the gene expression changes induced by HPV infection where they found downregulation of antimicrobial peptide expression. This finding led to an investigation of the interactions between the vaginal microbiota and these peptides, where they surprisingly show that vaginal Lactobacilli are not only resistant to the antimicrobial activity of these small proteins, but can utilize them as a carbon source. As my area of expertise is primarily in microbiology and the microbiome, I have focused my review on these areas. In sum, the microbiome data is insufficiently analyzed with inappropriate statistical tests, lack of needed statistical tests and an overall apparent lack of understanding of how to analyze microbiome data.

We totally agree with the Reviewer's general comment. Initially, the 16S rDNA-seq metagenomic data were analyzed by non-specialists in the field. As judiciously requested, the collected raw data were completely re-evaluated by both Prof. Georges Daube and Dr. Bernard Taminiau (Department of Food Sciences and Microbiology, University of Liege) who have a 10-year experience in this particular research domain and have published over 30 articles (with numerous collaborators) in the last 5 years. The detailed procedure (16S rDNA sequencing, sequence and data analysis) was added in the "Materials and Methods" section. Of note, during the post-reviewing/correction period, the number of analyzed mice was increased from 10 to 12 per group in order to improve the quality, robustness and statistical significance of presented results. These novel experiments were approved by the institutional ethics committee.

Major Critiques

It is not clear from the experiments what the DTT treatment did to the action of the antimicrobial peptides. The authors present a single control in figure 5c that I believe represents exposure with or without DTT. However, this control is not present in the experimental samples where the bacteria are exposed to the antimicrobial peptides. Thus, one cannot interpret if the DTT had any effect on the activity of the antimicrobial peptides as described in the results section. The authors mention that they did this experiment because reducing environment is known to affect AP action, however, as presented, there is no evidence that this occurred in their experiment, thus, it is not apparent that the DTT had any effect. The authors mention that DTT has this effect, but it cannot be seen as presented. Perhaps we are expected to compare the values from Fig 5b to 5c? If so, statistical tests will be needed and presentation of this data made more clear.

We understand the misunderstanding about the rationale of performing this experiment as well as the expected comparison of results presented in Figures 6B and 6C [of note, the Figures were renumbered due to the addition of the retrospective cohort study (Figure 2)]. Actually, the reduction of disulfide bonds has been previously shown to modify the activity of some innate peptides (especially H β D1) against a few (not all) bacterial species (Schroeder et al. *Nature* 2011; Schroeder et al. *Mucosal Immunol* 2015). In order to test this hypothesis, all the host defense peptides were first incubated with DTT (2 mM) and the reduction of disulfide bridges (as well as the chemical peptide synthesis) were verified by MALDI mass spectrometry (Supplementary Figure 9). As shown in Figure 6C (and also mentioned by the Reviewer), the antimicrobial activity of defensins was globally unchanged in a reducing environment but, importantly, the significant protective effect displayed by both elafin and S100A7 on *Lactobacilli* was still observed. The text related to these experiments was amended (“Results” section) in order to avoid any misunderstanding and to clarify the conclusions. Given the absence of differences, we sincerely believe that these latter data do not merit further attention and, therefore, we did not create a graph for the “formal” comparison of results presented in Figure 6B and 6C. However, in case of acceptance, the valuable open access policy of the Journal (with the publication of source data) will allow to easily upload the raw data and ascertain all conclusions.

The authors also mention that pH did not modify the effect, but in Figure 5e. There appears to be a significant effect of pH on Lactobacillus jensenii? Can this be explained?

We understand the misunderstanding. We reproduced the experiments and similar results were obtained. In fact, the pH modifies the global in vitro growth of *Lactobacillus jensenii* (significantly higher at pH7.4 than pH5.8) but changing the pH from 7.4 to 5.8 did not repress the positive effect of both S100A7 and elafin on *Lactobacillus jensenii* survival. Indeed, whatever the pH [pH7.4 (Figure 6B) or pH5.8 (Figure 6E)], this significant beneficial effect was observed. This information was clarified in the “Results” section.

Presentation of the data in Fig 6a is suspect. One cannot see the sample size or distributions shown. It is paramount to observe the variation and sample size here to determine if the correct tests have been applied, especially considering the very small effect size on Lactobacillus iners. Furthermore, the authors do not plot longer time points than 3-6 hours. It appears that the effect is just starting, were the assays performed longer? Does the effect increase or go away?

Regarding the first part of Reviewer’s comment, the results (each time point) represent the means \pm SEM of at least four independent experiments performed in duplicate (more precisely, at 9h, 4 independent experiments were performed and up to 10 for the other time points). For the purpose of visual representation/gain of space, and because, in case of acceptance, all raw data will be published (no results will be “hidden” to the readers), in contrast to all the other graphs in the present study, each individual data point was not shown. *P* values were determined using a two-way ANOVA followed by Bonferroni’s multiple comparison post-hoc test. This information was added in Figure 7 legend. Overall, all statistical analyses were verified by a bioinformatician (Prof. Olivier Peulen, Metastasis Research Laboratory, University of Liege).

In order to determine whether the beneficial effect of innate peptides constitutively expressed by the vaginal/cervical mucosa appears directly or needs some latency, elafin and S100A7 were added to lactic acid bacteria and several time points were investigated. To address the second part of the Reviewer’s comment, longer incubations (9h) were also performed with both *L. crispatus*

and *L. jensenii*. The data were added in Figure 7A. Importantly, for all the *Lactobacillus* species, a significant increased percentage of surviving colonies was only detected after 3 to 6h of incubation which represents the necessary time for the bacteria to cleave, internalize and use these host secreted peptides as amino acid source (as shown in Figure 7B-E). As commonly practiced for both eliminating variables and focusing on the compounds of interest, no nutrient (serum) was added during the incubation (starvation assay), explaining why the number of surviving colonies was decreasing over the time. This important information is now mentioned in the "Results" section as well as in the Figure legend.

The data shown in Fig. 7B is insufficient for publication. Firstly, the statistical test applied did not reach significance to an alpha-level of 0.05. Thus, these experiments are not statistically significant. In fact it appears that much of the trend is driven by one or two animals, making this finding highly suspect. The authors will need to definitively show that there mouse model has reduced expression of the peptides they are measuring to make the downstream microbiology experiments relevant.

As mentioned by the Reviewer, despite a strong tendency for SLPI, S100A7 and m β D15, in the initial version of our manuscript, the statistical significance was not reached for 4 out of 6 mouse orthologs of human defensin(-like) peptides. As requested/suggested, the number of analyzed mice was increased and with the exception of m β D4, all antimicrobial peptides have been shown to be significantly down-regulated in HPV16-expressing epithelia (K14-HPV16 mice) compared to their normal (uninfected) counterparts (FVB/n mice). These results are shown in Figure 8B and mentioned in the "Results" section.

Was there a statistical test applied in Fig 7c to ensure that these measures are not different?

Given the next comment of the reviewer, the initial Figure 7C (renumbered 8C) was removed and replaced by bacterial α -diversity and richness analyses. The differences between defined groups were evaluated using a non-parametric Kruskal–Wallis test followed by a two-stage linear step-up procedure of Benjamini, Krieger and Yekutieli. This information was added in the "Materials and Methods" section as well as in Figure 8 legend.

The number of reads per mouse is an irrelevant measure of the microbial community differences. This is often more determinative of DNA extraction or PCR efficiency. Fig 7c

We totally agree with the Reviewer's comment. As mentioned above, the 16S rDNA-seq metagenomic data were completely re-evaluated by two local specialists. The ecological indices (intrinsic diversity, richness deduced from Chao1 index and evenness deduced from Simpson index) of microbial populations detected in the cervico-vaginal lavage samples were first assessed at both the genus and species levels. No significant difference was detected between the four defined groups [FVB/n (week 0 *versus* week 12) and K14-HPV16 (week 0 *versus* week 12)]. Strikingly, when data reported in K14-HPV16 mice were separated depending on the grade of (pre)neoplastic lesions observed within the lower genital tract, a significant increase in bacterial richness was detected in mice displaying a high-grade precancer (HSIL). Concurrently, both the bacterial α -diversity and evenness were reduced, meaning that, despite a higher global number of genera/species, the distribution of these latter (relative abundance) in vaginal lumen is less uniform in mice with histologically-proven HSIL compared to their control (week 0) counterparts

or those diagnosed with hyperplasia/LSIL. These important data supporting a dysbiosis of vaginal microbiome following HPV-related carcinogenesis are shown in Figure 8C-D as well in the Supplementary Figure 11. We thank the Reviewer for his comment which allowed to significantly improve our article and to avoid any misinterpretation of presented results.

Can the authors include more classic microbiome stacked bar graphs for Fig 7d? The heatmap and numbers in the matrix are hard to interpret and a very unusual way of showing microbiome data. What about PCoA plots and PERMANOVA testing? This is the classic method of testing microbiome beta-diversity differences. These classic microbiome statistics are necessary to determine if the microbiome is in fact different between the mouse groups.

As requested, the heatmap (present in the previous version of our manuscript) was replaced by stacked bar charts depicting the relative abundance of the twelve main bacterial orders detected in control (FVB/n) and K14-HPV16 mice. Accounting for over 90% of the total microbial community in the vast majority of samples (44/48, 91.7%), the bacteria belonging to the order of Lactobacillales or Pasteurellales were clearly predominant in the cervix/vagina of our mouse model, regardless of the HPV status. The results are shown in Figure 8F.

In the present study, the β -diversity of the vaginal microbial profile in K14-HPV16 mice was visualized using a Bray-Curtis dissimilarity matrix-based non-metric multidimensional scaling (NMDS) model. As revealed by AMOVA (analysis of molecular variance)-based group clustering testing and clearly illustrated by the β -diversity plot (Figure 8E), the microbial profiles detected in HSIL-positive K14-HPV16 mice were significantly distinct from those of hyperplasia/LSIL-positive ($p=0.002$) or control mice ($p=0.0005$). In addition, the homogeneity in the cervico-vaginal microflora tended to be different in mice diagnosed with a precancerous lesion compared to the control group, as indicated by HOMOVA (homogeneity of molecular variance) testing (HSIL *versus* control, $p=0.09$; hyperplasia/LSIL *versus* control, $p=0.08$). This information was added in the "Results" section. Once again, we sincerely thank the Reviewer for his judicious comment which allowed to significantly improve our results.

Microbiome data is not normally distributed, and this appears to also be the case for this study. Testing of the relative abundance of different bacterial genera as done in Fig. 7e and f cannot be done with t-tests. Mann-whitney u-tests or other non-parametric tests are required.

We agree with Reviewer's comment. Accordingly, the statistical differences between defined groups (for the relative abundance of specific bacterial family/genera) were re-evaluated using a non-parametric Kruskal–Wallis test. This information was added in the "Materials and Methods" section as well as in Figure 8 legend.

Overall the microbiome data is very poorly analyzed and I suggest the authors find someone with more experience with this type of data to present it more appropriately.

As judiciously suggested by the Reviewer, and as mentioned above, the 16S rDNA-seq metagenomic data were completely re-evaluated by two field specialists.

Minor critiques

Firstly, not having line numbers on the manuscript makes noting the location of specific issues difficult. All manuscripts submitted should have line numbers. I'll do my best to refer to areas of concern.

As requested, the line numbers were added in the main document.

I am concerned about the statistical approach in Fig. 2 f. It appears that the authors have combined the three classes of neoplasia together for comparison to the non-cancerous controls. This is odd, considering this is not done in any other part of the figure and lends one to wonder whether or not this was done merely to achieve statistical significance. I would expect to see more similar analysis to that of fig. b and c. If the difference is not significant within the different classes of neoplasia than that should be noted. It is fine to combine them in my opinion, but this should be made clearer and an explanation for why they were combined in Fig 2f, but not any others should be provided.

To address the Reviewer's comment, the 9 graphs presented in Figure 3F were re-created in order to mimic those in Figures 3B and 3C and to compare each group of HPV-positive (pre)neoplastic lesions (LSIL, HSIL or SCC) to the normal squamous epithelium from uninfected tissue specimens. Considering the discrete variables (immunohistochemistry scores), the data were analyzed using a non-parametric Kruskal–Wallis test followed by Dunn post-hoc test. This information was added in Figure 3 legend. Of note, new immunohistochemical experiments were performed to respond to one comment from Reviewer 3 and we used this opportunity to increase the number (from 45 to 65) of analyzed HPV-positive lesions.

It is also not clear in fig. 2b,c what the sample sizes are, compared to Fig. 2d where it is more clear.

As requested, the Figures 3B and 3C were re-created and each individual data point is now represented.

Many of the methodological workflow images, though helpful, are really not necessary for a journal at this level. Especially the image in Fig. 5a.

We understand the Reviewer's comment. However, we sincerely believe that the two methodological workflow images presented in Figures 3A and 6A are not only helpful for the readers (as mentioned by the Reviewer) but also show the "robustness" of used approaches in the present study. Indeed, in the majority of studies, transcriptional analyses are still performed using non-microdissected specimens (containing the cell population of interest with many others) without determining the amplification efficiency of each qPCR reaction (inducing an additional bias when the results are compared). Similarly, the bacterial growth is still frequently estimated by turbidimetric analysis (a fast but very imprecise approach) and not by counting the bacterial colony forming units (CFUs) on agar plates. By using computerized detection and counts, this latter procedure allows to obtain both more precise and standardized results. For all these reasons, we decided to keep these two Figures.

The meaning of 'physiological' and 'pathological' conditions referring to the different bacteria is not clear. This should be better explained or removed.

We understand the confusion, especially for the term “physiological condition”. Accordingly, it was removed from the entire manuscript. The adjective "pathological" was maintained (4 times in the text) when referring to a disorder/pathology [HPV infection, (pre)neoplastic lesions or bacterial vaginosis] without making mention of bacterial species.

What does ‘before being plated for up to 48h’ mean? Literally, this would mean that someone took 48 hours to plate the bacteria. Do the authors mean incubated for 48 hours after exposure to antimicrobial peptides for 6 ours? It is unclear from the methods and results how long the exposures and plating was done for.

Yes, the bacteria were exposed to antimicrobial peptides for 6h. Bacterial suspensions were then plated on Columbia Blood agar plates for 24 or 48h before calculating the CFUs. This information was added in both the “Results” and “Materials and Methods” sections.

Acid lactic bacteria needs to be switched for correctness.

This change was made in the manuscript.

Much of the description of the microbiome sequencing and primer testing in the results is unnecessary.

As requested, the text describing both the microbiome sequencing and primer testing was removed from the “Results” section.

Reviewer #2 (Remarks to the Author):

The authors describe a series of long experiments exploring whether HPV infection alters the vaginal microbiome dynamic equilibrium through impairing host mucosa-Lactobacillus spp. mutualism. The authors should be applauded for the series of experiments conducted to explore their hypothesis and the rigorous experimental design.

The findings that the HPV inhibits basal and pro-inflammatory-induced host defense peptide expression through subverting NF- κ B and Wnt/ β -catenin signaling cascades are novel. This HPV immune evasion adversely impacts on survival of Lactobacillus species promoting an imbalance in the vaginal flora which in turn they hypothesise drives progression.

Although this findings are novel and allow further exploration of causality, the current format of the manuscript is difficult to follow. The aims and course of experimental thinking and design does not come out in the manuscript. There is a long list of experiments although the reader struggles to follow the objectives of each experiment and the hypotheses the aims and experiments are targeting. Although the description would be appropriate for a thesis, the manuscript requires extensive re-writing for a research paper. The meta-analysis is slightly out of space and should be a separate paper. This has been conducted previously and I could find how the authors describe this in context with other meta-analysis. In the context of experiments exploring the vaginal microbiome through next generation sequencing, wouldn't a meta-analysis on the vaginal microbiome and HPV be more appropriate?

First of all, we thank the Reviewer for his/her positive comments regarding the novelty of presented data and the rigorous experimental design used in the present study.

Actually, the difficulty to follow the logical flow of the study was also mentioned by the Reviewer 3 and this is very likely linked to the high number of experiments as well as to the various technological approaches used. As detailed below, to address this issue, the manuscript was edited in deep and both the rationale of each experiments and the conclusions are now better explained.

We understand the rationale of Reviewer's comment regarding our meta-analysis. However, we sincerely believe that the related findings are helpful (if not essential) and fit now perfectly in the beginning of the present study. We decided to keep this analysis for 3 main reasons:

- 1) To the best of our knowledge (and as mentioned by the Reviewer as well as in the "Discussion" section), two previous systematic reviews of the literature were conducted for determining an association between HPV infection and the imbalance of vaginal microflora (BV) (Gillet et al. *BMC Infect Dis* 2011; Liang et al. *Infect Agent Cancer* 2019). However, these studies are either "outdated/old" or only partially analyze the existing literature. Performed in close collaboration with Dr Jean-Damien Combes (from Dr Gary Clifford's group, International Agency for Research on Cancer/WHO, Lyon, France), our meta-analysis includes many more studies (34 *versus* 12 and 13) and patients (39,819 *versus* 6,372 and 17,396) than the previously published ones, providing more accurate results. Moreover, sub-groups analyses (e.g. LSIL *versus* HSIL or low-risk *versus* high-risk HPV) were also, for the first time, carried out.

- 2) Without exactly determining the involved anaerobic bacteria species (although the frequent observation of clue cells under the microscope strongly suggests the presence of *Gardnerella vaginalis*), the routine diagnosis of BV (using, for example, the Amsel criteria or Nugent score) by experienced cytopathologists clearly attests a dysbiosis of vaginal microbiota. Therefore, in the present case, the costly use of 16S rDNA sequencing on thousands of women will very likely not bring to light other conclusions than those provided by our meta-analysis.

- 3) Overall, the combined odds ratio which emerged from the present meta-analysis was 1.62 (95% CI: 1.53-1.73, $p < 0.0001$), indicating a positive association between HPV infection and BV. However, because most published studies (32/34, 94.1%) were cross-sectional, this analysis only allowed to determine the association/coexistence but not the temporal sequence between these two pathological disorders. Based on these data, it was still unclear whether these two conditions were biologically related or whether they were simply frequently detected in the same individuals because of their common risk factors. To address this important point (highlighted by the Reviewer 3), we performed a retrospective cohort study including over 6,000 women and the results are shown in Figure 2. Therefore, our meta-analysis now perfectly "introduces" the second part of our results. This information was added in both the "Results" and "Discussion" sections.

Abstract:

The abstract in its current format is difficult to follow. Although the journal recommends unstructured abstracts, the presentation should follow the intro, aims, methods, results and conclusion 'informal' structure.

We understand the rationale of Reviewer's comment but following the introduction, aims, methods, results and conclusion in an abstract of approximately 150 words is very challenging (if not impossible). In the last few weeks, we read many articles recently published in *Nature Communications* as well as the submission guide of the Journal which recommends "to provide a general introduction to the topic and a brief non-technical summary of main results". Therefore,

in order to address as appropriately as possible the Reviewer's comment (and to still follow the author guidelines), the conclusions were entirely re-written and we only kept the main results.

Introduction:

- This is rather long and difficult to follow. The introduction should be shortened. Some results are presented in the introduction and the aims are not clearly stated.

As requested, the "Introduction" section was shortened (especially the last two paragraphs) and the unresolved issues (aims) are now clearly mentioned. Similarly to what we read in all articles published in *Nature Communications*, the last paragraph of the Introduction summarizes the main results of our study. Therefore, this paragraph was kept but shortened.

- I was not sure what the yellow section signify.

In the previous version of our manuscript (evaluated by the three Reviewers), the yellow sections highlighted our responses to Editors' comments (first assessment before sending the paper out for external reviewing).

Methods:

- what was the reason for inclusion of vulvar cases of cancer

Given that virtually all (>99.9%) (pre)neoplastic lesions arising from the uterine cervix are etiologically linked to high-risk HPV infection, it is impossible to compare the expression profile of one protein of interest (in the present case: nuclear p65 immunoreactivity) in HPV-positive cervical neoplasms and in their HPV-negative counterparts. To address this issue, 42 tissue specimens of vulvar cancer (24 HPV-positive and 18 HPV-negative) were included. This information was added in Figure 4 legend.

- although there is a long and thorough description of the experiments, there is no clear understanding of the Flow and reason for these experiments and the questions they are trying to answer. This should be shortened and many details could be transferred in the supplement. This section need to lead the reader on the results that they will read and the aims that these experiments are trying to answer and explore.

According to the author guidelines, the "Materials and Methods" section should be placed after the Discussion (and not between the Introduction and the Results). Therefore, this section was relocated accordingly which significantly improves the flow of the manuscript. Moreover, the submission guide of the Journal mentions the following recommendations: "the Methods section appears in all online original research articles and should contain all elements necessary for interpretation and replication of the results." In order to avoid any misinterpretation of presented results and to facilitate the reproducibility of our work, the "Materials and Methods" section was not shortened.

As mentioned above, the manuscript was edited and re-read by several colleagues in order to better explain both the justification of each experiment and the related conclusions.

Results:

- very detailed presentation of results but should be shortened and present main findings. Results showed in Figures and tables should not be repeated for ease of reading.

As requested, the results shown in Figures and Tables are no longer repeated in order to facilitate the reading.

We respectfully disagree with Reviewer's comment regarding the shortening of the results. In fact, the present study already contains 12 Supplementary Figures and 4 Supplementary Tables. Furthermore, as mentioned above, in the last few weeks, we read many health science articles recently published in *Nature Communications* and multiple panels are always presented in the main Figures.

- the authors have not added page numbers so it is difficult to comment

As requested by Reviewers 1 and 2, both the line and page numbers were added in the main document.

- the sentence: These latter results were, however, never confirmed/validated. This sentence is not clear and not clear. confirmed by whom? or validated? this should be commented rather in the limitations of the discussion.

We understand the Reviewer's misunderstanding. Actually, interestingly, both elafin and S100A7, which are drastically down-regulated in HPV-infected tissues (Figure 3), have been previously listed within the high-affinity group of c-myc targets (Fernandez et al. *Genes & development* 2003). However, these latter results obtained by high-throughput screening were never confirmed/validated (it is worth to note that many false positive results exist using large-scale screening methods). This information was added to explain why we originally focused our attention on c-myc and the Wnt/ β -catenin signaling pathway which activates its transcription. The text was edited accordingly in the "Results" section.

Discussion:

- not sure why the authors did a meta-analysis on BV and HPV and not vaginal microbiome and HPV

This point was already extensively discussed above.

- not sure what this sentence refers to as their meta-analysis did not include 16sRNA sequencing data 'Furthermore, the levels of both L. crispatus and L. jensenii were progressively decreasing with disease severity, irrespective of the detected HPV genotypes'

The above sentence refers to a paper published by Mitra et al. (*Scientific Reports* 2015) and showing reduced levels of *Lactobacillus* species following HPV-related carcinogenesis. The reference was added at the end of this sentence ("Discussion" section).

- Reference in the figure should not be made in the discussion but in results. The discussion should attempt to draw conclusions, describe strengths and limitation and bring in context with the existing literature.

As requested, references to the Figures are no longer present in the “Discussion” section.

- *The presenting the argument that the HPV causes the dysbiosis, but then that dysbiosis promotes disease progression. Do they have evidence for the latter? The authors conclude by saying 'but it could work in either direction'?*

Collectively, our study reveals a new viral immune evasion strategy (HPV-dependent down-regulation of both basal and pro-inflammatory-induced host defense peptides through subverting NF- κ B and Wnt/ β -catenin signaling cascades) which, by its negative impact on lactic acid bacteria (reduction of their carbon source), ultimately causes the dysbiosis of vaginal microbiota (BV). As mentioned by the Reviewer, we finished our conclusion by mentioning that the oxidative stress resulting from BV establishment/persistence would then promote the progression of HPV-related (pre)neoplastic lesions supporting that the association between HPV and BV is complex and work in both directions. Actually, BV has been presumed to be a risk factor for HPV-related carcinogenesis for a long time and, interestingly, this assumption was recently confirmed. Indeed, recent data (Mitra et al. *Nature Communications* 2020) reported that an anaerobic vaginal microbiome composition is associated with a lower regression rate of untreated cervical intraepithelial neoplasia (CIN2) lesions. This information is mentioned in both the “Introduction” and “Discussion” sections.

Reviewer #3 (Remarks to the Author):

The manuscript by Lebeau et al., provides a number of different studies from a literature review of BV and HPV, a mouse model and from tissue culture cells to suggest that HPV changes the vaginal microbiome to promote BV. The main problem with this story is that HPV infects only a small proportion of the cells in the cervical-vaginal region and it is hard to imagine how it could have intracellular influences on a majority of cells not infected.

Interesting point raised by the Reviewer. We have to confess that we discussed this peculiar point several times with both the team members and collaborators during the running of the project and its reviewing. Our extensive *in vitro* data, reinforced by findings from both an *in vivo* model (K14-HPV16 mice) and a large retrospective follow-up study including over 6,000 women (judiciously requested by the Reviewer and discussed below), unequivocally support a causal relationship between HPV infection and BV development. However, as mentioned by the Reviewer and discussed in our manuscript, an asymptomatic HPV infection which is cleared by the immune system after a few weeks most likely does not have the same impact on vaginal microbiota than an extensive persisting (pre)neoplastic lesion. It is also important to notice that the global decrease in innate peptide expression (carbon source for *Lactobacilli*) was not only detected in HPV-infected cells, but also in the morphologically normal (p16^{INK4a}-negative) squamous epithelium adjacent to HPV-positive (pre)neoplastic lesions. The well-known concomitant down-regulation of pro-inflammatory cytokines (e.g. TNF α , IL-1 β) resulting from both NF- κ B and Wnt/ β -catenin signaling impairment is very likely to explain the extension of this immunosuppressive effect in the lesional (micro)environment (absence of paracrine stimulation), showing that the impact of HPV extends well beyond their infected cells. These important results are shown in Supplementary Figure 4 and are mentioned in both "Results" and "Discussion" sections.

It is important to state which HPV type E7 is under investigation as the biochemical properties are not all uniform.

In the present study, we demonstrated that three host proteins involved in NF- κ B and/or Wnt/ β -catenin signaling pathways (NEMO, CK1 and β -TrCP) not only interact with E7 from high-risk alpha HPVs (HPV16, HPV18, HPV33 and HPV39) but also with E7 from beta HPV genotypes (HPV8, HPV38 and HPV49) which have recently emerged as “facilitators” in UV-related cutaneous SCC development. Altogether, our findings indicate the importance of these newly discovered targets for viral persistence/carcinogenesis. As requested, the description of the HPV genotype associated to the E7 oncoprotein under investigation is now better mentioned in the “Results” section as well as in the Figure legends.

That there is an association between HPV and BV or an altered vaginal microbiome is not hard to imagine, as the risk factors for one likely overlap the risk factors for the other. Strong data showing an etiological relationship requires proper prospective studies, which as the authors acknowledge are not very abundant. Thus, we are left with an association not strongly supporting the idea that HPV precedes BV, particularly when many studies recruit women and/or follow them with detection of HPV.

We totally agree with the Reviewer's comment. Most studies (32/34, 94.1%) available in the literature have a cross-sectional design and our meta-analysis only allowed to determine the association/coexistence but not the temporal sequence between HPV infection and BV. To address this important point, with the precious help of the Department of Pathology (Prof. Philippe Delvenne and Dr Roland Greimers, University Hospital Center of Liege), we performed a retrospective cohort study including 6,085 women. The technical details (e.g. HPV testing, exclusion criteria, BV diagnosis,...) were added in the “Materials and Methods” section. The data collection from personal health records was approved by the Ethics Committee of the University Hospital of Liege (#2021-117). Overall, we reviewed data from 18,475 patient visits (6,085 first and 12,390 follow-up visits) completed by the selected women. We first observed that each individual pathological disorder was more frequently diagnosed when the other one was preceding in time (HPV infection: OR=1.83, $p<0.0001$; BV development/diagnosis: OR=2.35, $p=0.0007$). To further evaluate the biological association between HPV and BV, the persistence of HPV infection according to BV status as well as the inverse evaluation were also determined. We showed that the duration of HPV infections was significantly longer in BV-positive women compared to their counterparts displaying a normal vaginal microbiome dominated by *Lactobacillus* spp ($p<0.0001$). Strikingly, the opposite observation was also made ($p<0.0001$). It is, however, important to note that the consensus guidelines recommend repeat cytology/HPV testing every 6 months for HPV-positive women while intervals of 3 years are usually acceptable in case of HPV negativity. This difference is likely to explain why the positive impact of HPV infection on BV persistence was only detected after a 2.5-year period and then increased over the time (bias of follow-up frequency between patient groups). These results clearly support that, besides the long-standing [and recently confirmed (Mitra et al. Nature Communications 2020)] assumption that BV increases the risk of HPV-related carcinogenesis, persistent/symptomatic HPV infections may also induce changes within the vaginal lumen that would ultimately disrupt the microbial balance. The dysbiosis of vaginal microbiome detected following HPV-related carcinogenesis in the K14-HPV16 transgenic mouse model further supports the retrospective clinical data and, overall, the novel paradigm highlighted in the present study. The results are

shown in Figure 2, mentioned in the "Results" section and discussed in the manuscript. We sincerely thank the Reviewer for his judicious comment which allowed to significantly improve our study.

The K14-HPV16 transgenic mouse model might be interesting, but the administration of hormones (estrogen) likely influences the mouse microbiome making it hard to interpret the reported studies.

In fact, we originally had the same apprehension as the Reviewer and estrogen-untreated control FVB/n mice (n=8) were part of the experience. After 12 weeks in the same housing conditions as estrogen-treated mice, cervico-vaginal lavage samples were collected, DNA was extracted, V5-V6 hypervariable regions of the 16S rRNA gene were amplified and sequencing was performed (Illumina MiSeq). Importantly, we showed that long-term estrogen treatment does not modify the composition of vaginal microbiota [the ecological indices (α and β -diversities, richness and evenness) were not significantly different between estrogen-treated and untreated mice]. This is in agreement with a previous study (De Gregorio et al. *Scand J Lab Anim Sci* 2018) which reported an altered estrous cycle for a few days following estrogen administration but the cultivable vaginal bacteria remained unchanged. These control results (not shown in the first version of our paper) are now shown in Supplementary Figure 12 and mentioned in the "Results" section.

Lastly, the cellular studies might be true for cells infected with HPV, but as noted above they represent a minority of the cells in the vagina and the authors have not accounted for this.

This point was already extensively addressed above.

Why did the investigators select the HPV E7 from the types listed (HPV16, 18, 33, 39, 8, 38 and 49)?

Most experiments were performed with E7 oncoprotein from HPV16 because this genotype is, by far, the most frequently detected in (pre)neoplastic lesions, irrespective of their anatomical origins. The results (especially the interactions with NEMO, CK1 and β -TrCP) were then confirmed with E7 protein from 3 other high-risk alpha HPVs (HPV18, HPV33 and HPV39) frequently detected in cervical (pre)cancers as well as with E7 from 3 beta HPV genotypes (HPV8, HPV38 and HPV49) which have recently emerged as "facilitators" in UV-related cutaneous SCC development (for a Review, see Rollison et al. *J Virol* 2019). This information was added in the "Discussion" section.

The manuscript was written in such a manner that it was often difficult to follow the logic of experiments.

The difficulty to follow the logical flow of the study was also mentioned by the Reviewer 2 and is probably linked to the high number of experiments as well as to the various technological approaches used. As mentioned above, to address this issue, the manuscript was rigorously edited, the introduction was shortened and both the rationale of each experiment and the conclusions are now better explained.

Many facts or data are stated without specific reference to the experiments or the statistical comparison p-values in the written document. For instance, what was the evidence that keratinocytes stably maintained episomal HPV18 genomes (page 24)?

As requested, additional references to the experiments and statistical results (*p-values*) were added throughout the manuscript.

For the three-dimensional organotypic raft culture model used in our study, keratinocytes (NIKS cells) were first transfected with recircularized HPV18 DNA before being seeded onto a dermal equivalent composed of rat-tail type 1 collagen and 1×10^6 human foreskin fibroblasts. The procedure is detailed in the “Materials and Methods” section. As shown in Figure 3G, HPV DNA was detected by *in situ* hybridization and, consistent with an episomal infection, a diffuse punctate pattern was observed. This is in agreement with the strong expression of both HPV E2 and L1 as well as the E2/E6 ratio of 1 previously reported in this model by our collaborators (Lambert et al. *Methods in molecular medicine* 2005; Meuris et al. *Plos Pathogen* 2016). This information was added in the Figure 3 legend.

REVIEWER COMMENTS

Reviewer #1 (Remarks to the Author):

I appreciate the authors extensive reanalysis of the microbiome data and am overall satisfied with their new additions. Appropriate statistics have been applied and the microbiome data is far more clear in its presentation. The pH affect on the Lactobacillus experiment is also much more clear after this new presentation. Furthermore, I greatly appreciate the increase in sample size included for figure 8d and the data is much more convincing. Overall, I commend the authors for their substantial revisions to improve their data presentation.

Minor Critiques

I believe that Fig 7a still needs to be revised. It is very difficult to see a trend, even though some type of statistical significance is now indicated. It is not clear to me why the time aspect of the experiment is shown. I believe it is complicating the data. Can the authors just show the 6h and 9hour time points, similar to how they have done if Fig 6e.

I still have some issues with the figure presentation as many of the images are not clear. The irony of Figure 6a where it describes high-resolution image analysis of the petri dishes that are too low resolution to easily view, will not be received well as presented.

Line 649 Chao1 is misspelled.

An important note in the review document is that the authors referred to myself and the other reviewers as 'his'. I highly doubt that the editor revealed my gender, making this assumption very sexist and comes off as extremely unprofessional. I suggest the authors receive some training on maintaining an inclusive culture in STEM fields.

Reviewer #4 (Remarks to the Author):

Here the Authors sought to explore the relationship between HPV infection and persistence and microbial dysbiosis. The authors found some interesting results and where by HPV downregulates anti-microbial peptides, which can be used by Lactobacillus species as a nutrient source. This reduction of available nutrients for Lactobacillus species leads to reduction of these bacteria and allows the outgrowth of dysbiotic bacteria. Initially the authors did not analyze and interpret the microbiome data correctly or sufficiently. However in this new version they included colleagues who had greater experience in microbiology and have now correctly and sufficiently analyzed and interpreted the microbiome data. The authors have also reworked multiple figures and included a retrospective analysis of samples which enhance their conclusions..

Reviewer #5 (Remarks to the Author):

The authors conclude, based on a series of experiments, that High Risk HPV type infection can influence the cervicovaginal microbiome. The two-way influence of HPV and microbiome could be relevant to cervical carcinogenesis.

I have studied HPV infections and vaginal microbiome for many years, although not as an experimentalist. I would like to ask some fundamental questions based on molecular epidemiologic observations in women.

In long-term prospective studies, different HPV genomes are found or not found over time based on longitudinal cervicovaginal sampling. Microbiome measurements are also variable and time dependent, with women changing with repeated measurements. There are more than 40 different HPVs that are found, and they come and go with seeming independence. Within each type are multiple isolates that may occur or not separately. As we follow women with repeated measures, HPVs come and go linked to sexual activity that leads to co-transmission and immune responses that are clearly isolate specific. There is, in the absence of general immunodeficiency, no general pattern of all HPVs disappearing together. There is no particular clustering of types that antagonize or promote each other's presence.

In the midst of this dynamic, we observe some HPV isolate persisting long-term, apparently not effectively controlled. If that type is one of the major carcinogens, precancerous changes follow in a few years to several years. A clonal event has occurred with change from productive to transforming

infection. Persistence without intervening periods of disappearance leading to progression is more common with HPV16 and related types.

In direct observations, the microbiome shifts over time, and it only slightly/weakly impacts on the continued presence or absence of HPVs, and not in any global way or risk of progression.

This is directly relevant to the experiments reported by the authors. Microbiome is complex, time-dependent, and does not have a strong impact on HPV natural history and risk of cervical cancer. Whatever effect is seen is type dependent and idiosyncratic for that woman.

The mass of experimental data in the test animals is difficult to consolidate and interpret with the perspective that in humans, the whole phenomenon is so complex and non-generalizable. I was left thinking that this paper is not presenting direct evidence of the role of HPV on microbiome feeding back in turn on HPV to cause cancer. There is no clear pathway of that kind observed so far, so the experimentation might not reflect human experience.

I admit I do not know how to interpret the experimental associations given the limited ability to find a general role in HPV in women.

Reviewer #6 (Remarks to the Author):

The hypothesis that HPV exerts influence on the VMB is rational, and has been suggested by Torcia (Int J Mol Sci ; Lu Int J Clin Exp Med 2015; Moscicki Front Cell Infect Microbiol 2020) but study of this has not been systematically pursued and the series of experiments produce novel results.

Abstract.

- It is imprecise to state “a meta-analysis first revealed a significant association between these two common conditions”. The current analysis is preceded by the meta-analysis of Brusselsaers et al. (AJOG 2018), so that “association” between BV and HPV is demonstrated prior to the current meta-analysis. The meta-analysis by Brusselsaers et al included >100,000 women examining vaginal dysbiosis in relation to incident HPV (RR=1.43; 95% CI 1.10-1.85), HPV persistence (RR=1.14; 95% CI: 1.01-1.28), oncogenic HPV types (RR=1.18; 95% CI: 1.01-1.38) (and of course HSIL/CC, RR=2.01; 95%

CI: 1.40-3.01). These findings come from primarily prospective studies and results are stratified by microscopy studies and molecular studies.

- The abstract should indicate which results were derived from human vs. mouse models, given the current state of development of murine models for BV/human VMB.

Introduction

- Line 90. It's unexpected to call out Pasteurellaceae (family) rather than genus or species level taxa that are commonly associated with human BV (Prevotella, Mobiluncus, Sneathia, etc. – e.g., Onderdonk Clin Microbiol Rev 2016 or from HMP). Moreover, pasteurellaceae are not common or abundant in the human vagina.

- Line 95. Suggest consider “women of African descent”

- Line 101. “Aside from causing benign (but unattractive) symptoms” Advise strongly to change to “In addition to causing symptoms for some women, BV has been shown..” (e.g., Bilardi JE PLoS One 2013 and others)

- Lines 114-116. That the temporal sequence is “unknown” is overstated. There are numerous longitudinal studies, and meta-analyses show temporal association between BV and incidence of HPV (e.g., Brusselaers). What is limited, is longitudinal study to assess the occurrence of BV following HPV acquisition.

- The rationale for meta-analysis as necessary basis for these experiments is not explained in Introduction

Materials and Methods

- Meta-analysis:

o Could the authors please clarify: eligibility criteria included “reliable method for diagnosis of BV (e.g., Amsel or Nugent)”; were there any studies with molecular characterization? How were these considered? Studies using just clue cells seem to be included. Authors please justify its inclusion as a sole criterion for indicating BV.

o How were data extracted? Were studies extracted in duplicate? Was there quality assessment? How was data synthesized and analyzed?

- Retrospective cohort analysis:

o “Incomplete clinico-cytological information” is an exclusion criterion: please specify these variables are and how many records were excluded for missing data.

o Why was presence of clue cells used as the diagnosis of BV? This needs explanation. What is the sensitivity and specificity of clue cells alone (as opposed to in concert with the other Amsel criteria) for diagnosis of BV?

o How were these data analysed? Is HPV or BV status treated as a time varying variable? E.g., if a woman had HPV+BV- at baseline, and was later BV positive, was it confirmed that she wasn't HPV+ in the interim?

- Mouse models: How were the mice vaginal microbiota colonized? Pasteurellales is not common in human vagina.

- Line 685. Says "Bacterial stains" I think you mean "Bacterial strains"?

Results

- Meta-analysis: How does this add to the existing literature, if it is primarily cross-sectional studies and does not address the question of HPV preceding BV?

- Retrospective cohort analysis:

o This is important findings, as few studies examine the occurrence of new onset BV among women HPV+ vs. HPV-. However, the methods reported are brief, opening uncertainty in the validity and reliability of the results.

o It's not really "unclear" (line 163) that HPV and BV are "associated"; they are very clearly epidemiologically associated. The way in which they are biologically associated has uncertainties, yes. But this retrospective analysis does not shed light on this.

- As different hospitals are included, this is good for generalizability; but how many and does this affect the results, due to different patient catchment/referral patterns and practices? Is this a potential source of bias/variability/confounding?

o What is median time to BV infection for those with and without HPV? Are these results adjusted for age, race, or other potential confounders?

o What is meant by "persistence of BV"? How is this defined? Does it mean women were diagnosed with BV at least twice in 3 years? "BV" at 2 time points quite separate in time does not mean persistent infection throughout the entire time period; they could be separate episodes. Was antibiotic treatment provided on the basis of solely having clue cells?

- The finding that long-term estrogen does not modify the composition of the VMB. How generalizable/relevant is this finding? In light of (e.g., Amabebe Front Endocrinology 2018)

Discussion

- Line 466. Detection bias may be a likely explanation for the association of HPV and BV “persistence” being detected only at 2.5 years. Use of “undoubtedly” has an editorial tone.
- VMB compositional differences between humans and mice is superficially addressed (lines 471-472). It’s more than just the difference in lactobacilli.

Reviewer #7 (Remarks to the Author):

The authors provide evidence that HPV infection can provide an environment that disfavors Lactobacilli and leads to dysbiosis and BV. They use both retrospective cohort data and mechanistic studies on the action of HPV 16 E6/E7 proteins to support their hypothesis. Some concerns about the data and the content of the figures exist. I will only comment on the HPV studies.

1. I would eliminate the meta analysis. On lines 140-141 the overall prevalence is listed as 40% with a range of 8-87%. 40% is not believable and the range is ridiculous. In the retrospective study the cumulative positivity is 9.1% (l. 172-173) and that is a much more reasonable number.
2. Figures 3-8 are much too dense. For those of us who print out the paper they cause eye strain, or are just unreadable. For example Figure 5 has 19 parts A-S, most of which are multi panel. Move some parts of the figures to the Supplement and make the most important panels bigger.
2. E7 has been reported to bind many things but the data is not well controlled. The IPs in Figures 4J, 5M, 5 O-Q need to show IPs in the reciprocal direction not simply IPs with FLAG-E7. Use antibodies for each of the targets e.g NEMO, IκB etc.
3. In the experiment where you use the N- and C-terminal halves of E7, show a positive control for each half.
4. The degradation of IκB in Fig 4E is not convincing. Fig 4G is hard to see where the staining is occurring.
5. Figures 5 J and L are uninterpretable.

Point by point response ("HPV infection alters vaginal microbiome dynamic equilibrium through impairing host mucosa-Lactobacillus spp. mutualism", *Nature Communications*, NCOMMS-20-14429C)

First of all, we sincerely thank all Reviewers for their time spent with our manuscript and their highly valuable and fair comments which have been addressed in full. We hope that you will find this second round of revisions thorough and satisfactory.

Reviewers' comments:

Reviewer #1 (Remarks to the Authors):

I appreciate the authors extensive reanalysis of the microbiome data and am overall satisfied with their new additions. Appropriate statistics have been applied and the microbiome data is far more clear in its presentation. The pH effect on the Lactobacillus experiment is also much more clear after this new presentation. Furthermore, I greatly appreciate the increase in sample size included for figure 8d and the data is much more convincing. Overall, I commend the authors for their substantial revisions to improve their data presentation.

We thank the Reviewer for his/her constructive comments which allowed to significantly improve our article and to avoid any misinterpretation of presented results.

Minor Critiques

I believe that Fig 7a still needs to be revised. It is very difficult to see a trend, even though some type of statistical significance is now indicated. It is not clear to me why the time aspect of the experiment is shown. I believe it is complicating the data. Can the authors just show the 6h and 9hour time points, similar to how they have done in Fig 6e.

We understand the Reviewer's misunderstanding. Actually, in order to determine whether the beneficial effect of innate peptides constitutively expressed by the vaginal/cervical mucosa appeared directly or needed some latency, these latter were added to lactic acid bacteria and several time points were investigated. As shown in Figure 6A, a significant increased percentage of surviving colonies was only detected after 3 or 6h of incubation. These important observations allowed to exclude a passive/non-specific mechanism of action and to "guide" our next experiments (flow cytometry and mass spectrometry, Figure 6B-E) which demonstrated that these peptides are, in fact, cleaved, internalized and used by the predominant *Lactobacillus* species as amino acid source sustaining their growth/survival. In order both to keep the time aspect of the experiments and, as requested by the Reviewer, to better highlight the most important time points, the Figure 6A was revised and histograms (similar to Figure 5E) are now showed as well. Of note, the Figures were renumbered due to the suppression of the meta-analysis (former Figure 1).

I still have some issues with the figure presentation as many of the images are not clear. The irony of Figure 6a where it describes high-resolution image analysis of the petri dishes that are too low resolution to easily view, will not be received well as presented.

The present study contains 8 main Figures, 12 Supplementary Figures and 2 Supplementary Tables. As mentioned in the submission guide of the Journal, we incorporated both the manuscript file and Figures into a single PDF and kept the size under 30 MB (by reducing the resolution of Figures). However, all our Figures have an original resolution of 600 dpi which guarantees a high quality of presented results upon publication.

Line 649 Chao1 is misspelled.

This change was made in the manuscript.

An important note in the review document is that the authors referred to myself and the other reviewers as 'his'. I highly doubt that the editor revealed my gender, making this assumption very sexist and comes off as extremely unprofessional. I suggest the authors receive some training on maintaining an inclusive culture in STEM fields.

We totally agree with the Reviewer's comment and sincerely apologize if we accidentally offended one or several Reviewer(s). This is an unprofessional grammatical error without any sexist intention/purpose. This comment represents a warning to us and this mistake will not be reproduced in the future.

Reviewer #4 (who replaces previous Reviewer #2) (Remarks to the Authors):

Here the Authors sought to explore the relationship between HPV infection and persistence and microbial dysbiosis. The authors found some interesting results and where by HPV downregulates anti-microbial peptides, which can be used by Lactobacillus species as a nutrient source. This reduction of available nutrients for Lactobacillus species leads to reduction of these bacteria and allows the outgrowth of dysbiotic bacteria. Initially the authors did not analyze and interpret the microbiome data correctly or sufficiently. However in this new version they included colleagues who had greater experience in microbiology and have now correctly and sufficiently analyzed and interpreted the microbiome data. The authors have also reworked multiple figures and included a retrospective analysis of samples which enhance their conclusions.

We first thank the Reviewer for having accepted to replace Reviewer #2 and for having carefully read our manuscript/point by point responses. We thank the Reviewer for the concise summary of our major findings, for pointing out the novelty and importance of our results as well as the scientific rigor of our revisions.

Reviewer #5 (who replace previous Reviewer #3) (Remarks to the Authors):

The authors conclude, based on a series of experiments, that high risk HPV type infection can influence the cervicovaginal microbiome. The two-way influence of HPV and microbiome could be relevant to cervical carcinogenesis. I have studied HPV infections and vaginal microbiome for many years, although not as an experimentalist. I would like to ask some fundamental questions based on molecular epidemiologic observations in women.

In long-term prospective studies, different HPV genomes are found or not found over time based on longitudinal cervicovaginal sampling. Microbiome measurements are also variable and time dependent, with women changing with repeated measurements. There are more than 40 different

HPVs that are found, and they come and go with seeming independence. Within each type are multiple isolates that may occur or not separately. As we follow women with repeated measures, HPVs come and go linked to sexual activity that leads to co-transmission and immune responses that are clearly isolate specific. There is, in the absence of general immunodeficiency, no general pattern of all HPVs disappearing together. There is no particular clustering of types that antagonize or promote each other's presence.

In the midst of this dynamic, we observe some HPV isolate persisting long-term, apparently not effectively controlled. If that type is one of the major carcinogens, precancerous changes follow in a few years to several years. A clonal event has occurred with change from productive to transforming infection. Persistence without intervening periods of disappearance leading to progression is more common with HPV16 and related types.

In direct observations, the microbiome shifts over time, and it only slightly/weakly impacts on the continued presence or absence of HPVs, and not in any global way or risk of progression.

This is directly relevant to the experiments reported by the authors. Microbiome is complex, time-dependent, and does not have a strong impact on HPV natural history and risk of cervical cancer. Whatever effect is seen is type dependent and idiosyncratic for that woman.

The mass of experimental data in the test animals is difficult to consolidate and interpret with the perspective that in humans, the whole phenomenon is so complex and non-generalizable. I was left thinking that this paper is not presenting direct evidence of the role of HPV on microbiome feeding back in turn on HPV to cause cancer. There is no clear pathway of that kind observed so far, so the experimentation might not reflect human experience.

I admit I do not know how to interpret the experimental associations given the limited ability to find a general role in HPV in women.

We first thank the Reviewer for having accepted to replace Reviewer #3 and for having carefully read our manuscript/point by point responses. Three points raised by the Reviewer have to be clarified/discussed:

- 1) As mentioned by the Reviewer, about 40 different HPV genotypes can infect the gynecological tract and, most of the time (>90%), these infections are asymptomatic and disappear spontaneously within a few weeks. We totally agree with the Reviewer that these latter (which represent a significant percentage of HPV-positive women in most epidemiological studies) only weakly (and very likely not durably) influence the vaginal ecosystem. Consequently, these asymptomatic/spontaneously regressive infections reduce the acquired odds ratios and compel the investigators to follow a large number of patients during a few years for determining a potential interplay between HPV and bacterial vaginosis (BV). Indeed, it seems reasonable to think that the viral strain (e.g. HPV16), the persistence of the infection and the symptomatology are important parameters which influence the impact of HPV infection on cervico-vaginal microbiome. Despite these inevitable variabilities (or, in other words, that not all HPV infections can substantially alter the vaginal microbiome), it is interesting to notice that our retrospective follow-up study including more than 6,000 women clearly highlights an increased risk of developing a critical microbial imbalance (BV) in HPV-positive women compared to their uninfected counterparts. This point is now discussed in the manuscript.
- 2) Regarding both the complexity and the changes in the vaginal microbiota composition, we understand the Reviewer's comment and totally agree that all human

microbiomes are inherently dynamic. Although the vaginal ecosystem is associated with a low microbial diversity [largely dominated (>90%) by a few *Lactobacillus* species] compared to the skin and the gut, non-pathological, relatively mild and/or time-dependent variations are undeniably observed in each woman. However, the present study did not focus on these harmless/asymptomatic changes but on BV acquisition which represents a severe (and usually durable) microbial imbalance associated with several clinical manifestations (e.g. vaginal discharges, fishy odor, presence of clue cells,...) and consequences (e.g. increased risk of preterm delivery as well as gynecologic complications such as endometritis, cervicitis and postoperative pelvic infections). As shown in Figure 1 (retrospective follow-up study), this disorder was significantly linked to an increased risk of HPV persistence, supporting that an anaerobic bacteria-dominant vaginal microbiome can affect the HPV natural history as well as the subsequent risk of (pre)cancer development. These results confirm those recently published by Mitra et al. (*Nature Communications* 2020) demonstrating that type IV bacterial community states [as observed in the vast majority (>90%) of BV patients] are associated with CIN2 persistence and slower regression. This information is mentioned in both the “Introduction” and “Discussion” sections.

- 3) Regarding the *in vivo* model (K14-HPV16 transgenic mice) used in the present study, similarly to all animal models, it has strengths (e.g. the possibility to induce cervico-vaginal HPV-related carcinogenesis) and weaknesses (e.g. the bacterial diversity differences between humans and mice). It is, however, interesting to notice that, a dysbiosis of vaginal microbiome (characterized by significant alterations in bacterial diversity, richness and evenness) was detected in mice displaying a high-grade precancer (HSIL), reinforcing both the *in vitro* and follow-up data. Taken together, if the increased risk of BV acquisition in case of persistent/symptomatic HPV infection is constantly highlighted (despite the intrinsic variations of each model/experiment/technique/data collection), some might argue that this latter is all the more interesting. This information was added in the “Discussion” section.

Reviewer #6 (Remarks to the Authors):

The hypothesis that HPV exerts influence on the VMB is rational, and has been suggested by Torcia (Int J Mol Sci ; Lu Int J Clin Exp Med 2015; Moscicki Front Cell Infect Microbiol 2020) but study of this has not been systematically pursued and the series of experiments produce novel results.

We thank the Reviewer for pointing out the novelty and importance of our results.

Abstract

It is imprecise to state “a meta-analysis first revealed a significant association between these two common conditions”. The current analysis is preceded by the meta-analysis of Brusselsaers et al. (AJOG 2018), so that “association” between BV and HPV is demonstrated prior to the current meta-analysis. The meta-analysis by Brusselsaers et al included >100,000 women examining vaginal dysbiosis in relation to incident HPV (RR=1.43; 95% CI 1.10-1.85), HPV persistence (RR=1.14; 95% CI: 1.01-1.28), oncogenic HPV types (RR=1.18; 95% CI: 1.01-1.38) (and of course HSIL/CC, RR=2.01; 95% CI: 1.40-3.01). These findings come from primarily prospective studies and results are stratified by microscopy studies and molecular studies.

We agree with the Reviewer and this sentence was removed from our abstract.

Furthermore, as requested by the Reviewer 7 (and also strongly suggested by the Editor in his email to the authors), our meta-analysis evaluating the association between BV and HPV infection in studies published during the 2000 to 2020 time period was removed from our manuscript. Although we carefully assessed the quality of all included studies, we have to confess that publication biases regarding the diagnostic method for BV (Nugent score, Amsel criteria,...), the sensitivity of HPV tests (GP5+/6+, MY09/11, HC2, Linear array,...), the age and number of patients from study to study, the geographical regions (North America, Asia, Europa or Africa) or the study design (follow-up or cross sectional) are unavoidable. Actually, this point was already partially debated in two recent letters to the Editor related to the meta-analysis by Brusselaers et al. (AJOG 2019). Given that the conclusions of our meta-analysis were similar to those obtained by the other previously published ones (Gillet et al. *BMC Infect Dis* 2011; Brusselaers et al. *AJOG* 2019; Liang et al. *Infect Agent Cancer* 2019) and that, as mentioned by the present Reviewer, our retrospective cohort analysis provides more informative/original results, we agree that referencing the previously cited systematic reviews is sufficient to mention the context of our study and its novelty [retrospective clinical data, *in vitro* (pathology, microbiology, molecular biology) experiments, *in vivo* model]. The “Introduction”, “Results” and “Discussion” sections were revised accordingly.

The abstract should indicate which results were derived from human vs. mouse models, given the current state of development of murine models for BV/human VMB.

As requested, indications related to the human *versus* murine origin of results are now mentioned in the abstract.

Introduction

Line 90. It's unexpected to call out Pasteurellaceae (family) rather than genus or species level taxa that are commonly associated with human BV (Prevotella, Mobiluncus, Sneathia, etc. – e.g., Onderdonk Clin Microbiol Rev 2016 or from HMP). Moreover, pasteurellaceae are not common or abundant in the human vagina.

This change was made and bacterial genera commonly associated with human BV are now mentioned in the “Introduction” section. In addition, the review by Onderdonk et al. (*Clin Microbiol Rev* 2016) was added in the Reference list.

Line 95. Suggest consider “women of African descent”

This change was made in the manuscript.

Line 101. “Aside from causing benign (but unattractive) symptoms” Advise strongly to change to “In addition to causing symptoms for some women, BV has been shown..” (e.g., Bilardi JE PLoS One 2013 and others)

This change was made in the “Introduction” section.

Lines 114-116. That the temporal sequence is “unknown” is overstated. There are numerous longitudinal studies, and meta-analyses show temporal association between BV and incidence of HPV (e.g., Brusselaers). What is limited, is longitudinal study to assess the occurrence of BV following HPV acquisition.

We agree with the Reviewer. This information is now mentioned in the “Introduction” section.

The rationale for meta-analysis as necessary basis for these experiments is not explained in Introduction

As mentioned by the Reviewer (and already extensively discussed above), several meta-analyses evaluating the "association" between BV and HPV infection were recently published (e.g. Brusselaers et al. *AJOG* 2019). Therefore, our meta-analysis does not provide any extra information and no longer represents a necessary basis for all the experiments presented in the present manuscript. For this reason and others (differences between available studies in term of BV diagnosis, HPV testing, geographical regions,...), as requested by the Reviewer 7 (and also suggested by the Editor), the meta-analysis was removed from our manuscript. We totally agree that this change allows to further emphasize the importance/novelty of the retrospective cohort analysis (now presented in Figure 1).

Materials and Methods

- Meta-analysis:

Could the authors please clarify: eligibility criteria included “reliable method for diagnosis of BV (e.g., Amsel or Nugent)” ; were there any studies with molecular characterization? How were these considered? Studies using just clue cells seem to be included. Authors please justify its inclusion as a sole criterion for indicating BV.

In brief and just for the anecdote/transparency (given that the meta-analysis has been justifiably removed from the present paper), studies with the sole molecular characterization of BV were not included (we arbitrarily decided to use the microscopic examination of vaginal fluid as inclusion criterion). Despite its relatively low sensitivity (<50%), studies analyzing the sole presence of clue cells were taken into account considering the very high specificity (>95%) of this parameter for the diagnosis of BV (reported in many articles such as Sha et al. *J Clin Microbiol* 2005 or Bhujel et al. *BMC infect Dis* 2021). In these studies, we have to admit that the percentage of BV patients was very likely slightly underestimated.

How were data extracted? Were studies extracted in duplicate? Was there quality assessment? How was data synthesized and analyzed?

Similarly to the previous one, this comment is no longer necessary after the removal of the meta-analysis. Just for the record/transparency, retrieved studies were extracted in duplicate, independently reviewed and synthesized. Discrepancies were resolved by consensus. The "quality" was determined by ascertaining that the used HPV tests were sensitive and well-accepted (e.g. PCR-based assays or Hybrid Capture II test) and that reliable/specific methods for BV diagnosis were used (e.g. Amsel criteria or Nugent score). Finally, a funnel plot was performed and no obvious sign of asymmetry was observed, indicating that the publication bias for the included studies was well controlled.

- Retrospective cohort analysis:

“Incomplete clinico-cytological information” is an exclusion criterion: please specify these variables are and how many records were excluded for missing data.

“Incomplete clinico-cytological information” means the absence of HPV testing in case of abnormal cytological findings, lack of data related to patient age, vaginal bacterial flora and potential immunosuppressive conditions. In total, 32 patients (32/6,117, 0.52%) were excluded for missing data during the follow-up period. This information was added in the “Materials and Methods” section. Moreover, the exclusion criteria are now better defined.

Why was presence of clue cells used as the diagnosis of BV? This needs explanation. What is the sensitivity and specificity of clue cells alone (as opposed to in concert with the other Amsel criteria) for diagnosis of BV?

Interesting point. In the present retrospective cohort analysis [and overall, in the University Hospital of Liege (Belgium) for many years], BV is diagnosed by evaluating the presence of clue cells associated to a shift from a vaginal flora rich in *Lactobacilli* to a polymicrobial (anaerobic) microbiome. To do this, a simpler version of the Nugent’s grading score is routinely used. First described by Ison and Hay in 2002 (*Sex Transm Infect*), vaginal flora is divided into three categories: Grade I (normal, *lactobacillus spp.* morphotype only), Grade II (intermediate, reduced *lactobacillus spp.* morphotype and equal amount of mixed bacterial morphotypes) and Grade III (BV, mixed bacterial morphotypes with few or absent *lactobacillus spp.* morphotype). As previously shown (e.g. Ison and Hay *Sex Transm Infect* 2002; Chawla et al. *Biomed Res Intern* 2013; Antonucci et al. *Clin Invest* 2017), smears assessed as Grade II using Hay/Ison scoring system were mostly (>90%) found among patients diagnosed as normal using Nugent's or Amsel's criteria. Considering these results and, as mentioned above, the very high specificity (>95%) of clue cells for the diagnosis of BV (reported in many articles such as Sha et al. *J Clin Microbiol* 2005 or Bhujel et al. *BMC infect Dis* 2021), Grade II patients without clue cells were, therefore, classified as normal whereas their counterparts exhibiting clue cells were considered as BV-positive and grouped with Grade III patients. Finally, it is important to notice that the sensitivity, specificity and predictive value of positive/negative results with Hay/Ison classification system have been shown to be over 90% when Nugent’s or Amsel's score were used as the gold standard (e.g. Ison and Hay *Sex Transm Infect* 2002; Chawla et al. *Biomed Res Intern* 2013; Antonucci et al. *Clin Invest* 2017). Hence, compared to Nugent's method, Hay/Ison scoring system is less related to the slide reader’s expertise and allows to gain considerable time while displaying the same high performance for BV diagnosis. We apologize for omitting this information which is now clearly mentioned in the "Materials and Methods" section. We sincerely thank the Reviewer for his/her comment which allowed to significantly improve our work and its reproducibility as well as to avoid any misinterpretation of presented results.

How were these data analysed? Is HPV or BV status treated as a time varying variable? E.g., if a woman had HPV+BV- at baseline, and was later BV positive, was it confirmed that she wasn't HPV- in the interim?

At each follow-up visit, data related to a potential abnormal cytology result (Bethesda system), the existence of BV (Hay/Ison scoring system) and high-risk HPV infection (Abbott RealTime

High-Risk HPV assay) were available for all patients. Therefore, if a HPV+/BV- woman at first visit was later diagnosed with BV, we were able to ascertain that she was still HPV+ and that she was still infected with the same high-risk HPV genotype. Of note, eight HPV-/BV- patients at first visit were positive for both HPV and BV at the same follow-up visit. For them, it was impossible to determine which disorder developed first. This information was added in the “Results” section.

Mouse models: How were the mice vaginal microbiota colonized? Pasteurellales is not common in human vagina.

The gynecological tract of both control and K14-HPV16 mice was colonized naturally (mother's impact at birth and the environment). Vaginal microbiome was not influenced by the investigators. This information was added in the "Materials and Methods" section.

Regarding the bacteria belonging to the order of Pasteurellales, we totally agree with the Reviewer that they are not common in human vagina. As mentioned above (Reviewer 5's comments), similarly to all animal models, the *in vivo* model used in the present study has strengths (e.g. the possibility to induce cervico-vaginal HPV-related carcinogenesis) and weaknesses (e.g. the bacterial diversity differences between humans and mice). It is, however, interesting to notice that a dysbiosis of vaginal microbiome (characterized by significant alterations in bacterial diversity, richness and evenness) was detected in mice displaying a high-grade HPV-positive precancer (HSIL), reinforcing both the *in vitro* and follow-up data. Taken together, if the increased risk of BV acquisition in case of persistent/symptomatic HPV infection is constantly highlighted (despite the intrinsic variations of each model/experiment/technique/data collection), some might argue that this latter is all the more interesting. This point is now discussed in our manuscript.

Line 685. Says “Bacterial stains” I think you mean “Bacterial strains”?

This change was made in the manuscript.

Results

- Meta-analysis: How does this add to the existing literature, if it is primarily cross-sectional studies and does not address the question of HPV preceding BV?

We agree with the Reviewer that our meta-analysis was not essential. It confirmed previously published studies but, in contrast to our retrospective cohort analysis, did not (even partially) resolve novel questions. As already mentioned, the meta-analysis was justifiably removed from our manuscript.

- Retrospective cohort analysis:

o This is important findings, as few studies examine the occurrence of new onset BV among women HPV+ vs. HPV-. However, the methods reported are brief, opening uncertainty in the validity and reliability of the results.

We thank the Reviewer for pointing out the novelty and importance of our results. As requested (and already mentioned above), important details (e.g. criteria for BV diagnosis, patient's exclusion criteria, number of excluded patients for missing data,...) were added in the “Materials

and Methods” section in order to avoid any misinterpretation of results and improve their reproducibility.

o It's not really “unclear” (line 163) that HPV and BV are “associated”; they are very clearly epidemiologically associated. The way in which they are biologically associated has uncertainties, yes. But this retrospective analysis does not shed light on this.

This sentence was corrected in order to avoid any misunderstanding and better mention what is limited (most particularly the occurrence of BV following HPV acquisition).

As different hospitals are included, this is good for generalizability; but how many and does this affect the results, due to different patient catchment/referral patterns and practices? Is this a potential source of bias/variability/confounding?

Patients who underwent routine gynecological exams at the University Hospital of Liege or at its associated regional hospitals [Citadelle Regional Hospital (Liege, Belgium), ND Bruyeres (Chenee, Belgium), Regional Hospital of Huy, Bois de l'abbaye Hospital (Seraing, Belgium)] were eligible for the present retrospective cohort analysis. All samples are daily centralized at University Hospital of Liege where both HPV testing and microscopic examinations (by the same group of experienced cytologists) were performed. Therefore, the risk of bias/variability between specimens/patients is extremely limited. This information was added in the “Materials and Methods” section.

o What is median time to BV infection for those with and without HPV? Are these results adjusted for age, race, or other potential confounders?

The median time for BV development in HPV-infected and -uninfected patients was 34.53 and 59.44 months ($p < 0.0001$), respectively. These interesting/important results, further confirming the impact of HPV on vaginal microbiome equilibrium, are now mentioned in the "Results" section and shown in Figure 1. We thank the Reviewer for this comment which allowed to significantly improve our work.

The results presented in Figure 1 were not adjusted for age given that no significant difference were detected between the four patient groups. Regarding patients' ethnicity, we can reasonably suppose that most (>85%) were Caucasians but this information is not systematically mentioned in patients' medical records. This information was added in the "Results" section.

o What is meant by “persistence of BV”? How is this defined? Does it mean women were diagnosed with BV at least twice in 3 years? “BV” at 2 time points quite separate in time does not mean persistent infection throughout the entire time period; they could be separate episodes. Was antibiotic treatment provided on the basis of solely having clue cells?

Important/interesting point. Given that a high proportion (>50%) of women experience BV recurrence within 6-12 months following antibiotic therapy (e.g. Bradshaw et al. *J Infect Dis* 2006), in the present follow-up analysis, two consecutive negative tests were required to consider a patient really/durably cured of BV. This information was added in Figure 1 legend.

Regarding the second question, as mentioned above, BV was diagnosed using Hay/Ison scoring system. Grade II patients presenting clue cells as well as Grade III women were considered as

positive for BV (and received antibiotic treatment). Despite the very high specificity (>95%) of clue cells for the diagnosis of BV (reported in many articles), this parameter has a relatively low sensitivity (<50%), meaning that approximately half of patients having a critical change in the bacterial composition of their vaginal microbiome (and diagnosed BV-positive, Grade III) did not actually display any clue cells. Therefore, the presence of clue cells is, in the vast majority of cases, associated with BV diagnosis (and, consequently, with antibiotic treatment) but the absence of clue cells is absolutely not synonym of BV negativity.

The finding that long-term estrogen does not modify the composition of the VMB. How generalizable/relevant is this finding? In light of (e.g., Amabebe Front Endocrinology 2018)

We understand the misunderstanding. As judiciously requested above by the Reviewer (for the Abstract), the indication related to the human *versus* murine origin of results is here very important. This information was added in the "Results" section. Indeed, in women, estrogen promotes an increased thickness of the stratified squamous epithelium and its levels are correlated with a vaginal microbiome dominated by *Lactobacillus spp.* In contrast, as shown in our article and others (e.g. De Gregorio et al. *Scand J Lab Anim Sci* 2018), a lower abundance of *Lactobacilli* is observed in mice, very likely explaining the absence of long-term effect of estrogen on murine vaginal microbiome. Of note, these data, requested by the original Reviewer 3, are interesting and bring another information but are all in all not essential given that, for the results presented in Figure 7 (as well as in Supplementary Figure 11), both control (FVB/n) and K14-HPV16 transgenic mice were treated the same way in order to avoid any potential bias [subcutaneous implantation of a E2 implant, allowing reproducible long-term release of 17 β -estradiol which is necessary for (pre)cancer development within the lower genital tract in this specific HPV-dependent model (Arbeit et al. *PNAS* 1996; Elson et al. *Cancer Res* 2000)]. To avoid any misunderstanding, this information was also added in the "Results" section.

Discussion

Line 466. Detection bias may be a likely explanation for the association of HPV and BV “persistence” being detected only at 2.5 years. Use of “undoubtedly” has an editorial tone.

The word “undoubtedly” was removed from this sentence and, as suggested, replaced by “likely”.

VMB compositional differences between humans and mice is superficially addressed (lines 471-472). It’s more than just the difference in lactobacilli.

As requested, other differences between human and murine composition of vaginal microbiome were added in the “Discussion” section.

Reviewer #7 (Remarks to the Authors):

The authors provide evidence that HPV infection can provide an environment that disfavors Lactobacilli and leads to dysbiosis and BV. They use both retrospective cohort data and mechanistic studies on the action of HPV 16 E6/E7 proteins to support their hypothesis. Some concerns about the data and the content of the figures exist. I will only comment on the HPV studies.

1. I would eliminate the meta analysis. On lines 140-141 the overall prevalence is listed as 40% with a range of 8-87%. 40% is not believable and the range is ridiculous. In the retrospective study the cumulative positivity is 9.1% (l. 172-173) and that is a much more reasonable number.

As requested by the Reviewer (and also strongly suggested by the Editor in his email to the authors), the systematic review evaluating the association between BV and HPV infection in studies published during the 2000 to 2020 time period was removed from our manuscript. We totally agree with Reviewer's comment that, similarly to the other previously published ones (Gillet et al. *BMC Infect Dis* 2011; Brusselaers et al. *AJOG* 2019; Liang et al. *Infect Agent Cancer* 2019), our meta-analysis suffers from the many differences between available studies [study design (follow-up or cross sectional), HPV detection (GP5⁺/6⁺, MY09/11, HC2, Linear array,...), BV diagnosis (Nugent score, Amsel criteria or clue cells alone), number of analyzed patients (from 25 to 10,546), age (from 18 to 84), geographical region (North America, Asia, Europa or Africa),...]. Overall, these collected data are very difficult to group together and the results from our retrospective study (n=6,085) are certainly more controlled/informative. Moreover, both the acquisition and persistence of BV in case of presence or absence of HPV infection have been analyzed (which is not systematically pursued in the vast majority of studies available in the literature).

2. Figures 3-8 are much too dense. For those of us who print out the paper they cause eye strain, or are just unreadable. For example Figure 5 has 19 parts A-S, most of which are multi panel. Move some parts of the figures to the Supplement and make the most important panels bigger.

In the last few months, we read many health science articles recently published in *Nature Communications* and multiple panels (n>10) are very frequently presented in the main Figures. In addition, we rigorously followed the author guidelines to create the Figures and we guarantee that the resolution, length/width as well as the typeface (Arial) and the size (between 5pt and 8pt) are appropriate/valid. However, we understand the rationale of Reviewer's comment and agree that some panels [especially in the Figure 5, renumbered Figure 4 due to the suppression of the meta-analysis (former Figure 1)] may be part of the supplemental data. Therefore, in order to address as appropriately as possible the Reviewer's comment (and to still follow the author guidelines), when possible, the font size was increased [e.g. in the Figure 2 (legends of graphs B and C), Figure 5 (legend of graph B), Figure 6 (legend of graphs B, C and D), all WB/co-IP quantifications (Figures 3 and 4 as well as Supplementary Figures 4, 5, 6, 7, 8),...], the blots were enlarged (Figure 4) and several graphs/blots were displaced in Supplementary Figure 6.

3. E7 has been reported to bind many things but the data is not well controlled. The IPs in Figures 4J, 5M, 5 O-Q need to show IPs in the reciprocal direction not simply IPs with FLAG-E7. Use antibodies for each of the targets e.g NEMO, IκB etc.

As requested, IP-validated antibodies recognizing the three novel E7 targets highlighted in the present study were purchased [anti-NEMO (ab178872, Abcam); anti-CK1 (ab206652, Abcam) and anti-β-TrCP (clone D13F10, Cell Signaling)]. Already detected by both GPCA and co-IP using the anti-FLAG antibody for the immunoprecipitation, the E7-NEMO, E7-CK1 and E7-β-TrCP interactions were further confirmed by co-IP experiments in the inverse direction. This information was added in both the "Materials and Methods" and "Results" sections and the blots are shown in Supplementary Figure 4. Overall, considering the use of two different techniques

(GPCA and co-IP), the characterization with E7 from several genotypes (high-risk alpha: HPV16, 18, 33 and 39; beta: HPV8, 38 and 49) as well as the observed "biological" consequences [reduced stability/half-life of NEMO and CK1, inhibition of CK1-dependent phosphorylation of β -catenin (ser45)], we think that we can reasonably affirm that these newly discovered E7 targets are "real"/valid and very likely important for viral persistence.

4. In the experiment where you use the N- and C-terminal halves of E7, show a positive control for each half.

We thank the Reviewer for his/her comment. Actually, PTPN14 and Rb1 were systematically used as positive control for binding to the C-terminal region and the LxCxE motif (included in the N-terminal half) of E7, respectively. The results are shown in Supplementary Figure 6 and this information was added in the legend.

5. The degradation of I κ B in Fig 4E is not convincing. Fig 4G is hard to see where the staining is occurring.

As requested, Western blotting experiments analyzing the degradation of I κ B (following TNF α addition) in keratinocytes stably transduced or not with HPV16 E6 or E7 have been reproduced and the results are shown in Figure 3E (former Figure 4E). Regarding the Figure 3G (former Figure 4G), the absence of colocalization between p65 and Dapi in E7-transduced cells (cytoplasmic sequestration of p65) should be noticed. This information was added in Figure 3 legend. In order to facilitate the visual representation, the Dapi and Phalloidin stainings were also inverted in the panel.

6. Figures 5 J and L are uninterpretable.

We agree with Reviewer's comment. In order to facilitate the interpretation of both Figure 4J and L (former Figure 5J and L), important additional details were added in both the Figure and its legend.

REVIEWER COMMENTS

Reviewer #1 (Remarks to the Author):

The authors have satisfied my critiques.

Reviewer #6 (Remarks to the Author):

Thank you for removing the meta-analysis.

Thank you for clarifying the mouse results in the abstract.

Thank you for modifications in Introduction.

Thank you for clarification on the classification of BV (Grade III, or Grade II with clue cells). This is much more precise, and more importantly - replicable.

Thank you for adding median time to BV infection; I think this is helpful information for readers.

Retrospective analysis:

I appreciate the responses and additions for more reproducible/valid results. A few questions remain.

1. Lines 546-549: While it is now clear how many women in the analytic sample were excluded with missing data during the follow-up (n=32), it is still not clear to what extent the analytic sample may or may not be representative of women with BV and/or women with abnormal cytological findings.

How many women were excluded from the analytic sample due having HPV testing/abnormal cytological findings but lack of data related to vaginal bacterial flora?

2. Analysis of the retrospective data is still lacking some details. While authors have explained their ability to classify outcome over time, it was previously asked, and now again, how the data was analyzed. It does not seem to incorporate a longitudinal approach. I assume logistic regression is used since odds ratios are reported, but Figure 1E shows Kaplan Meier curves.

3. Regarding the inclusion of multiple hospitals, thank you for adding the explanation of the centralization of labs; this builds confidence that variability/potential bias in lab assays is minimized. However, centralization of lab assays doesn't translate to reduced variability between patients in terms of exposures or outcomes. One could control for hospital in the analysis, or assess random effect/ cluster based variance. It could be handled analytically in a way that assesses/reduces bias.

4. It should be noted as a limitation that results are not adjusted for age, race, or other potential confounders (e.g., SES, HIV status, etc.).

5. Previous questions on definition of "persistence of BV" and provision of treatment have not been answered satisfactorily. Authors have repeated the definition of BV diagnosis in their response (Hay/Ison), which was not provided as the basis of diagnosis in the previous version, causing confusion. Authos explain that two consecutive negative tests are required for "durably" cured BV, based on the high recurrence rate within 6-12 months. What is the timing of these two negative test points? Do the observations need to have two negative tests occurring over at least 6 months time period? Or 12 months time period? This is still not a replicable definition of persistence/cure.

6. In the previous submission, it was asked whether treatment was provided on the basis of clue cells - *because* the definition of BV was previously reported to be based on clue cells. In the previous version it read: "BV was diagnosed by evaluating the presence of clue cells associated to a shift from a vaginal flora rich in Lactobacilli to a polymicrobial (anaerobic) microbiome." Therefore, based on this - that definition of BV was based on clue cells - it was asked if treatment was provided on clue cells.

But what I am asking, is what is the basis of treatment for BV in this cohort? Were they treated based on the Hays/Ison criteria? Is there documentation of whether women are treated, and

whether this differs by HPV status or other factors? This is important for understanding natural history in this context and what may be a persistent/incident infection.

Reviewer #7 (Remarks to the Author):

The revised manuscripts generally address my concerns.

The authors should not say that their IPs demonstrate a direct interaction. Proteins can IP as part of a complex; direct interactions can only be proven with purified proteins, not cell lysates.

A very specific binding motif has been shown by others to be required for b-TRCP binding. Is this motif present in E7? If not how to explain the binding?

Point by point response ("HPV infection alters vaginal microbiome dynamic equilibrium through impairing host mucosa-*Lactobacillus* spp. mutualism", *Nature Communications*, NCOMMS-20-14429D)

Once again, we sincerely thank all Reviewers for their time spent with our manuscript and their highly valuable and fair comments which have been addressed in full. We hope that you will find this third round of revisions thorough and satisfactory.

Reviewers' comments:

Reviewer #1 (Remarks to the Authors):

The authors have satisfied my critiques.

We thank the Reviewer for his/her constructive and fair comments which allowed to significantly improve our work.

Reviewer #6 (Remarks to the Authors):

Thank you for removing the meta-analysis.

Thank you for clarifying the mouse results in the abstract.

Thank you for modifications in Introduction.

Thank you for clarification on the classification of BV (Grade III, or Grade II with clue cells).

This is much more precise, and more importantly - replicable.

Thank you for adding median time to BV infection; I think this is helpful information for readers.

We sincerely thank the Reviewer for his/her comments which allowed to significantly improve our work and its reproducibility as well as to avoid any misinterpretation of presented results.

Retrospective analysis:

I appreciate the responses and additions for more reproducible/valid results. A few questions remain.

1. Lines 546-549: While it is now clear how many women in the analytic sample were excluded with missing data during the follow-up (n=32), it is still not clear to what extent the analytic sample may or may not be representative of women with BV and/or women with abnormal cytological findings. How many women were excluded from the analytic sample due having HPV testing/abnormal cytological findings but lack of data related to vaginal bacterial flora?

In order to address the Reviewer's question, we carefully re-examined the viral-cytological data of all women who underwent at least 2 routine cervical Pap smear during the period 2010-2018 at the University Hospital of Liege (Belgium) or at its associated regional hospitals. Importantly, we did not find any patient having a positive HPV testing/abnormal cytological findings and a lack of data related to vaginal bacterial flora, clearly supporting that the analytic cohort is representative of women with BV and/or abnormal cytological findings. Actually, these results can easily be explained by the fact that, during a decade, the cytologists working at the University Hospital of Liege have systematically assessed the vaginal bacterial flora (using Hay/Ison

grading system) and mentioned the collected results in patients' records (even in absence of a specific request by the gynecologists). This information was added in the "Materials and Methods" section.

2. Analysis of the retrospective data is still lacking some details. While authors have explained their ability to classify outcome over time, it was previously asked, and now again, how the data was analyzed. It does not seem to incorporate a longitudinal approach. I assume logistic regression is used since odds ratios are reported, but Figure 1E shows Kaplan Meier curves.

In order to determine the probability/risk of acquiring one condition (HPV infection or BV development) when the other one was already present or not, both odds ratios and corresponding 95% confidence intervals were calculated using a Fisher's exact test (contingency table) and the results were visualized in a Forest plot (Figure 1C). To further evaluate the interplay between HPV and BV, the persistence of each pathological disorder in presence or absence of the other one was then compared using a log-rank (Mantel-Cox) test. The results were presented as Kaplan-Meier plots (Figure 1E-F). This information is now clearly mentioned in the "Materials and Methods" section. Overall, similarly to what we did for the other Figures, all statistical analyses related to the retrospective cohort analysis were verified/validated by Dr. Olivier Peulen (Metastasis Research Laboratory, University of Liege) who is warmly thanked in the manuscript ("acknowledgements" section). Among others, Dr Peulen gives bioinformatics/statistics courses at the University of Liège (Faculty of Medicine).

3. Regarding the inclusion of multiple hospitals, thank you for adding the explanation of the centralization of labs; this builds confidence that variability/potential bias in lab assays is minimized. However, centralization of lab assays doesn't translate to reduced variability between patients in terms of exposures or outcomes. One could control for hospital in the analysis, or assess random effect/ cluster based variance. It could be handled analytically in a way that assesses/reduces bias.

We sincerely think that this comment is difficult to address with precision, especially when using the different hospitals as variables. Indeed, some gynecologists consult in several different hospitals and women do not always go to the closest hospital, inducing biases for investigating potential clusters (or, in other words, for determining the variability between patients in terms of exposures or outcomes). To overcome this problem and to respond as precisely as possible to the Reviewer's comment, we decided to collect the zip code of patients from individual health records. The data collection was previously approved by the Ethics Committee of the University Hospital of Liege (#2021-117). This information was available for the large majority of enrolled women (6,045/6,085, 99.3%) and the geographic repartition of patients in the four defined groups (HPV/BV⁻, HPV⁺/BV⁻, HPV/BV⁺, HPV⁺/BV⁺) was traced on maps. The distribution of patients who acquired BV or became positive for HPV during the follow-up period was also determined. Interestingly, the large majority of patients lived within a 50km radius around Liege (Belgium) and no obvious difference in terms of geographic repartition of patients was noticed between the groups, suggesting that these latter can be compared with each other due to similar/close exposures or outcomes. These data are now mentioned in the "Results" section and presented in Supplementary Figure 1. Of note, the Supplementary Figures were renumbered due to the addition of these results.

4. *It should be noted as a limitation that results are not adjusted for age, race, or other potential confounders (e.g., SES, HIV status, etc.).*

As requested, this information was added in the “Results” section. In addition, these sources of potential imprecision (e.g. the lack of data about the socioeconomic status of patients) are now also discussed in our manuscript. This latter information as well as others already mentioned (e.g. inclusion/exclusion criteria, sample centralization and analysis, number of patients, BV diagnosis, HPV test, missing data during the follow-up, research ethics approval,...) or added in the text during this reviewing process [e.g. the mean follow-up time, the used statistical test (see above), the geographic repartition of patients (see above),...] allow to ensure that this retrospective cohort analysis includes as many items as possible reported by the STROBE statement/checklist (as suggested by the Editor in the email to the authors).

5. *Previous questions on definition of “persistence of BV” and provision of treatment have not been answered satisfactorily. Authors have repeated the definition of BV diagnosis in their response (Hay/Ison), which was not provided as the basis of diagnosis in the previous version, causing confusion. Authors explain that two consecutive negative tests are required for “durably” cured BV, based on the high recurrence rate within 6-12 months. What is the timing of these two negative test points? Do the observations need to have two negative tests occurring over at least 6 months time period? Or 12 months time period? This is still not a replicable definition of persistence/cure.*

We agree with the Reviewer. We forgot to mention the duration period between the two negative tests. Actually, two consecutive negative results (using Hay/Ison grading system) at least 12 months apart were required to consider a patient really/durably cured of BV. This important information was added in Figure 1 legend.

6. *In the previous submission, it was asked whether treatment was provided on the basis of clue cells - *because* the definition of BV was previously reported to be based on clue cells. In the previous version it read: “BV was diagnosed by evaluating the presence of clue cells associated to a shift from a vaginal flora rich in Lactobacilli to a polymicrobial (anaerobic) microbiome.” Therefore, based on this - that definition of BV was based on clue cells - it was asked if treatment was provided on clue cells.*

But what I am asking, is what is the basis of treatment for BV in this cohort? Were they treated based on the Hays/Ison criteria? Is there documentation of whether women are treated, and whether this differs by HPV status or other factors? This is important for understanding natural history in this context and what may be a persistent/incident infection.

Firstly, we totally agree with the Reviewer that the definition of BV diagnosis was confusing in the initial version of our manuscript. This comment was extensively addressed previously (second round of revisions) and the routine use of the Hay/Ison grading system for BV diagnosis is now clearly mentioned in both the "Materials and Methods" and "Results" sections. Once again, we thank the Reviewer for his/her comment which allowed to significantly improve our work and its reproducibility as well as to avoid any misunderstanding of presented results.

Secondly, another interesting point was raised by the Reviewer. In order to address this comment, in parallel to collecting the zip code of all patients, we looked for potential treatments following BV diagnosis in patients' health records. When the information was available (in contrast to the

diagnosis which is systematically reported, the treatment prescriptions are not always mentioned, especially for the "common" conditions diagnosed almost 10 years ago), BV women were typically (>90%) treated with antibiotics (metronidazole or clindamycin). Importantly, we did not notice any difference in treatment according to HPV status. The follow-up of HPV-positive women did not differ by BV status either, supporting that the results presented in our study cannot be explained by differences in patient management between defined groups. This important information was added in the "Discussion" section.

Reviewer #7 (Remarks to the Authors):

The revised manuscripts generally address my concerns.

We thank the Reviewer for his/her comments which allowed to significantly improve our article.

The authors should not say that their IPs demonstrate a direct interaction. Proteins can IP as part of a complex; direct interactions can only be proven with purified proteins, not cell lysates.

We agree with the Reviewer that our IPs (showed in Figures 3 and 4 as well as in Supplementary Figures 4, 5, 6 and 8) do not demonstrate a direct interaction between HPV E7 and NEMO, CK1 and β -TrCP. However, in the present study, these three novel E7 targets were also confirmed/validated by *Gaussia princeps* luciferase complementation assay (GPCA) which is a very sensitive method to identify binary protein-protein interactions in mammalian cells (Cassonnet et al. *Nature Methods* 2011 ;Neveu et al. *Methods* 2012). In this assay, expression vectors (pSPICA-N1 and N2) allow the expression of two complementary fragments of the *Gaussia princeps* luciferase (Gluc1 or Gluc2) linked to the N-terminal ends of tested proteins by a flexible hinge polypeptide of 20 amino acid residues. Given the small size of the hinge polypeptide, the direct interaction between two proteins is generally considered as required for the enzyme reconstitution (and, ultimately, the emission of luminescence following luciferin addition). Moreover, in this assay, the two tested proteins are overexpressed compared to potential endogenous extra partners, preventing the reconstitution of most protein complexes (and, therefore, the emission of luminescence) in case of requirement of other protein partner(s). Taken together, with the positive results also collected by GPCA, we think that we can reasonably affirm that these newly discovered E7 targets are "real"/valid and very likely interact directly with the viral oncoprotein. As requested, the word "direct" was removed from our manuscript and this latter information was added in the "Discussion" section.

A very specific binding motif has been shown by others to be required for β -TRCP binding. Is this motif present in E7? If not how to explain the binding?

As mentioned by the Reviewer, previous studies have demonstrated that a putative DSG(X)_{2+n}S motif is an important characteristic of protein substrates for the ubiquitin E3 ligase β -TrCP (involved in the binding and subsequent target protein degradation) (Busino et al. *Nature* 2003). As requested, we examined the sequence of HPV E7 from different genotypes for the presence of a potential DSG motif [and more particularly the C-terminal region which has been shown to interact with β -TrCP (results presented in Figure 4N)]. Interestingly, nothing was found, supporting that the viral oncoprotein is actually not a substrate for β -TrCP but, in the opposite, E7 binds to this latter protein, impeding its activity. Therefore, in parallel to both NF- κ B and Wnt/ β -

catenin signaling impairment related to E7-NEMO and E7-CK1 bindings (Figures 3 and 4), the interaction between β -TrCP and HPV E7 oncoprotein could not only further increase β -catenin half-life but also participate to NF- κ B inhibition due to the fact that β -TrCP has been shown to be involved in the degradation of both β -catenin and I κ B α (Kanarek et al. *Immunol Rev* 2012; Liu *PNAS* 1999). This information is now clearly mentioned in the “Discussion” section.

REVIEWERS' COMMENTS

Reviewer #6 (Remarks to the Author):

Thank you for these clarifications. It is much appreciated the definitions and additional scrutiny of the retrospective data, to the extent possible, given the importance of the findings.

My questions have been addressed.

Point by point response ("HPV infection alters vaginal microbiome through down-regulating host mucosal innate peptides used by *Lactobacilli* as amino acid sources", *Nature Communications*, NCOMMS-20-14429E)

Reviewers' comments:

Reviewer #6 (Remarks to the Author):

Thank you for these clarifications. It is much appreciated the definitions and additional scrutiny of the retrospective data, to the extent possible, given the importance of the findings. My questions have been addressed.

Once again, we sincerely thank the Reviewer for his/her constructive comments which allowed to significantly improve our work.